# Invariance-Aware Randomized Smoothing Certificates

**Jan Schuchardt**
Technical University of Munich
`j.schuchardt@tum.de`

**Stephan Günnemann**
Technical University of Munich
`s.guennemann@tum.de`

## Abstract

Building models that comply with the invariances inherent to different domains, such as invariance under translation or rotation, is a key aspect of applying machine learning to real world problems like molecular property prediction, medical imaging, protein folding or LiDAR classification. For the first time, we study how the invariances of a model can be leveraged to provably guarantee the robustness of its predictions. We propose a gray-box approach, enhancing the powerful black-box randomized smoothing technique with white-box knowledge about invariances. First, we develop gray-box certificates based on group orbits, which can be applied to arbitrary models with invariance under permutation and Euclidean isometries. Then, we derive provably tight gray-box certificates. We experimentally demonstrate that the provably tight certificates can offer much stronger guarantees, but that in practical scenarios the orbit-based method is a good approximation.

## 1 Introduction

It is well-established that machine learning models are susceptible to adversarial attacks [1–4]. Even without malevolent actors, adversarial attacks can be considered worst case scenarios in environments with noisy, erroneous or otherwise corrupted data, thus necessitating robust machine learning methods.

Invariance is a central design principle that has so far received little dedicated attention in the realm of adversarially robust machine learning. Over the past decades, there has been ongoing research into developing machine learning models that comply with the invariances inherent to different data types and tasks. Prominent recent examples include Deep Sets [5], PointNet [6], Group Equivariant CNNs [7], Spherical CNNs [8] and Graph Convolutional Networks [9], but the study of invariant models significantly precedes the surge in popularity of deep learning methods [10–15].

For the first time, we explore the following question: *Can a-priori knowledge about invariances be leveraged in deriving provable guarantees for a model's robustness to adversarial attacks?*

Going by a loose categorization of prior work, we could adopt one of two possible approaches for our exploration: A white-box or black-box one. White-box certificates (e.g. [16–23]) analyze a model's internals, such at its weights and non-linearities, to provably guarantee that a prediction does not change under adversarial attack. Black box certificates – specifically randomized smoothing [24–26] – use statistical methods to provide provable guarantees that hold for all models sharing the same prediction probabilities under a random input distribution, irrespective of their internals.

We opt for a randomized smoothing approach, as it allows us to focus on the interplay between invariances and robustness, rather than the specific means of implementing these invariances. By combining white-box knowledge about invariances with black-box knowledge about prediction probabilities we obtain *gray-box* certificates. We first derive a gray-box certificate that follows a post-processing paradigm: It takes an existing black-box certificate and augments the certified region using information about the model's invariances. This *orbit-based* certificate serves as a baseline for the provably tight gray-box certificates we derive in subsequent sections.

36th Conference on Neural Information Processing Systems (NeurIPS 2022).

For our exploration, we focus on models operating on spatial data, rather than structured data (e.g. images and sequences) – both because spatial symmetries can be elegantly formalized using algebraic concepts and because there is an ongoing trend towards using machine learning for real-world applications with inherent spatial invariances, e.g. molecular property prediction [27–34], LiDAR classification [35–38], drug discovery [39–41], particle physics [42–44] and protein folding [45–49].

Our main contributions are

- the first study on the interplay of invariance and certifiable robustness,
- a principled method for deriving tight invariance-aware randomized smoothing certificates,
- tight certificates for models invariant to translations and rotations.

We further demonstrate experimentally that the orbit-based certificates offer a good approximation of our tight certificates, if the variance of the smoothing distribution is small.

## 2  Related work

**Invariant machine learning.** Given the diversity of approaches to invariant machine learning, and the fact that our approach is model-agnostic, we refrain from attempting a survey and instead refer to [50] for a principled, high-level introduction into the realm of learning with invariances and equivariances.

**Invariance and robustness.** Two recent empirical studies [51, 52] demonstrate that data augmentation meant to increase robustness to $\ell_p$-norm adversarial attacks reduces robustness to semantically meaningful transformations (e.g. rotation) and vice-versa, suggesting an inherent invariance-robustness trade-off. However, neither study models that are invariant by design. In [53], the negative effect of translation-invariance on the robustness of image classifiers is investigated. Note that our work is not meant to resolve potential trade-offs, but to tightly bound the actual robustness of models.

**Gray-box certificates.** While we propose the first gray-box certificate for invariant models, there exists prior work on combining white-box knowledge with black-box certification. In [54] and [55], knowledge about a classifier's gradients is used to derive tighter randomized smoothing certificates. In [56], knowledge about a graph neural network's receptive fields is used to derive collective gray-box certificates for multiple predictions. In [57], the message-passing scheme of graph neural networks is used for gray-box certification against adversaries that control all features of multiple nodes.

**Adversarial attacks on spatial data.** Adversarial attacks on spatial data either modify [58–66], insert [65, 67, 68], or delete [69–71] points in space and have been particularly actively studied for point cloud-classification. While the earliest work simply adopted gradient-based attacks from the image domain [58], more recent work has developed a rich assortment of domain-specific methods, for example to preserve object smoothness [64, 72], leverage critical points [69, 70] or craft physically realizable attacks (e.g. to attack models through LiDAR sensors) [61, 73–75]. Of particular note are attacks via isometries (e.g. rotations and translations) [76, 77], whose effectiveness motivates the use of invariant models. Note that we certify the robustness of invariant models to arbitrary point modification attacks – not just isometry attacks (see also "orthogonal research directions" below). Like in other domains, empirical defenses have been proposed [59, 78, 79] and subsequently broken [80–82], motivating the development of black-box [83–85] and white-box [86] robustness certificates for spatial data. It should be noted that Gaussian randomized smoothing, without invariance information, has already been used in prior work – either as a baseline [83] or as a special case of the respective certificate [85].

**Orthogonal research directions.** One related but orthogonal research direction is transformation-specific certification [23, 85–93]. There, a model is assumed to potentially change its prediction under adversarial parametric transformations (e.g. rotations) and one certifies robustness for specific parameter ranges. White-box methods [23, 86–89] over-approximate the set of inputs reachable by a transformation and then propagate it through a model using existing white-box techniques. Black-box methods – namely transformation-specific randomized smoothing [90, 94] – randomize the transformation parameters to provide robustness guarantees for arbitrary models via statistical methods. Later work generalized this principle to spatial data [85], vector-field deformations [92] and multiplicative parameters [93]. Different from these methods, we assume our model to be invariant under a set of transformations, i.e. never change its prediction. We use this property as a tool for certification against arbitrary perturbations. Aside from that, there exist white-box certification techniques for specific *operations with invariances* (e.g. global max-pooling [86], message passing [95, 96] and batch normalization [97]). But prior work does not use or even discuss this property – it treats these invariant

operations as coincidental building blocks of the models it is trying to certify. Our work is the first to study *invariance itself* in the context of provable robustness and how to leverage it for certification.

## 3 Background

### 3.1 Randomized smoothing

Randomized smoothing is a black-box certification technique that can be adapted to various data types, tasks and threat models [98–103, 91, 104–106]. Instead of directly certifying a classifier $g$, it constructs a smoothed classifier $f$ that returns the most likely prediction under random perturbations of its input. It then certifies the robustness of this smoothed classifier. We present the tight Gaussian smoothing certificate derived by Cohen et al. [26] and its generalization to matrix data [83, 85].

Assume a continuous $(N \times D)$-dimensional input space $\mathbb{R}^{N \times D}$, label set $\mathbb{Y}$ and *base classifier* $g : \mathbb{R}^{N \times D} \to \mathbb{Y}$. Let $\mu_{\boldsymbol{X}}(\boldsymbol{Z}) = \prod_{d=1}^{D} \mathcal{N}\left(\boldsymbol{Z}_{:,d} \mid \boldsymbol{X}_{:,d}, \sigma^2 \mathbf{I}_N\right)$ be the isotropic matrix normal distribution with mean $\boldsymbol{X}$ and standard deviation $\sigma$. Let $p_{\boldsymbol{X},y} = \Pr_{\boldsymbol{Z} \sim \mu_{\boldsymbol{X}}}[g(\boldsymbol{Z}) = y]$ be the probability of $g$ predicting class $y$ under this *smoothing distribution*. One can then define a *smoothed classifier* $f(\boldsymbol{X}) = \mathrm{argmax}_{y \in \mathbb{Y}} p_{\boldsymbol{X},y}$ that returns the most likely prediction of $g$ under $\mu_{\boldsymbol{X}}$.

Let $y^* = f(\boldsymbol{X})$ be a smoothed prediction and $\boldsymbol{X}' = \boldsymbol{X} + \boldsymbol{\Delta}$ a perturbed input. One can show $f(\boldsymbol{X}') = y^*$ by proving that, for perturbed input $\boldsymbol{X}'$, $y^*$ is more likely than all other classes combined. That is, $p_{\boldsymbol{X}',y^*} > 0.5$. A tighter certificate can be obtained by proving that $y^*$ is more likely than the second most likely class, i.e. $p_{\boldsymbol{X}',y^*} > \max_{y' \neq y^*} p_{\boldsymbol{X}',y'}$. For the sake of exposition, we use the first approach throughout the main text and generalize all results to the second one in Appendix H. One can lower-bound $p_{\boldsymbol{X}',y^*}$ by finding the *worst-case classifier* from a set of functions $\mathbb{H}$ with $g \in \mathbb{H}$:

$$p_{\boldsymbol{X}',y^*} \geq \min_{h \in \mathbb{H}} \Pr_{\boldsymbol{Z} \sim \mu_{\boldsymbol{X}'}}[h(\boldsymbol{Z}) = y^*]. \tag{1}$$

For $\mathbb{H} = \left\{ h : \mathbb{R}^{N \times D} \to \mathbb{Y} \mid \Pr_{\boldsymbol{Z} \sim \mu_{\boldsymbol{X}}}[h(\boldsymbol{Z}) = y^*] \geq p_{\boldsymbol{X},y^*} \right\}$, the classifiers that are at least as likely as $g$ to classify $\boldsymbol{X}$ as $y^*$, the exact solution is given by the Neyman-Pearson lemma [107] (see Appendix F.2). The optimal value is $\Phi\left(\Phi^{-1}\left(p_{\boldsymbol{X},y^*}\right) - \frac{||\Delta||_2}{\sigma}\right)$, where $\Phi$ is the standard-normal CDF and $|| \cdot ||_2$ is the Frobenius norm. If $||\boldsymbol{\Delta}||_2 < \sigma \Phi^{-1}\left(p_{\boldsymbol{X},y^*}\right)$, then $p_{\boldsymbol{X}',y^*} > 0.5$ and the prediction is provably robust. Because Eq. (1) was solved exactly, this is a *tight certificate*, i.e. the best possible certificate that can be obtained by only using black-box knowledge about prediction probability $p_{\boldsymbol{X},y^*}$.

**Probabilistic certificates.** For neural networks, the prediction probability $p_{\boldsymbol{X},y^*}$ can usually not be computed analytically. Instead, one has to use Monte Carlo sampling to compute a lower confidence bound $\underline{p_{\boldsymbol{x}',y^*}}$ that holds with high probability $1 - \alpha$. The resulting certificate is a probabilistic one.

### 3.2 Group invariance

Let $f : \mathbb{R}^{N \times D} \to \mathbb{Y}$ be a classifier and $\mathbb{T}$ a group, i.e. a set and associated operator $\cdot : \mathbb{T} \times \mathbb{T} \to \mathbb{T}$ that is closed and associative under the operation, has an inverse $t^{-1}$ for each element $t \in \mathbb{T}$ and features an identity element $e$. Further let $\mathbb{T}$ act on the input space via a *group action* $\circ : \mathbb{T} \times \mathbb{R}^{N \times D} \to \mathbb{R}^{N \times D}$ that preserves the group structure, i.e. $(t \cdot t') \circ \boldsymbol{X} = t \circ (t' \circ \boldsymbol{X})$. Group actions naturally partition the domain into *orbits*, i.e. sets that can be reached by applying group actions to inputs:

**Definition 1** (Orbits). *The orbit of an input $\boldsymbol{X} \in \mathbb{R}^{N \times D}$ w.r.t. a group $\mathbb{T}$ is $[\boldsymbol{X}]_{\mathbb{T}} = \{t \circ \boldsymbol{X} \mid t \in \mathbb{T}\}$.*

Classifier $f$ is said to be invariant under group $\mathbb{T}$ if $\forall \boldsymbol{X} \in \mathbb{R}^{N \times D}, \forall \boldsymbol{X}' \in [\boldsymbol{X}]_{\mathbb{T}} : f(\boldsymbol{X}) = f(\boldsymbol{X}')$.

### 3.3 Haar measures

Haar measures [108] are a generalization of Lebesgue measures for integration over groups. Their key property is invariance, meaning the group operator does not affect the measure.

**Definition 2** (Right Haar measure). *Let $\eta$ be a finite, regular measure on the Borel subsets of $\mathbb{T}$. If $\eta(\mathbb{S}) = \eta(\{s \cdot t \mid s \in \mathbb{S}\})$ for all $t \in \mathbb{T}$ and Borel subsets $\mathbb{S} \subseteq \mathbb{T}$, then $\eta$ is a right Haar measure.*

For the groups we consider, the Haar measure is unique up to a multiplicative constant [109, 110]. In Section 6.1 we use Haar measures as a notational tool to summarize certificates for different groups. However, an understanding of measure theory is not required to follow any part of the derivations.

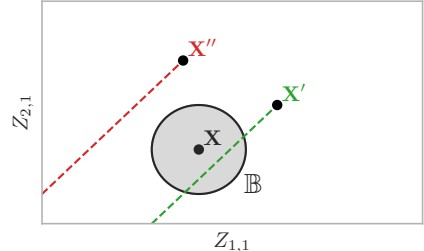

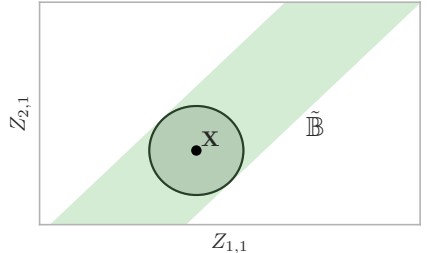

(a) Robustness for individual points          (b) Augmented certified region

Figure 1: Orbit-based gray-box certificate for translation invariance and $N = 2, D = 1$. a.) Input $\boldsymbol{X}'$ can be translated into certified region $\mathbb{B}$. It is not an adversarial example. Input $\boldsymbol{X}''$ can not be translated into $\mathbb{B}$. It might be an adversarial example. b.) The prediction $y^* = f(\boldsymbol{X})$ is certifiably robust to all perturbed inputs from augmented region $\tilde{\mathbb{B}}$, the union over all translations of $\mathbb{B}$.

## 4 Problem setting

We consider a similar setup to that described in Section 3.1, i.e. we have a smoothed classifier $f : \mathbb{R}^{N \times D} \to \mathbb{Y}$ that is the result of randomly smoothing a base classifier $g$ with an isotropic matrix normal distribution $\mu_{\boldsymbol{X}}$. Given a clean prediction $y^* = f(\boldsymbol{X})$ with clean prediction probability $p_{\boldsymbol{X},y^*}$, we want to determine whether $f(\boldsymbol{X}') = y^*$ for an adversarially perturbed input $\boldsymbol{X}' = \boldsymbol{X} + \boldsymbol{\Delta}$.

Different from all prior work, we additionally assume that the *base classifier g* is invariant under a group $\mathbb{T}$. This does not necessarily mean that the *smoothed classifier f* shares the same invariances. But, we can use the isotropy of smoothing distribution $\mu_{\boldsymbol{X}}$ to prove (see Appendix D) that randomized smoothing preserves invariance under Euclidean isotropies (rotation, reflection and translation) and permutation, which we shall leverage in Section 5.

**Theorem 1.** *Let base classifier* $g : \mathbb{R}^{N \times D} \to \mathbb{Y}$ *be invariant under group* $\mathbb{T}$ *with* $\mathbb{T} \subseteq E(D)$ *or* $\mathbb{T} \subseteq S(N)$, *where* $E(D)$ *is the Euclidean group and* $S(N)$ *is the permutation group. Then the isotropically smoothed classifier* $f$, *as defined in Section 3.1, is also invariant under* $\mathbb{T}$.

Note that the above result also holds for subgroups of $E(D)$, such as the translation group $T(D)$, the rotation group $SO(D)$ and the roto-translation group $SE(D)$. We define these groups and their actions on $\mathbb{R}^{N \times D}$ more formally in Appendix C.

## 5 Orbit-based gray-box certificates

Since we are the first to consider robustness certification for invariant models, we begin by defining a baseline that other certificates can be benchmarked against: It is based on the insight that certificates correspond to sets of *inputs* $\mathbb{B} \subseteq \mathbb{R}^{N \times D}$ that preserve clean prediction $y^*$, and group $\mathbb{T}$ corresponds to sets of *transformations* that preserve predictions. By transitivity, a perturbed input $\boldsymbol{X}'$ that can reach $\mathbb{B}$ via a transformation from $\mathbb{T}$, i.e. $\exists t \in \mathbb{T} : t \circ \boldsymbol{X}' \in \mathbb{B}$, must fulfill $f(\boldsymbol{X}') = y^*$ and cannot be an adversarial example. Combining all such inputs, i.e. combining the orbits of elements of $\mathbb{B}$ (see Definition 1), yields an augmented certified region $\tilde{\mathbb{B}} \supseteq \mathbb{B}$ (proof in Appendix E.1):

**Theorem 2** (Orbit-based certificates). *Let* $f \in \mathbb{R}^{N \times D} \to \mathbb{Y}$ *be invariant under a group* $\mathbb{T}$. *Let* $y^* = f(\boldsymbol{X})$ *be a prediction that is certifiably robust to a set of perturbed inputs* $\mathbb{B} \subseteq \mathbb{R}^{N \times D}$, *i.e.* $\forall \boldsymbol{Z} \in \mathbb{B} : f(\boldsymbol{Z}) = y^*$. *Let* $\tilde{\mathbb{B}} = \cup_{\boldsymbol{Z} \in \mathbb{B}} [\boldsymbol{Z}]_{\mathbb{T}}$. *Then* $\forall \boldsymbol{X}' \in \tilde{\mathbb{B}} : f(\boldsymbol{X}') = y^*$.

This orbit-based approach is illustrated in Fig. 1. While we focus on randomized smoothing, Theorem 2 holds for arbitrary models and certified regions $\mathbb{B}$. However, obtaining $\mathbb{B}$ via other means may not be possible. For instance, there exist no white-box certificates for rotation-invariant models.

Because randomized smoothing preserves invariance to Euclidean isometries and permutation (see Theorem 1), we may apply the orbit-based approach with $\mathbb{T} \subseteq E(D)$ or $\mathbb{T} \subseteq S(N)$ to the certified region $\mathbb{B} = \{\boldsymbol{X} + \boldsymbol{\Delta} \mid ||\boldsymbol{\Delta}||_2 < r\}$ with $r = \sigma \Phi^{-1}(p_{\boldsymbol{X},y^*})$ that we derived in Section 3.1.

While $\tilde{\mathbb{B}} = \cup_{\boldsymbol{Z} \in \mathbb{B}} [\boldsymbol{Z}]_{\mathbb{T}}$, is a valid certificate, one may desire an equivalent, but more explicit characterization of the certified region $\tilde{\mathbb{B}}$ to determine whether a specific perturbed input preserves the prediction $y^*$. We discuss such explicit characterizations in Appendix E. In particular, translation invariance (i.e. $\mathbb{T} = T(D)$) leads to certified region $\tilde{\mathbb{B}} = \{\boldsymbol{X} + \boldsymbol{\Delta} \mid ||\boldsymbol{\Delta} - \mathbf{1}_N \overline{\boldsymbol{\Delta}}||_2 < r\}$, where

$\overline{\boldsymbol{\Delta}} \in \mathbb{R}^{1 \times D}$ are column-wise averages and $r$ is defined as above. Rotation invariance (i.e. $\mathbb{T} = SO(D)$) leads to certified region $\tilde{\mathbb{B}} = \{\boldsymbol{X}' \mid ||\boldsymbol{X}'\boldsymbol{R}^T - \boldsymbol{X}||_2 < r\}$, where $\boldsymbol{R}$ is an optimal rotation matrix defined by the singular value decomposition of $\boldsymbol{X}^T\boldsymbol{X}'$ [111, 112] (see Appendix E.2.2).

# 6 Tight gray-box certificates

Now that we have established a baseline for gray-box certification, one may naturally wonder about its optimality. We answer this by deriving tight gray-box certificates, i.e. the best certificates that can be obtained for prediction $y^*$ using only the invariances of base classifier $g$ under group $\mathbb{T}$ and its prediction probability $p_{\boldsymbol{X},y^*}$ under clean smoothing distribution $\mu_{\boldsymbol{X}}$. Similar to Section 3.1, we do so by finding a worst-case classifier. In addition to constraining its clean prediction probability, we constrain the classifier to be invariant under $\mathbb{T}$, i.e. solve $\min_{h \in \mathbb{H}_{\mathbb{T}}} \Pr_{\boldsymbol{Z} \sim \mu_{\boldsymbol{X}'}}[h(\boldsymbol{Z}) = y^*]$ with

$$\mathbb{H}_{\mathbb{T}} = \left\{ h : \mathbb{R}^{N \times D} \to \mathbb{Y} \mid \Pr_{\boldsymbol{Z} \sim \mu_{\boldsymbol{X}}}[h(\boldsymbol{Z}) = y^*] \geq p_{\boldsymbol{X},y^*}, \forall \boldsymbol{Z}, \forall \boldsymbol{Z}' \in [\boldsymbol{Z}]_{\mathbb{T}} : h(\boldsymbol{Z}) = h(\boldsymbol{Z}') \right\}, \quad (2)$$

where $[\boldsymbol{Z}]_{\mathbb{T}}$ is the orbit of $\boldsymbol{Z}$ w.r.t. $\mathbb{T}$ (see Definition 1). In the following, we show how to solve the above optimization problem and then apply our certification methodology to specific invariances.

## 6.1 Certification methodology

To work with invariance constraints, it is convenient to not think of $h$ as a function, but a family of variables $(h_{\boldsymbol{Z}}) \in \mathbb{Y}$ indexed by $\mathbb{R}^{N \times D}$. The invariance constraint states that all variables from an orbit should have the same value, i.e. $\forall \boldsymbol{Z}, \forall \boldsymbol{Z}' \in [\boldsymbol{Z}]_{\mathbb{T}} : h_{\boldsymbol{Z}} = h_{\boldsymbol{Z}'}$. Intuitively, the constraint can be enforced by replacing all these variables with a single variable. We propose to formalize this idea by using canonical maps, which map all inputs from an orbit to a single, distinct representative:

**Definition 3** (Canonical map). *A canonical map for invariance under a group of transformations $\mathbb{T}$ is a function $\gamma : \mathbb{R}^{N \times D} \to \mathbb{R}^{N \times D}$ with*

$$\forall \boldsymbol{Z} \in \mathbb{R}^{N \times D} : \gamma(\boldsymbol{Z}) \in [\boldsymbol{Z}]_{\mathbb{T}}, \quad (3)$$

$$\forall \boldsymbol{Z} \in \mathbb{R}^{N \times D}, \forall \boldsymbol{Z}' \in [\boldsymbol{Z}]_{\mathbb{T}} : \gamma(\boldsymbol{Z}) = \gamma(\boldsymbol{Z}'). \quad (4)$$

In Appendix F.1, we prove that canonical maps let us discard the invariance constraints:

**Lemma 1.** *Let $g : \mathbb{R}^{N \times D} \to \mathbb{Y}$ be invariant under group $\mathbb{T}$ and let $\mathbb{H}_{\mathbb{T}}$ be defined as in Eq. (2). If $\gamma : \mathbb{R}^{N \times D} \to \mathbb{R}^{N \times D}$ is a canonical map for invariance under $\mathbb{T}$, then*

$$\min_{h \in \mathbb{H}_{\mathbb{T}}} \Pr_{\boldsymbol{Z} \sim \mu_{\boldsymbol{X}'}}[h(\boldsymbol{Z}) = y^*] = \min_{h : \mathbb{R}^{N \times D} \to \mathbb{Y}} \Pr_{\boldsymbol{Z} \sim \mu_{\boldsymbol{X}'}}[h(\gamma(\boldsymbol{Z})) = y^*] \ s.t. \Pr_{\boldsymbol{Z} \sim \mu_{\boldsymbol{X}}}[h(\gamma(\boldsymbol{Z})) = y^*] \geq p_{\boldsymbol{X},y^*}.$$

Like in Section 3.1, the optimization problem without invariance constraints from Lemma 1 could now be solved exactly using the Neyman-Pearson lemma – if we could find the distribution of $\gamma(\boldsymbol{Z})$, i.e. the distribution of representatives[1]. This can be achieved for groups $T(D)$, $SO(D)$ and $SE(D)$ via careful change of variables into an alternative parameterization of $\mathbb{R}^{N \times D}$ (see Appendix F.3).

This leads us to our main result, which we derive more formally in Appendix F.3. In the following, let $\langle \boldsymbol{A}, \boldsymbol{B} \rangle_{\mathrm{F}}$ be the Frobenius inner product $\sum_{n=1}^{N}(\boldsymbol{A}_n)^T \boldsymbol{B}_n$ and recall from Section 3.3 that a right Haar measure is a measure for integration over a group $\mathbb{T}$.

**Theorem 3.** *Let $g : \mathbb{R}^{N \times D} \to \mathbb{Y}$ be invariant under $\mathbb{T}$ with $\mathbb{T}$ chosen from $\{T(D), SO(D), SE(D)\}$. For $SO(D)$ and $SE(D)$, let $D \in \{2, 3\}$. Let $\mathbb{H}_{\mathbb{T}}$ be defined as in Eq. (2) and $\eta$ be a right Haar measure on $\mathbb{T}$. Define the indicator function $h^* : \mathbb{R}^{N \times D} \to \{0, 1\}$ with*

$$h^*(\boldsymbol{Z}) = \mathbb{1}\left[\frac{\beta_{\boldsymbol{X}'}(\boldsymbol{Z})}{\beta_{\boldsymbol{X}}(\boldsymbol{Z})} \leq \kappa\right], \ \text{where} \ \beta_{\boldsymbol{X}}(\boldsymbol{Z}) = \int_{t \in \mathbb{T}} \exp\left(\langle t \circ \boldsymbol{Z}, \boldsymbol{X} \rangle_{\mathrm{F}} / \sigma^2\right) d\eta(t) \quad (5)$$

$$\text{and} \ \kappa \in \mathbb{R} \ \text{such that} \ \mathop{\mathbb{E}}_{\boldsymbol{Z} \sim \mu_{\boldsymbol{X}}}[h^*(\boldsymbol{Z})] = p_{\boldsymbol{X},y^*}. \quad (6)$$

*Then*

$$\min_{h \in \mathbb{H}_{\mathbb{T}}} \Pr_{\boldsymbol{Z} \sim \mu_{\boldsymbol{X}'}}[h(\boldsymbol{Z}) = y^*] = \mathop{\mathbb{E}}_{\boldsymbol{Z} \sim \mu_{\boldsymbol{X}'}}[h^*(\boldsymbol{Z})]. \quad (7)$$

---

[1]which, by Definition 3, is equivalent to a distribution over orbits.

The indicator function $h^*$ in Eq. (5) corresponds to the worst-case invariant classifier for clean prediction probability $p_{\boldsymbol{X},y^*}$. To classify an input sample $\boldsymbol{Z}$, it integrates Gaussian kernels of $(t \circ \boldsymbol{Z}, \boldsymbol{X}')$ and $(t \circ \boldsymbol{Z}, \boldsymbol{X})$ over group $\mathbb{T}$ (e.g. over all possible rotations). If the ratio of these integrals is below a threshold $\kappa$, it classifies $\boldsymbol{Z}$ as $y^*$. The constraint in Eq. (6) ensures that the probability of predicting $y^*$ under the clean smoothing distribution $\mu_{\boldsymbol{X}}$ matches that of the actual base classifier $g$. The expected value with respect to perturbed smoothing distribution $\mu_{\boldsymbol{X}'}$ in Eq. (7) is the optimal value of our optimization problem. As discussed in Section 3.1, the prediction is certifiably robust if this optimal value is greater than $1/2$.

**Applying the certificate** to a group $\mathbb{T}$ requires three steps: 1.) Calculating the Haar integrals in Eq. (5). 2.) Solving Eq. (6) for classification threshold $\kappa$. 3.) Evaluating the expected value in Eq. (7).

**Connection to prior work.** This result differs from black-box randomized smoothing with Gaussian noise, where the worst-case classifier is a linear model [26]. The *group averaging* performed by the worst-case invariant classifier, i.e. integrating a function over group $\mathbb{T}$, is a key technique for building invariant models [7, 113–117]. Group-averaged kernels have been proposed in [15]. It is fascinating to see them naturally materialize from nothing but an invariance constraint.

## 6.2 Translation invariance

In the case of translation invariance (i.e. $\mathbb{T} = T(D)$), we can evaluate the worst-case classifier (Eq. (5), solve for threshold $\kappa$ (Eq. (6)) and evaluate the perturbed prediction probability (Eq. (7)) analytically. This leads to the following result (proof in Appendix F.4.1):

**Theorem 4.** *Let $g : \mathbb{R}^{N \times D} \to \mathbb{Y}$ be invariant under $\mathbb{T} = T(D)$ and $\mathbb{H}_{\mathbb{T}}$ be defined as in Eq. (2). Then*

$$\min_{h \in \mathbb{H}_{\mathbb{T}}} \Pr_{\boldsymbol{Z} \sim \mu_{\boldsymbol{X}'}} [h(\boldsymbol{Z}) = y^*] = \Phi\left( \Phi^{-1}(p_{\boldsymbol{X},y^*}) - \frac{1}{\sigma} \left|\left| \boldsymbol{\Delta} - \mathbf{1}_N \overline{\boldsymbol{\Delta}} \right|\right|_2 \right),$$

*where $\overline{\boldsymbol{\Delta}} \in \mathbb{R}^{1 \times D}$ are the column-wise averages of $\Delta = \boldsymbol{X}' - \boldsymbol{X}$ and $\sigma$ is the standard deviation of the isotropic matrix normal smoothing distribution $\mu_{\boldsymbol{X}}$.*

**Certificate parameters.** For a fixed smoothing standard deviation $\sigma$, the certificate depends on a single parameter: The norm $\left|\left| \boldsymbol{\Delta} - \mathbf{1}_N \overline{\boldsymbol{\Delta}} \right|\right|_2$ of the mean-centered perturbation matrix $\boldsymbol{\Delta}$.

**Comparison to the orbit-based certificate.** Substituting into robustness condition $\min_{h \in \mathbb{H}_{\mathbb{T}}} \Pr_{\boldsymbol{Z} \sim \mu_{\boldsymbol{X}'}} [h(\boldsymbol{Z}) = y^*] > \frac{1}{2}$ shows that $f(\boldsymbol{X}') = y^*$ if $\left|\left| \boldsymbol{\Delta} - \mathbf{1}_N \overline{\boldsymbol{\Delta}} \right|\right|_2 < \sigma \Phi^{-1}(p_{\boldsymbol{X},y^*})$. This result, obtained via our tight certification methodology, is identical to the orbit-based certificate for translation invariance from the end of Section 5. In other words: Despite its simplicity, the orbit-based certificate is the best possible gray-box certificate for translation invariance.

## 6.3 Rotation invariance in 2D

Considering the previous result, one may suspect the orbit-based certificate to be tight for arbitrary invariances. This is not the case. For instance, it is not tight for rotation invariance ($\mathbb{T} = SO(D)$):

**Theorem 5.** *Let $g : \mathbb{R}^{N \times D} \to \mathbb{Y}$ be invariant under $\mathbb{T} = SO(D)$ and $\mathbb{H}_{\mathbb{T}}$ be defined as in Eq. (2). Assume that perturbed input $\boldsymbol{X}'$ is not obtained via rotation of $\boldsymbol{X}$, i.e. $\nexists \boldsymbol{R} \in SO(D) : \boldsymbol{X}' = \boldsymbol{X}\boldsymbol{R}^T$. Further assume that $p_{\boldsymbol{X},y^*} \in (0,1)$. Then, for all $\boldsymbol{R} \in SO(D)$:*

$$\min_{h \in \mathbb{H}_{\mathbb{T}}} \Pr_{\boldsymbol{Z} \sim \mu_{\boldsymbol{X}'}} [h(\boldsymbol{Z}) = y^*] > \Phi\left( \Phi^{-1}(p_{\boldsymbol{X},y^*}) - \frac{1}{\sigma} \left|\left| \boldsymbol{X}'\boldsymbol{R}^T - \boldsymbol{X} \right|\right|_2 \right). \tag{8}$$

In other words: The tight certificate is strictly stronger. Proof in Appendix G.

Next, we apply Theorem 3 to obtain the strictly stronger, tight certificate for rotation invariance in 2D. In the following, let $\boldsymbol{R}(\theta) \in SO(2)$ be the matrix that rotates counter-clockwise by angle $\theta$ and $\mathcal{I}_0$ be the modified Bessel function of the first kind and order $0$. The Haar integral from Eq. (5) that defines the worst-case classifier can be evaluated analytically (see Appendix F.4.2)):

$$\beta_{\boldsymbol{X}}(\boldsymbol{Z}) = \mathcal{I}_0\left( \frac{1}{\sigma^2} \sqrt{\langle \boldsymbol{Z}, \boldsymbol{X} \rangle_{\mathrm{F}}^2 + \langle \boldsymbol{Z}, \boldsymbol{X}\boldsymbol{R}(-\pi/2)^T \rangle_{\mathrm{F}}^2} \right), \tag{9}$$

We can substitute this into Theorem 3 and use the fact that Eq. (9) depends on linear transformations of the matrix normal random variable $\boldsymbol{Z}$ to obtain the following certificate (proof in Appendix F.4.2):

**Theorem 6.** *Let $g : \mathbb{R}^{N \times 2} \to \mathbb{Y}$ be invariant under $\mathbb{T} = SO(2)$ and $\mathbb{H}_{\mathbb{T}}$ be defined as in Eq. (2). Define the indicator function $\rho : \mathbb{R}^4 \to \{0, 1\}$ with*

$$\rho(\boldsymbol{q}) = \mathbb{1}\left[\mathcal{I}_0\left(\sqrt{q_1^2 + q_2^2}\right) \Big/ \mathcal{I}_0\left(\sqrt{q_3^2 + q_4^2}\right) \leq \kappa\right], \tag{10}$$

$$\text{with } \kappa \in \mathbb{R} \text{ such that } \underset{\boldsymbol{q} \sim \mathcal{N}\left(\boldsymbol{m}^{(2)}, \boldsymbol{\Sigma}\right)}{\mathbf{E}}[\rho(\boldsymbol{q})] = p_{\boldsymbol{X}, y^*}. \tag{11}$$

*Then*

$$\min_{h \in \mathbb{H}_{\mathbb{T}}} \Pr_{\boldsymbol{Z} \sim \mu_{\boldsymbol{X}'}}[h(\boldsymbol{Z}) = y^*] = \underset{\boldsymbol{q} \sim \mathcal{N}\left(\boldsymbol{m}^{(1)}, \boldsymbol{\Sigma}\right)}{\mathbf{E}}[\rho(\boldsymbol{q})], \tag{12}$$

*where $\boldsymbol{m}^{(1)}, \boldsymbol{m}^{(2)} \in \mathbb{R}^4$, $\boldsymbol{\Sigma} \in \mathbb{R}^{4 \times 4}$ are linear combinations (see Appendix F.4.2) of $||\boldsymbol{X}||_2^2 / \sigma^2$, $||\boldsymbol{\Delta}||_2^2 / \sigma^2$ and parameters $\epsilon_1 = \langle \boldsymbol{X}, \boldsymbol{\Delta} \rangle_{\mathrm{F}}$, $\epsilon_2 = \langle \boldsymbol{X}\boldsymbol{R}(-\pi/2)^T, \boldsymbol{\Delta} \rangle_{\mathrm{F}}$.*

**Certificate parameters.** This certificate depends on the perturbation norm $||\boldsymbol{\Delta}||_2$ and clean data norm $||\boldsymbol{X}||_2$, relative to the smoothing standard deviation $\sigma$. It further depends on parameters $\epsilon_1$ and $\epsilon_2$, which are Frobenius inner products between $\boldsymbol{\Delta}$ and the clean input $\boldsymbol{X}$ before and after a rotation by $-\pi/2$. These parameters capture the orientation of $\boldsymbol{X}' = \boldsymbol{X} + \boldsymbol{\Delta}$ relative to $\boldsymbol{X}$, as one would expect from a rotation-invariance aware certificate.

**Monte Carlo evaluation.** Evidently, we do not have a closed-form expression for the expectations in Eqs. (11) and (12). However, recall from Section 3.1 that randomized smoothing already involves an intractable expectation, namely the prediction probability $p_{\boldsymbol{X}, y^*}$. One has to use Monte Carlo sampling to compute a lower confidence bound that hold with high probability $1 - \alpha$. We adopt the same approach: First, we lower-bound threshold $\kappa$ from Eq. (11), i.e. make the classifier slightly less likely to predict $y^*$. Then, we lower-bound bound expectation $\mathbf{E}_{\boldsymbol{q} \sim \mathcal{N}\left(\boldsymbol{m}^{(1)}, \boldsymbol{\Sigma}\right)}[\rho(\boldsymbol{q})]$ from Eq. (12). Because the resulting value is slightly smaller than the optimal value of our optimization problem $\min_{h \in \mathbb{H}_{\mathbb{T}}} \Pr_{\boldsymbol{Z} \sim \mu_{\boldsymbol{X}'}}[h(\boldsymbol{Z}) = y^*]$, this procedure yields a valid certificate. We discuss the full algorithm and how to ensure that all bounds simultaneously hold in Appendix F.5. Because the expectations only require sampling from four-dimensional normal distributions and do not depend on base classifier $g$, one can use a large number of samples to obtain narrow bounds at little computational cost (e.g. 0.59s for 100000 samples per confidence bound on an Intel Xeon E5-2630 CPU).

### 6.4 Rotation invariance in 3D

To evaluate the tight certificate for 3D rotation invariance (i.e. $\mathbb{T} = SO(3)$), we adopt the same Monte Carlo evaluation approach discussed in Section 6.3. Different from 2D rotation invariance, evaluating the worst-case classifier analytically is not tractable. It can however be evaluated by numerical integration over an Euler angle parameterization of rotation group $SO(3)$ (see Appendix F.4.3).

### 6.5 Roto-translation invariance in 2D and 3D

In Appendix F.4.4 we prove that additionally enforcing translation invariance (i.e. $\mathbb{T} = SE(D)$) is equivalent to centering $\boldsymbol{X}$ and $\boldsymbol{\Delta}$ before evaluating the certificates for rotation invariance. This is consistent with our result from Section 6.2, i.e. the orbit-based approach being optimal for translation.

## 7 Limitations and broader impact

The main limitation of our work lies in its exploratory nature. We have derived the first certificates for models invariant under arbitrary Euclidean isometries and/or permutation. While these invariances are of key importance to many practical applications , there is a vast swath of invariances that we have not covered, such as spatio-temporal invariances [44], invariance under graph isomorphisms [9] or invariances for planar images [7]. Furthermore, we have not yet derived tight certificates for reflections and permutations (though our certification methodology could conceivably be applied to them).

**Broader impact.** With the growing prevalence of machine learning in safety-critical and sensitive domains like autonomous driving [118, 119] or healthcare [120, 121], trustworthy models promise to become increasingly important. Certification is one pillar of trustworthiness, alongside concepts like fairness [122, 123] or differential privacy [124, 125]. Unique to our work is that we certify models

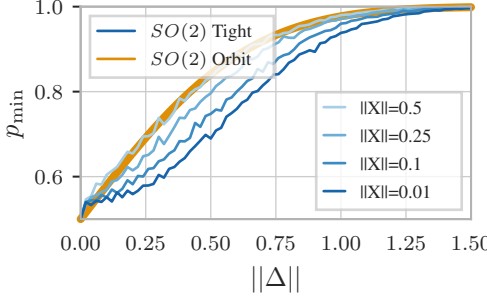

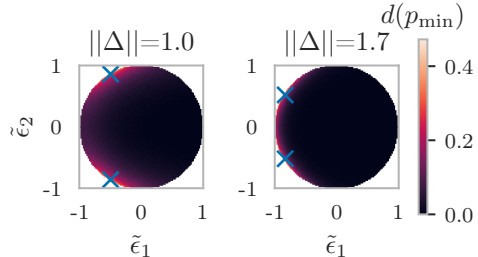

Figure 2: Comparison of tight and orbit-based certificates applied to adversarial scaling for $\sigma = 0.5$ and varying $||\mathbf{\Delta}||$ and $||\mathbf{X}||$ (smaller $p_{\min}$ is better). As $||\mathbf{X}||$ increases, the difference between the certificates shrinks.

Figure 3: Difference in $p_{\min}$ between the tight certificate for 2D rotation invariance and the black-box baseline for $\sigma = 0.5$, $||\mathbf{X}|| = 1.0$. Crosses correspond to adversarial rotations. Large $d(p_{\min})$ can be observed near adversarial rotations.

with spatial invariances. These invariances naturally occur in the physical sciences, including those with a direct societal impact like pharmacology and biochemistry. Our work can be seen as a first step towards provable trustworthiness for tasks like drug discovery [39–41] or protein folding [45–49].

# 8 Experimental evaluation

We already know that the orbit-based certificate for translation invariance is tight and certifies robustness for an infinitely larger volume than black-box randomized smoothing (see Fig. 1). Therefore, we focus our experiments on the certificates for rotation invariance. Recall that the orbit-based certificate guarantees robustness for the set $\tilde{\mathbb{B}} = \left\{ \mathbf{X}' \mid ||\mathbf{X}'\mathbf{R}^T - \mathbf{X}||_2 < r \right\}$ with $r = \sigma \Phi^{-1}\left(p_{\mathbf{X}, y^*}\right)$ and optimal rotation matrix $\mathbf{R}$. In other words: It certifies robustness for perturbations with rotational components that can be eliminated to bring $\mathbf{X}'$ into distance $r$ of $\mathbf{X}$. We want to understand whether the tight certificates offer any benefit beyond that, or if the strict inequality in Theorem 5 is due to some negligible $\epsilon$. To this end, we first thoroughly examine the four-dimensional parameter space of the tight certificate for rotation invariance in 2D, before applying our certificates to rotation invariant point cloud classifiers.

All parameters and experimental details are specified in Appendix B. We use 10000 samples per confidence bound and set $\alpha = 0.001$, i.e. all certificates hold with 99.9% probability. A reference implementation will be made available at https://www.cs.cit.tum.de/daml/invariance-smoothing.

## 8.1 Tight certificate parameter space

The tight certificate for 2D rotation invariance depends on $||\mathbf{X}||_2 / \sigma$, $||\mathbf{\Delta}||_2 / \sigma$ and parameters $\epsilon_1$ and $\epsilon_2$, which capture the orientation of the perturbed point cloud and fulfill $\sqrt{\epsilon_1^2 + \epsilon_2^2} \leq ||\mathbf{X}||_2 \cdot ||\mathbf{\Delta}||_2$ (see Appendix J). To avoid clutter, we define $\tilde{\epsilon}_k := \epsilon_k / (||\mathbf{X}||_2 \cdot ||\mathbf{\Delta}||_2)$. As our metric for this section, we report $p_{\min}$, the smallest probability $p_{\mathbf{X}, y^*}$ for which a prediction can still be certified [2]

**Adversarial scaling.** First, we assume that $\mathbf{X}' = (1 + c)\mathbf{X}$, i.e. the input is adversarially scaled. In this case, we have $\tilde{\epsilon}_1 = 1$ and $\tilde{\epsilon}_2 = 0$. We then vary $||\mathbf{\Delta}||_2$ and $||\mathbf{X}||_2$ and evaluate our certificates. Note that such attacks have no rotational component, i.e. the orbit-based certificate is identical to the black-box one. Fig. 2 shows that, even in the absence of rotations, the tight certificate can yield significantly stronger guarantees. For $\sigma = 0.5$ and $||\mathbf{X}||_2 = 0.01$, the baseline can only certify robustness for a prediction with $p_{\mathbf{X}, y^*} = 0.8$ if $||\mathbf{\Delta}||_2 \leq 0.4$. The tight certificate can certify robustness up to $||\mathbf{\Delta}||_2 = 0.73$. However, the gap shrinks as the norm of the clean data increases.

**Effect of data norm.** We would like to see if this is a pervasive pattern. To this end, we fix $||\mathbf{\Delta}||_2$, gradually increase $||\mathbf{X}||_2$, and evaluate the tight certificate for $\tilde{\epsilon}_1, \tilde{\epsilon}_2$ on a $100 \times 100$ rasterization of $[0, 1] \times [0, 1]$. We then measure the difference $d(p_{\min})$ to the black-box certificate (we will discuss

---

[2] We discuss how to compute these inverse certificates in Appendix I.

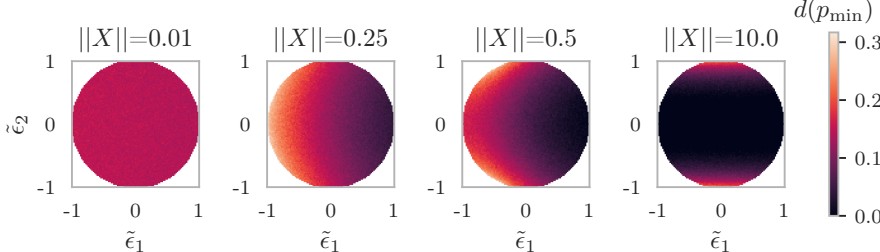

Figure 4: Difference in $p_{\min}$ between the tight certificate for 2D rotation invariance and the black-box baseline for $\sigma = 0.5$, $||\Delta|| = 0.5$ under varying $||X||$, $\tilde{\epsilon}_1$ and $\tilde{\epsilon}_2$. As $||X||$ increases, the regions where the tight certificate outperforms the black-box baseline shrink.

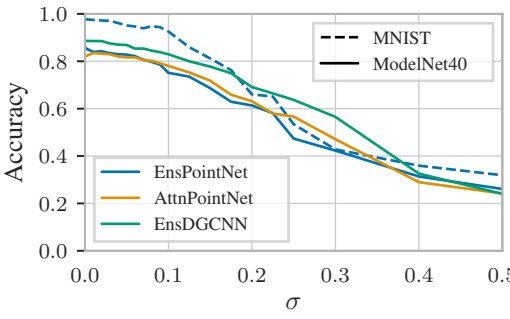

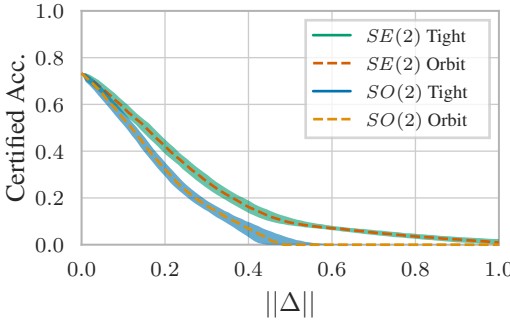

Figure 5: Test set accuracy of smoothed translation and rotation invariant point cloud classifiers on the ModelNet40 and MNIST pointcloud datasets, under varying standard deviation $\sigma$.

Figure 6: Comparison of certificates for adversarial scaling of MNIST with EnsPointNet and $\sigma = 0.15$. The tight and orbit-based certificates yield similar certified accuracies.

how these results relate to the orbit-based one shortly). Fig. 4 shows that for small $||X||_2$, the tight certificate outperforms the black-box certificate for arbitrary perturbations of norm $||\Delta||_2$. But, as $||X||_2$ increases, the regions where it outperforms the black-box one shrink.

**Rotational components.** Simple algebra (see Appendix J) shows that for any combination of $||X||_2$ and $||\Delta||_2$, adversarial rotations, i.e. $X' = XR^T$ such that $||X' - X|| = ||\Delta||$, correspond to two specific points with $\sqrt{\tilde{\epsilon}_1^2 + \tilde{\epsilon}_2^2} = 1$ in the certificate's parameter space. In Fig. 3 we fix $\sigma = 0.5$ and $||X||_2 = 1$, vary $||\Delta||$ and highlight these two points. We observe that the tight certificate only outperforms the black-box certificate for values of $\tilde{\epsilon}_1, \tilde{\epsilon}_2$ that are close to adversarial rotations. However, robustness to such attacks can also be readily certified by the orbit-based approach. Combined with our previous observations, and because the certificate only depends on norms relative to smoothing standard deviation $\sigma$, i.e. $||X||_2/\sigma$ and $||\Delta||_2/\sigma$, we expect the tight and the orbit-based certificate to perform similarly well, assuming that the smoothing standard deviation $\sigma$ is small relative to $||X||_2$.

## 8.2 Application to point cloud classification

To verify whether our observations hold in practice, we apply our certificates to point cloud classification. We consider two datasets: 3D point cloud representations of ModelNet40 [126], which consists of CAD models from 40 different categories, and 2D point cloud representations of MNIST [127]. We apply the same pre-processing steps as in [6]. Certification is performed on the default test sets. As our base classifiers, we use rotation and translation (i.e. $SE(D)$) invariant versions of two well-established models that are used in prior work on robustness certification for point clouds: PointNet [6] and DGCNN [128]. To implement the invariances, we center the input data, perform principal component analysis, apply the model to all possible poses (see discussion in [129, 130]) and average the output logits (EnsPointNet and EnsDGCNN). In addition, we consider a more refined model [129] that combines canonical poses via a self-attention mechanism (AttnPointNet). Throughout this section, we report both probabilistic upper and lower bounds for the Monte Carlo evaluation of the tight certificates (recall Section 6.3).

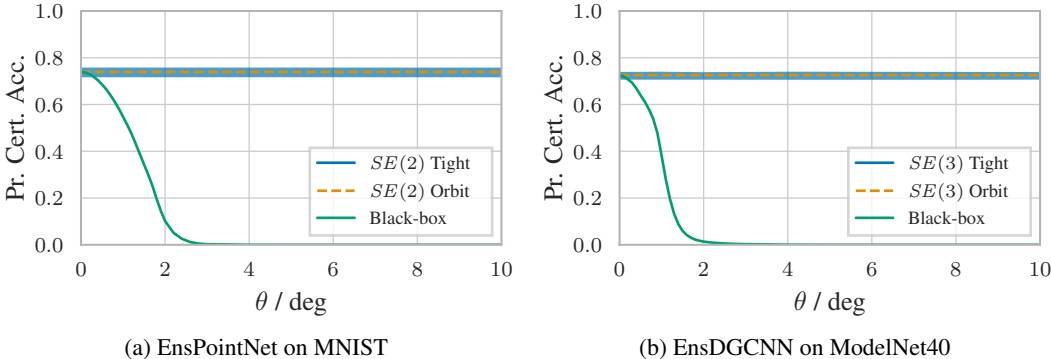

(a) EnsPointNet on MNIST           (b) EnsDGCNN on ModelNet40

Figure 7: Comparison of tight, orbit-based and black-box certificates for randomly perturbed inputs with $||\boldsymbol{\Delta}|| = \sigma = 0.1$, rotated by angle $\theta$. The gray-box certificates effectively eliminate the induced rotation, while the black-box method cannot certify robustness for large $\theta$.

**Practical smoothing parameters.** Fig. 5 shows the test set accuracies of randomly smoothed models under varying standard deviations $\sigma$. Values of $\sigma$ that preserve an accuracy above $50\%$ are small, relative to the average norm of the test sets (10.67 for MNIST, 19.17 for ModelNet40). Going by our previous results, we expect the tight and orbit-based certificates to perform similarly well.

**Adversarial scaling.** In Fig. 6 we again consider adversarial scaling, i.e. attacks without rotational components, but applied to the MNIST point cloud dataset with $\sigma = 0.15$. We report the certified accuracy, i.e. the percentage of correct and provably robust predictions, for certification with $(SE(2))$ and without $(SO(2))$ translation invariance. The tight and orbit-based certificate yield similar results. We further observe that enforcing translation invariance increases the certified accuracy and extends the range of $||\Delta||$ for which robustness can be certified.

**Rotational components.** Finally, we study perturbations with rotational components. We fix $||\boldsymbol{\Delta}||$, randomly sample perturbations of the specified norm and then rotate $\boldsymbol{Z} = \boldsymbol{X} + \boldsymbol{\Delta}$ by a specified angle $\theta \in [0, 10°]$ (in the case of ModelNet40, around one randomly chosen axis). For each element of the test set and each $\theta$, we generate 10 such samples. We then compute the percentage of samples $\boldsymbol{X}'$ for which $f(\boldsymbol{X})$ is correct and $f(\boldsymbol{X}') = f(\boldsymbol{X})$ is provably guaranteed ("probabilistic certified accuracy"). Fig. 7 shows results for MNIST and ModelNet40 evaluated with $||\boldsymbol{\Delta}|| = \sigma = 0.1$. The black-box baseline's probabilistic certified accuracy drops close to 0 for $\theta = 2°$. The gray-box certificates are almost constant in $\theta$, i.e. effectively eliminate any induced rotation. However, the tight certificate did not offer any meaningful benefit beyond that. Using the lower bound for Monte Carlo evaluation, there was not a single sample for which only the tight certificate could guarantee robustness.

In Appendix A we repeat the experiments from this and the previous section for various other combinations of parameter values. All results are consistent with the ones presented here, confirming that the orbit-based approach offers a good approximation of the tight certificates in practice.

## 9 Conclusion

For the first time, we have studied the use of invariances for robustness certification. We proposed a gray-box approach, combining white-box knowledge about invariances with black-box randomized smoothing. We have derived a orbit-based procedure for certification that can be applied to arbitrary models with invariance to permutations and Euclidean isometries. We have proven that the orbit-based certificate for translation invariance is tight and derived strictly stronger certificates for rotation invariance. Our experiments are to be interpreted in two ways: Firstly, the fact that it is possible to derive tight invariance-aware certificates and that there exist scenarios in which they offer stronger guarantees for arbitrary perturbations should be an exciting inspiration for future work. Secondly, the fact that the orbit-based certificates are easily interpretable and offer good approximations of our tight certificates should invite their application to real-world tasks with inherent invariances.

**Acknowledgements.** The authors would like to thank Johannes Gasteiger for valuable discussions on invariant deep learning and Lukas Gosch for constructive criticism of the manuscript. This research was supported by the German Research Foundation, grant GU 1409/4-1.

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
