# A  Additional experiments

In the following, we repeat the experiments from Section 8 for a wider range of certificate and model parameters. In Appendix A.1, we investigate the parameter space of the tight certificate for rotation invariance in 2D. In Appendix A.2, we again apply our certificates to rotation invariant point cloud classifiers, this time considering all three models (EnsPointNet, AttnPointNet and EnsDGCNN) and different smoothing distribution parameters. As before, we take $10000$ samples per confidence bound and set $\alpha = 0.001$, i.e. all certificates hold with $99.9\%$ probability.

All results support our main conclusion from Section 8: The tight certificates for rotation invariance can offer much stronger robustness guarantees than their orbit-based counterparts. However, in practical scenarios, where the smoothing standard deviation is small relative to the norm of the clean data, the orbit-based approach offers a very good approximation.

## A.1 Tight certificate parameter space

For this section, recall that the tight certificate for 2D rotation invariance depends on $||\boldsymbol{X}||_2$, $||\boldsymbol{\Delta}||_2$ and parameters $\epsilon_1 = \langle \boldsymbol{X}, \boldsymbol{\Delta} \rangle_{\mathrm{F}}$, $\epsilon_2 = \left\langle \boldsymbol{X} \boldsymbol{R}\left(-\pi/2\right)^T, \boldsymbol{\Delta} \right\rangle_{\mathrm{F}}$, as well as smoothing standard deviation $\sigma$. We define $\tilde{\epsilon}_1 = \frac{\epsilon_1}{||\boldsymbol{X}|| \cdot ||\boldsymbol{\Delta}||}$ and $\tilde{\epsilon}_2 = \frac{\epsilon_2}{||\boldsymbol{X}|| \cdot ||\boldsymbol{\Delta}||}$. Note that $\sqrt{\tilde{\epsilon}_1^2 + \tilde{\epsilon}_2^2} \leq 1$ (see also Appendix J).

### A.1.1 Adversarial scaling

We begin with adversarial scaling, which corresponds to $\tilde{\epsilon}_1 = 1, \tilde{\epsilon}_2 = 0$. We consider $\sigma \in \{0.1, 0.15, 0.2, 0.25, 0.5, 1.0\}$. For each $\sigma$, we vary $||\boldsymbol{X}||_2$ and $||\boldsymbol{\Delta}||$ and compute $p_{\min}$, the smallest value of $p_{\boldsymbol{X},y^*}$ for which a prediction can still be certified (see Appendix I). Lower $p_{\min}$ mean that the certificate can guarantee robustness for models that are less consistent in their predictions.

Fig. 8 shows that, if the clean data norm is small, e.g. $||\boldsymbol{X}|| = 0.01$, then the tight certificate yields much lower $p_{\min}$ than the orbit-based one, save for very small and very large values of $||\boldsymbol{\Delta}||$. That is, the tight certificate for rotation invariance is beneficial even for adversarial perturbations that do not have any rotational components. However, as $||\boldsymbol{X}||$ approaches $\sigma$, this difference shrinks, i.e. both approaches offer guarantees of similar strength.

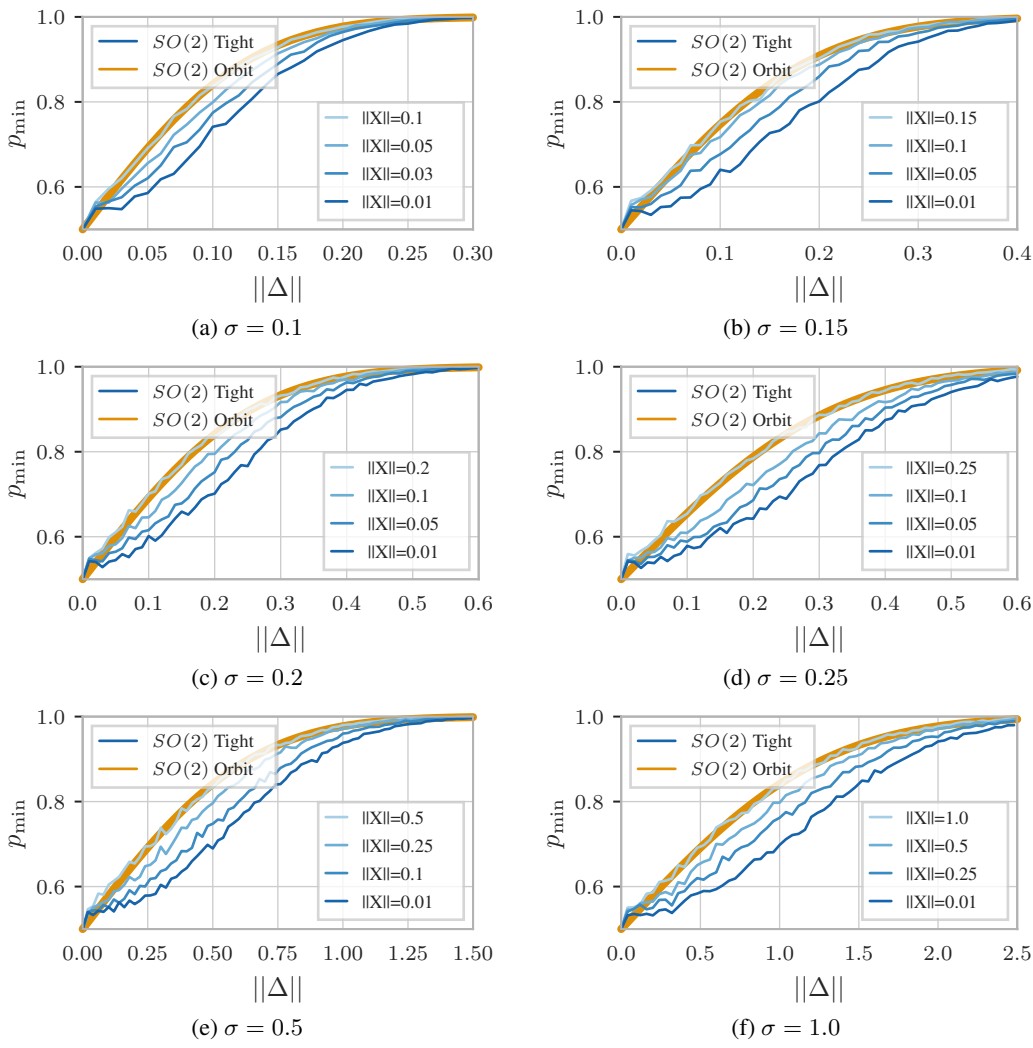

Figure 8: Comparison of tight and orbit-based certificates for rotation invariance in 2D, applied to adversarial scaling for varying $||\boldsymbol{\Delta}||$, $||\boldsymbol{X}||$ and smoothing standard deviation $\sigma$ (smaller $p_{\min}$ is better). As $||\boldsymbol{X}||$ increases, the difference between the certificates shrinks.

### A.1.2 Effect of clean data norm on certificates for arbitrary perturbations

Next, we study whether the same effect can be observed with arbitrary perturbations, i.e. arbitrary values of $\tilde{\epsilon}_1$ and $\tilde{\epsilon}_2$. In Fig. 4 from Section 8.1, we fixed a specific combination of $\sigma$ and $||\boldsymbol{\Delta}||$, varied $||\boldsymbol{X}||_2$ and evaluated the tight certificate for $\tilde{\epsilon}_1, \tilde{\epsilon}_2$ taken from a $100 \times 100$ rasterization of $[0, 1] \times [0, 1]$. We then measured the difference $d(p_{\min})$ to the black-box certificate.

Here, we consider different combinations of $\sigma$, $||\boldsymbol{\Delta}||$ and $||\boldsymbol{X}||$. We can reduce the dimensionality of the parameter space that needs to be explored by observing that both the tight gray-box and the black-box certificate do not depend on the absolute value of these parameters, but their value relative to $\sigma$. The black-box baseline has $p_{\min} = \Phi(||\boldsymbol{\Delta}||_2 / \sigma)$. As specified in Appendix F.4.2, the tight certificate depends on $||\boldsymbol{X}||_2^2 / \sigma^2$, $||\boldsymbol{\Delta}||_2^2 / \sigma^2$, $\epsilon_1 / \sigma^2 = \frac{1}{\sigma^2}||\boldsymbol{X}||_2 \cdot ||\boldsymbol{\Delta}||_2 \cdot \cos(\theta_1)$ and $\frac{1}{\sigma^2} / ||\boldsymbol{X}||_2 \cdot ||\boldsymbol{\Delta}||_2 \cdot \cos(\theta_1)$, where $\theta_1, \theta_2$ are angles between $2N$-dimensional vectors. Therefore, we can fix an arbitrary value of $\sigma$ and then choose $||\boldsymbol{X}||_2$ and $||\boldsymbol{\Delta}||_2$ relative to it.

We set set $\sigma = 1$ and consider $||\boldsymbol{\Delta}||_2 \in \{\frac{1}{4}\sigma, \frac{1}{2}\sigma, \sigma, 2\sigma, 3\sigma\}$, i.e. perturbation norms that are much smaller, similar to or much larger than the smoothing standard deviation. For each $||\boldsymbol{\Delta}||_2$, we evaluate the tight certificate $||\boldsymbol{X}||_2 \in \{\frac{1}{100}\sigma, \frac{1}{2}\sigma, \sigma, 10\sigma\}$, i.e. clean data norms that are much smaller, similar to or much larger than the smoothing standard deviation.

Fig. 9 shows the resulting $d(p_{\min})$. In the case where $||\boldsymbol{\Delta}||_2 \leq 2||\boldsymbol{X}||_2$ we have additionally highlighted the two values of $(\tilde{\epsilon}_1, \tilde{\epsilon}_2)$ that correspond to adversarial rotations with blue crosses (see Appendix J) Note that, to improve readability, the colorbar is scaled differently for each choice of $||\boldsymbol{\Delta}||$, i.e. each row. Within each row, the same colorbar is used.

We can make four observations.

Firstly, in the case that $||\boldsymbol{\Delta}||_2 \leq \sigma$ and $||\boldsymbol{X}|| = 0.01$ the tight certificate yields lower (i.e. better) $p_{\min}$ for arbitrary perturbations, not just those corresponding to adversarial rotations.

Secondly, as the clean data norm $||\boldsymbol{X}||_2$ increases, the region where the tight certificate outperforms the black-box baseline shrinks. It concentrates around values of $(\tilde{\epsilon}_1, \tilde{\epsilon}_2)$ corresponding to adversarial rotations.

Thirdly, the difference in $p_{\min}$ is not as drastic when $||\boldsymbol{\Delta}||_2$ is small. This is to be expected. For instance, with $||\boldsymbol{\Delta}||_2 = \sigma / 4$, the black-box certificate has $p_{\min} = \Phi(\sigma / 4) \approx 0.6$. Here, the difference to the smallest possible value $0.5$ is small, meaning there is little room for improvement. We already observed this in Fig. 8.

Finally, when $||\boldsymbol{\Delta}||_2$ is large relative to $\sigma$, and so large that $\boldsymbol{\Delta}$ cannot be the result of an adversarial rotation (e.g. $||\boldsymbol{\Delta}||_2 = 3$, $||\boldsymbol{X}||_2 = 0.01$), then the tight certificate is almost identical to the black-box baseline. But, as $||\boldsymbol{X}||_2$ increases and an adversarial rotation becomes possible (e.g. $||\boldsymbol{\Delta}||_2 = 3$, $||\boldsymbol{X}||_2 = 10$) then the tight certificate does again yield much better $p_{\min}$ – but only for values of $(\tilde{\epsilon}_1, \tilde{\epsilon}_2)$ that are very close to adversarial rotations.

In summary, these observations support our claim that, if $||\boldsymbol{X}||$ is large relative to $\sigma$, then the tight certificate only improves upon the black-box baseline for perturbations that are similar to adversarial rotations. Combined with the fact that the orbit-based gray-box certificate for rotation invariance also accounts for rotational components, this suggests that the orbit-based certificate should be a good approximation in practice.

In Fig. 10, we repeat the same experiments with $\sigma$, $||\boldsymbol{X}||_2$ and $||\boldsymbol{\Delta}||_2$ scaled by a factor of $\frac{1}{2}$, to verify that it is indeed sufficient to only consider a single, arbitrary value of $\sigma$ and the relative value of all other parameters. As expected, the results are identical to Fig. 9.

In Fig. 11 we perform the same experiment for fixed values of $\sigma$ and $||\boldsymbol{X}||_2$ and two values of $||\boldsymbol{\Delta}||_2$ chosen such that adversarial rotations correspond to $\tilde{\epsilon} = \approx 0.5$ and $\tilde{\epsilon}_2 \approx \pm 0.5$, respectively (similar to Fig. 3 from Section 8.1). In other words: The adversarial rotations move in $30°$-increments in the parameter space. Again, we can observe that the regions with large $p_{\min}$ move with the adversarial rotations.

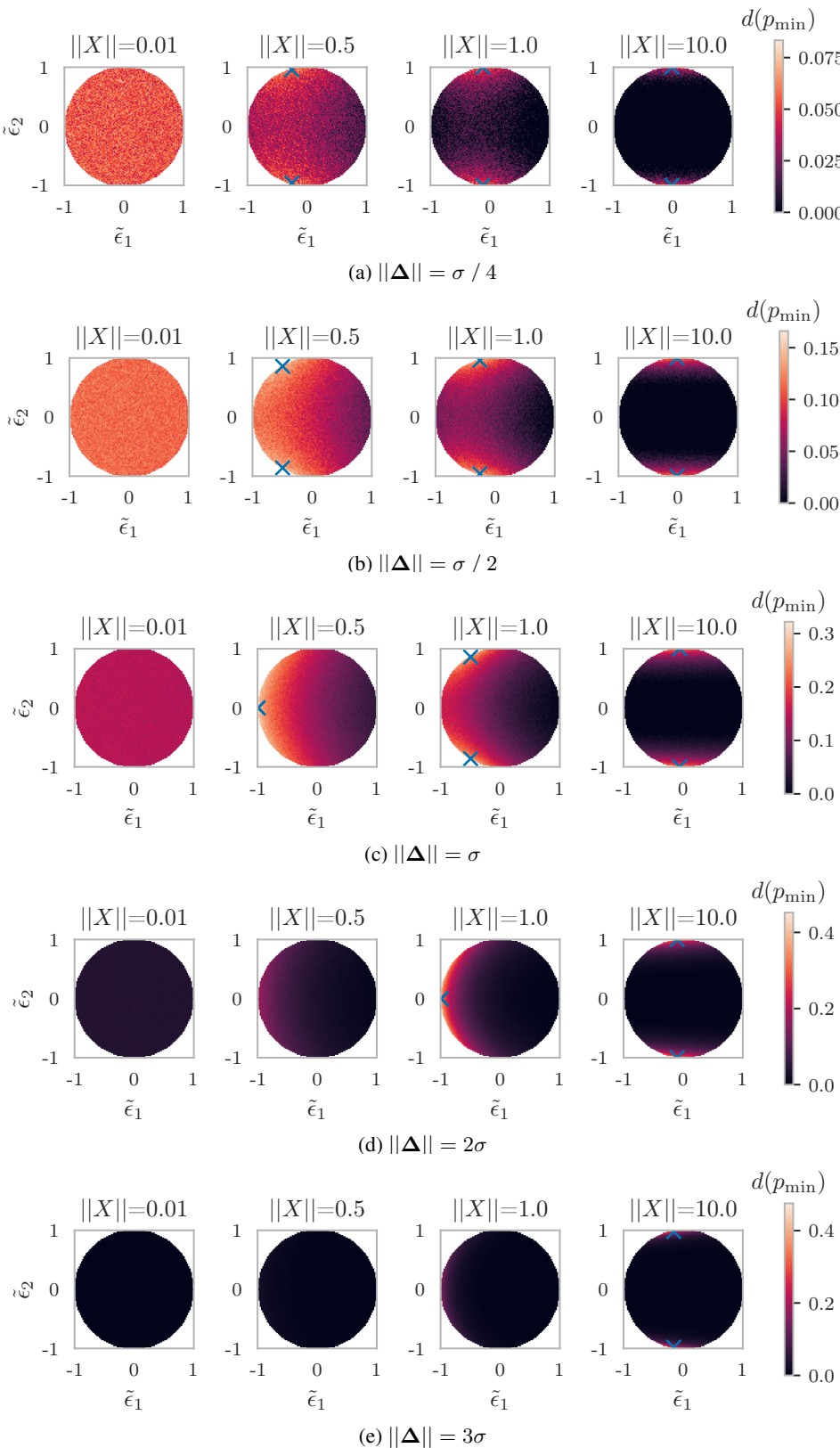

Figure 9: Difference in $p_{\min}$ between the tight certificate for 2D rotation invariance and the black-box baseline for $\sigma = 1$ under varying $||\boldsymbol{X}||$, $||\boldsymbol{\Delta}||$, $\tilde{\epsilon}_1$ and $\tilde{\epsilon}_2$. Blue crosses indicate adversarial rotations.

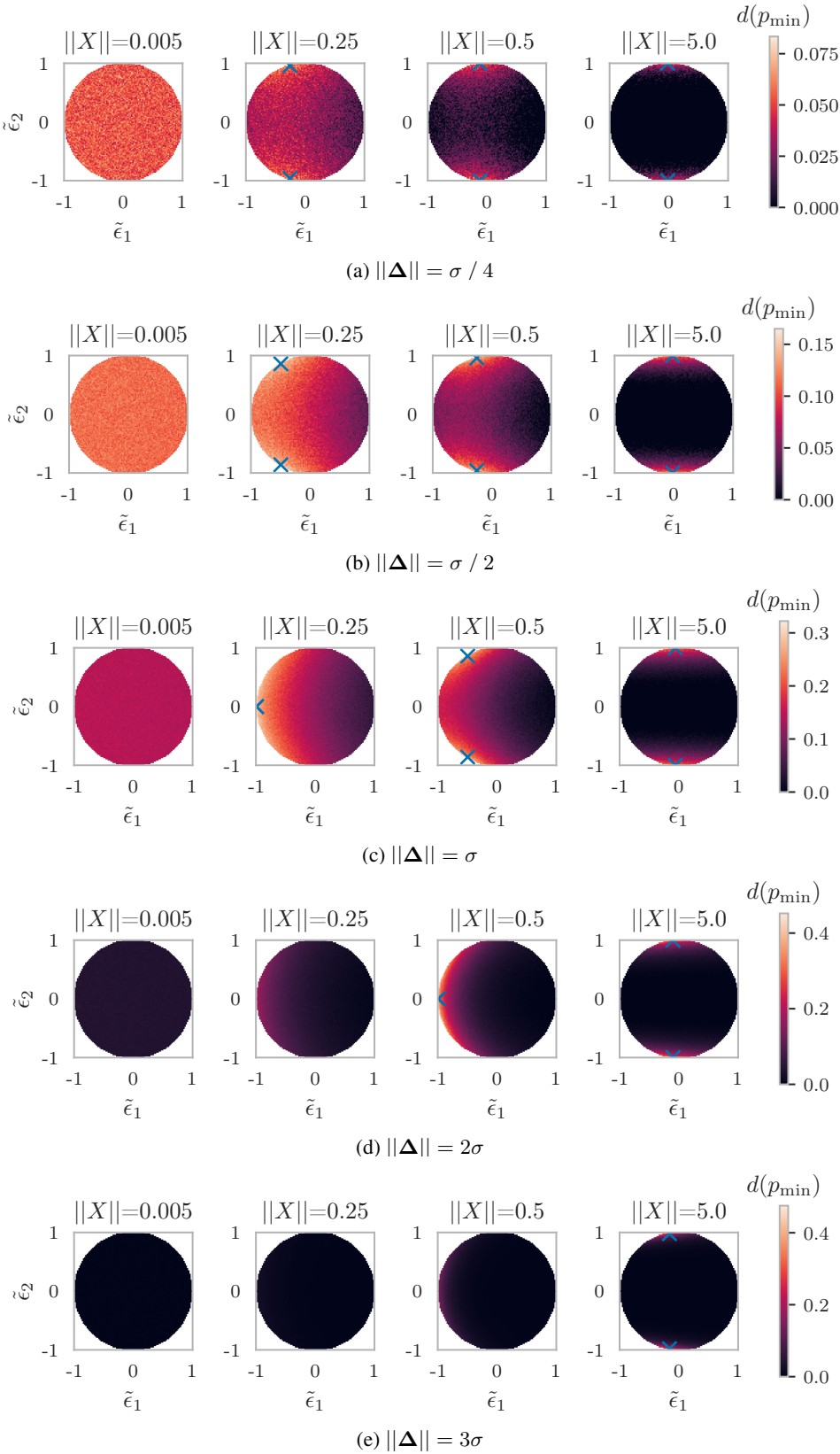

Figure 10: Difference in $p_{\min}$ between the tight certificate for 2D rotation invariance and the black-box baseline for $\sigma = 1 / 2$ under varying $||\bar{X}||$, $||\Delta||$, $\tilde{\epsilon}_1$ and $\tilde{\epsilon}_2$. The results are identical to Fig. 9, because the certificate only depends on the value of the parameters relative to $\sigma$.

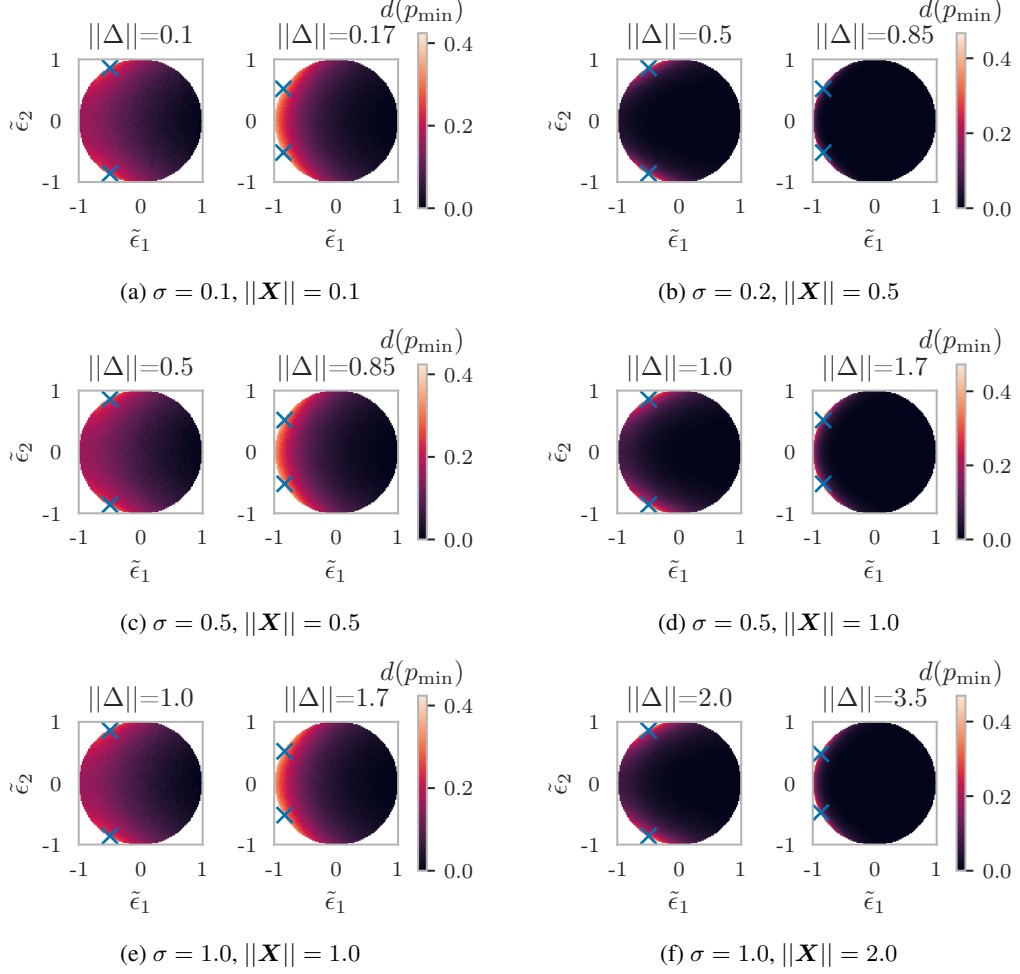

Figure 11: Difference in $p_{\min}$ between the tight certificate for 2D rotation invariance and the black-box baseline for different combinations of $\sigma$, $||\boldsymbol{X}||_2$ and $||\boldsymbol{\Delta}||_2$. Parameter $||\boldsymbol{\Delta}||_2$ is chosen such that adversarial rotations (indicated with blue crosses) correspond to $\tilde{\epsilon} = \approx 0.5$ or $\tilde{\epsilon}_2 \approx \pm 0.5$. The regions with large $p_{\min}$ are concentrated around adversarial rotations.

## A.2 Application to point cloud classification

Next, we repeat the experiments from Section 8.2 for diverse combinations of parameters and classifier architectures. We focus on the case $\sigma \leq 0.25$, where the randomly smoothed models achieve at least 50% accuracy (see Fig. 5), as even smaller accuracies are too low to be of any practical use.

We first consider adversarial scaling on the point clouds constructed from MNIST (Appendix A.2.1). We then consider perturbations with random rotation components on MNIST (Appendix A.2.2) and ModelNet40 (Appendix A.2.3). For the tight certificates, we show both probabilistic upper and lower bounds obtained via Monte Carlo evaluation (recall Section 6.3). Note that only the lower bound is guaranteed to be a valid certificate with high probability.

### A.2.1 Adversarial scaling on MNIST

Fig. 12 shows the certified accuracy (i.e. the number of correct and certifiably robust predictions) under adversarial scaling (i.e. $\tilde{\epsilon}_1 = 1, \tilde{\epsilon}_2 = 0$) for the randomly smoothed EnsPointNet with different $\sigma$ on the MNIST test set. We evaluate the tight and orbit-based certificate for rotation invariance ($SO(2)$) and those for simultaneous rotation and translation invariance ($SE(2)$).

In all cases, the tight and orbit-based certificates yield similar certified accuracies. The certified accuracies of the tight certificate deviate slightly from the orbit-based ones, because we compute a probabilistic bound that holds with high probability $1 - \alpha$, rather than evaluating it exactly. The gap between the probalistic lower bound and the orbit-based certificate is particularly large for $\sigma = 0.1$, which can be explained by our observations from Appendix A.1: The smaller $\sigma$ is, relative to $||\boldsymbol{X}||_2$ (which is defined by each datapoint of the test set and thus constant), the smaller the benefit of using it over the orbit-based baseline becomes.

Additionally enforcing translation invariance yields stronger robustness guarantees. For instance, with $\sigma = 0.2$, the $SE(2)$ certificates can still certify some predictions for $||\boldsymbol{\Delta}||_2 = 0.8$, whereas the $SO(2)$ certificates can not certify robustness beyond $||\boldsymbol{\Delta}||_2 \approx 0.62$.

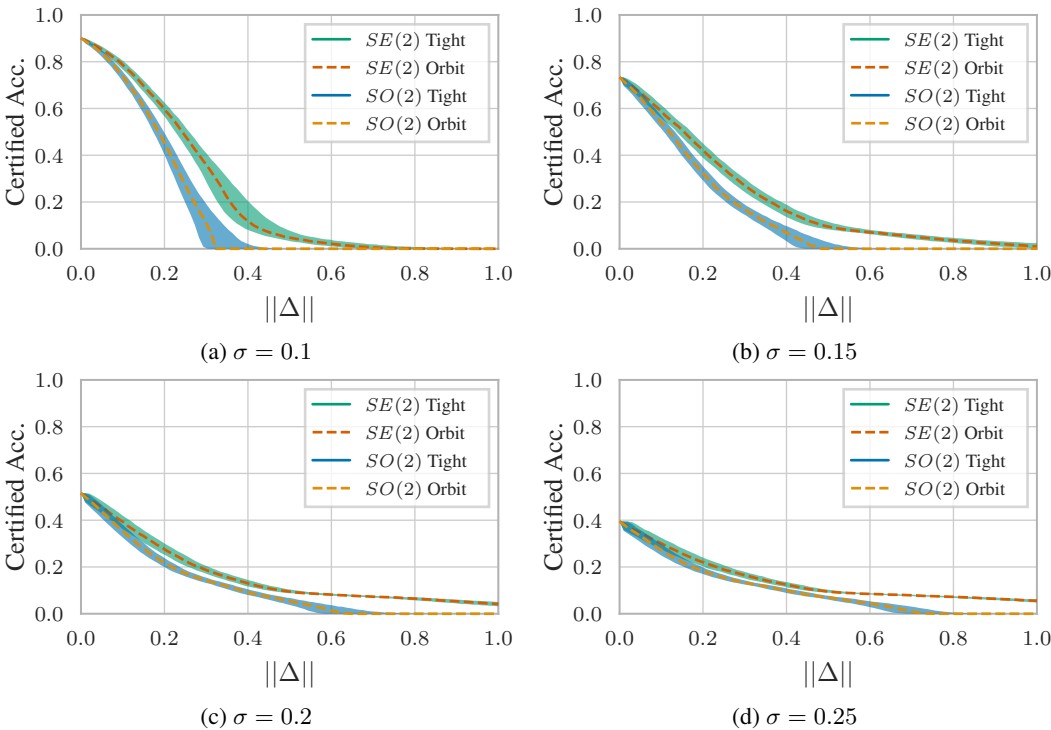

Figure 12: Comparison of certificates for adversarial scaling of MNIST with EnsPointNet and $\sigma \in \{0.1, 0.15, 0.2, 0.25\}$. Tight and orbit-based certificates yield similar certified accuracies.

### A.2.2 Perturbations with random rotation components on MNIST

Like in Section 8.2, we construct perturbations with rotational components by fixing $||\Delta||$, randomly sampling perturbations of the specified norm and then rotating $Z = X + \Delta$ by a specified angle $\theta$. We obtain the original, unrotated perturbations via sampling from an isotropic matrix normal distribution and then scaling the sample to be of the desired norm. Per value of $\theta$, we sample ten such perturbed inputs per element of the test set (i.e. $100000$ per $\theta$). We then compare the tight and the orbit-based certificate for simultaneous rotation and translation invariance ($SE$), as well as the black-box baseline. As our metric, we use probabilistic certified accuracy, i.e. the percentage of samples $X'$ for which $f(X)$ is correct and $f(X') = f(X)$ is provably guaranteed.

Fig. 13 shows the results for $\sigma \in \{0.05, 0.1, 0.2\}$, $||\Delta||_2 \in \{\sigma\ /\ 2, \sigma, 2\sigma\}$ and $\theta \in [0°, 90°]$. While the black-box baseline fails to certify robustness even for small rotation angles, both gray-box certificates effectively eliminate the rotations, i.e. are constant in $\theta$. However, the tight gray-box certificate does not offer any notable benefit beyond that. The probabilistic lower bound never yields better probabilistic certified accuracy than the orbit-based certificate.

These results are consistent with our observartions about the tight certificate's parameter space: All values of $\sigma$ that retain high accuracy are small, relative to the average norm of the test set (10.67).

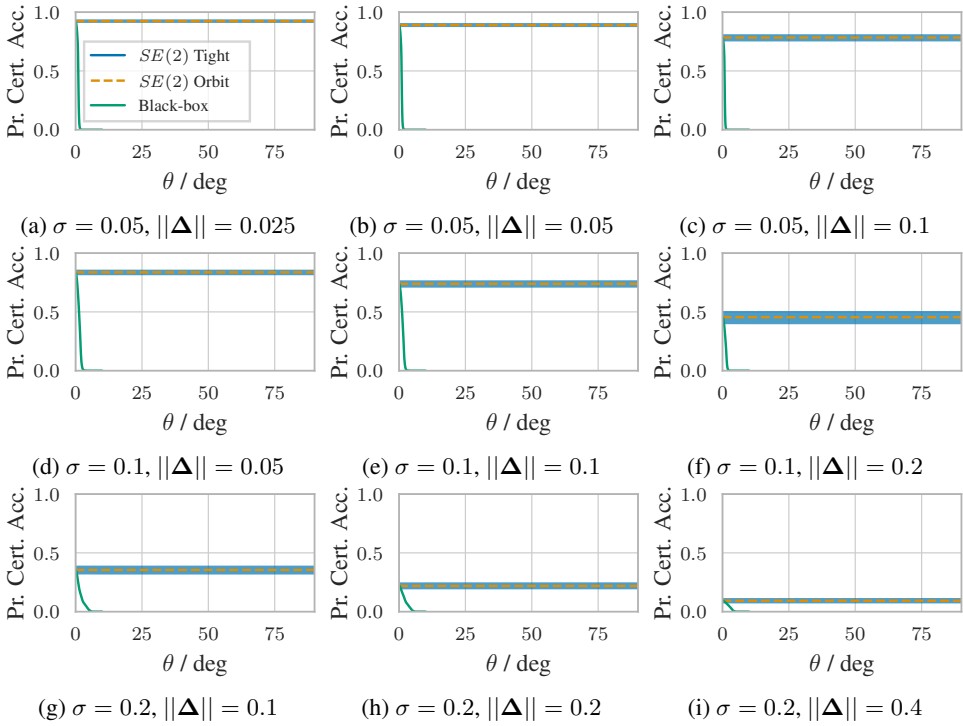

Figure 13: Comparison of tight gray-box, orbit-based gray-box and black-box certificates for MNIST with EnsPointNet and different $\sigma$, with respect to probabilistic certified accuracy. Perturbed inputs are generated by sampling perturbations with $||\Delta||_2 \in \{\sigma\ /\ 2, \sigma, 2\sigma\}$ and rotating by angle $\theta$. Group $SE(2)$ refers to simultaneous rotation and translation invariance. The orbit-based certificate is a good approximation of the tight certificate.

### A.2.3 Perturbations with random elemental rotation components on ModelNet40

Next, we repeat the same experiment on ModelNet40. After sampling a perturbation $\mathbf{\Delta}$ of specified norm $||\mathbf{\Delta}||_2$, we randomly rotate $X' = X + \mathbf{\Delta}$ by angle $\theta$ around a random axis. We obtain this axis by sampling from a three-dimensional isotropic normal distribution and normalizing the result.

We then compare the tight certificate for simultaneous invariance under rotation and translation to the orbit-based one and the black-box baseline.

Figs. 14 to 16 show the results for EnsPointNet, AttnPointNet and EnsDGCNN. In all cases, both the tight- and orbit-based certificate are constant in $\theta$, but the tight certificate does not noticeably improve upon the orbit-based one w.r.t. probabilistic certified accuracy.

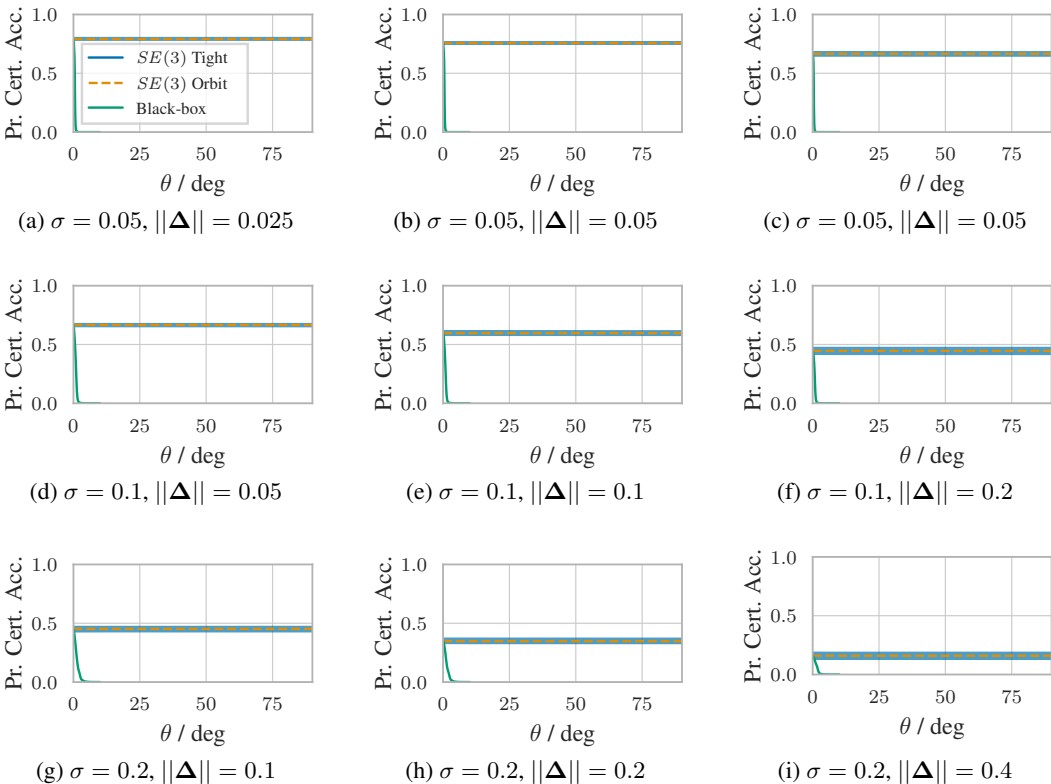

Figure 14: Comparison of tight gray-box, orbit-based gray-box and black-box certificates for Model-Net40 with EnsPointNet and different $\sigma$, w.r.t. probabilistic certified accuracy. Perturbed inputs are generated by sampling perturbations with $||\mathbf{\Delta}||_2 \in \{\sigma \,/\, 2, \sigma, 2\sigma\}$ and rotating around a random axis by angle $\theta$. Group $SE(3)$ refers to simultaneous rotation and translation invariance. The orbit-based certificate is a good approximation of the tight certificate.

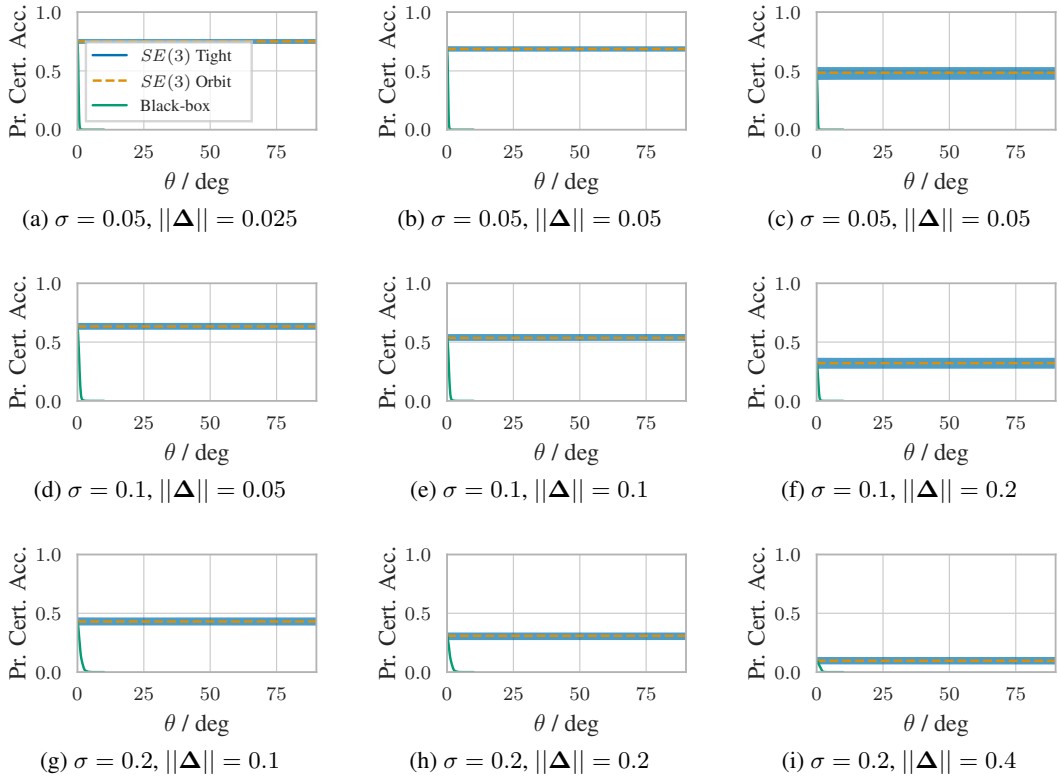

Figure 15: Comparison of tight gray-box, orbit-based gray-box and black-box certificates for Model-Net40 with AttnPointNet and different $\sigma$, w.r.t. probabilistic certified accuracy. Perturbed inputs are generated by sampling perturbations with $||\mathbf{\Delta}||_2 \in \{\sigma\,/\,2, \sigma, 2\sigma\}$ and rotating around a random axis by angle $\theta$. Group $SE(3)$ refers to simultaneous rotation and translation invariance. The orbit-based certificate is a good approximation of the tight certificate.

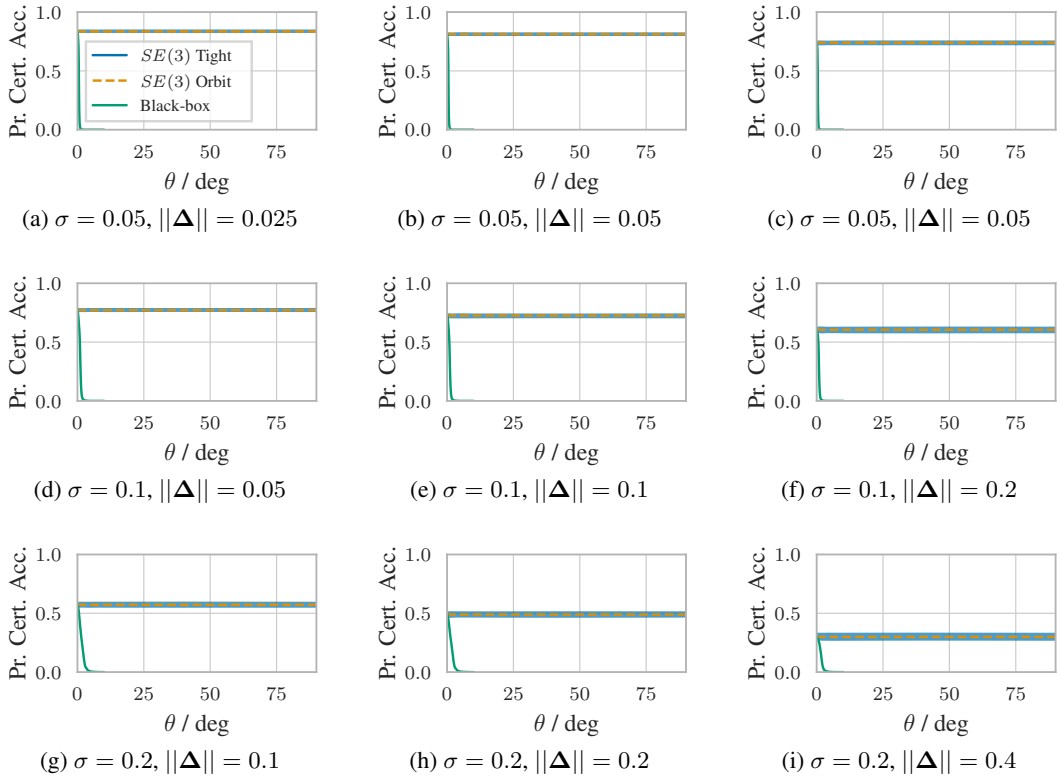

Figure 16: Comparison of tight gray-box, orbit-based gray-box and black-box certificates for Model-Net40 with EnsDGCNN and different $\sigma$, w.r.t. probabilistic certified accuracy. Perturbed inputs are generated by sampling perturbations with $||\boldsymbol{\Delta}||_2 \in \{\sigma \,/\, 2, \sigma, 2\sigma\}$ and rotating around a random axis by angle $\theta$. Group $SE(3)$ refers to simultaneous rotation and translation invariance. The orbit-based certificate is a good approximation of the tight certificate.

# B   Full experimental setup

In the following, we provide a full specification of the used models, data sets and training parameters (Appendix B.1, randomized smoothing parameters (Appendix B.2), as well as the used computational resources (Appendix B.3) and third-party assets (Appendix B.4).

The code and all configuration files needed for reproducing the experimental results will be made available at https://www.cs.cit.tum.de/daml/invariance-smoothing.

## B.1   Models, data and training parameters

### B.1.1   Models

The experiments in Section 8.2 and Appendix A.2 are performed with three models: EnsPointNet, EnsDGCNN and AttnPointNet, which are rotation and translation invariant versions of PointNet [6], and the Dynamic Graph Convolutional Neural Network [128].

**EnsPointNet** is based on a standard PointNet architecture with an input T-Net but without a feature T-Net (we did not find the feature T-Net to improve the accuracy of the rotation invariant model). The input T-Net uses three convolutional layers with kernel size $1 \times 1$ (64, 128 and 1024 filters) and three linear layers (512, 256 and 9 neurons). The PointNet itself uses three convolutional layers with kernel size $1 \times 1$ (64, 128 and 1024 filters) and three linear layers (512, 256 and $|\mathbb{Y}|$ neurons). All layers, except the last one, use BatchNorm ($\epsilon = 1e - 05$, $\mathrm{momentum} = 0.1$). The second linear layer uses dropout ($p = 0.4$). To achieve rotation and translation invariance, the input data is centered and the two (for $2D$ data) or three (for $3D$ data) principal components are computed. Principal components are not unique. One has to account for order and sign ambiguities to ensure rotation invariance (see discussion in [129, 130]). If two or more eigenvalues are numerically close (relative tolerance $1e^{-5}$, absolute tolerance $1e^{-8}$), we iterate over all possible eigenvector signs and orders ($8 \cdot 6$), multiply the input data with the principal components, and pass it through the PointNet. If the eigenvalues are distinct, we sort them in ascending order and iterate over all possible signs (8). The 8 or 48 logit vectors are then averaged to obtain a prediction.

**EnsDGCNN** is based on a standard DGCNN architecture with an input spatial transform. The spatial transform uses uses three convolutional layers with kernel size $1 \times 1$ (64, 128 and 1024 filters) and three linear layers (512, 256 and 9 neurons). The DGCNN encodes the spatially transformed input with three DGC layers (64, 64, 64 and 128 filters and $k = 20$). The encoder output and residuals are concatenated and passed through a convolution with kernel size $1 \times 1$ (1024 filters), followed by max-pooling and three linear layers (512, 256, $|\mathbb{Y}|$ neurons). All layers use BatchNorm ($\epsilon = 1e - 05$, $\mathrm{momentum} = 0.1$). The first two linear layers use dropout ($p = 0.5$). We use the same ensembling approach as for EnsPointNet to achieve rotation and translation invariance.

**AttnPointNet** combines PointNet with the attention-based mechanism for combining canonical poses proposed in [129]. It uses the same encoder as EnsPointNet. After passing the different PCA-based canonical poses through the encoder, the hidden vectors are combined via a self-attention layer (1024 neurons each for query, key and value transform) and then passed through the same decoder as in EnsPointNet. To reduce memory usage, we only consider sign combinations that correspond to proper rotation matrices (see discussion in [130]).

### B.1.2   Data

**ModelNet40** [126] consists of 12311 CAD models from 40 categories, split into 9843 training samples and 2468 test samples. We subdivide the original training set into $80\%$ train data and $20\%$ validation data. The same split is used for all experiments. Each CAD model is then transformed into a 3D point cloud with 1024 points using the same pre-processing steps as in [6], i.e. randomly sampling from the mesh faces according to surface area and normalizing the resulting point cloud into the unit sphere.

**MNIST** [127] consists of 70000 handwritten digits, split into 60000 training samples and 10000 test samples. We subdivide the original training set into $80\%$ train data and $20\%$ validation data. The same split is used for all experiments. Each image is transformed into a 2D point with 1024 points using the same pre-processing steps as in [6], i.e. mapping all pixels with values greater than 128 to x-y coordinates, then uniformly sub-sampling or padding with a randomly chosen point and finally

normalizing into the unit sphere. We additionally combine classes 6 and 9, because these digits cannot be expected to be differentiated by a rotation-invariant model.

### B.1.3 Training parameters

All models are trained on samples from the smoothing distribution (i.e. the model that is evaluated and certified with $\sigma$ is trained on data augmented with Gaussian noise sampled from an isotropic normal distribution with standard deviation $\sigma$). Per element of a training batch, we take exactly one sample from the smoothing distribution. We do not use consistency regularization.

**EnsPointNet** is trained with cross entropy loss and the TNet regularization from [6] (weight $0.0001$). Training is performed for 200 epochs with batch size 128 using Adam ($\beta_1 = 0.9, \beta_2 = 0.99$, $\epsilon = 1e-8$, weight_decay $= 1e^-4$). The learning rate starts at $0.001$ and is multiplied with $0.7$ every 20 epochs. The input data is randomly scaled by a factor from $[0.8, 1.25]$ and then transformed via principal component analysis, using a single set of eigenvectors with randomized sign and order.

**EnsDGCNN** is trained in the same manner as EnsPointNet, but with batch size 32, 400 epochs and learning rate decay every 40 epochs.

**AttnPointNet** is trained with batch size 64, 400 epochs and learning rate decay every 40 epochs. The other parameter values are identical to those for EnsPointNet. In order to train the self-attention layer, we do not transform the input with a single set of eigenvectors, but pass all possible eigenvectors corresponding to rotations through the network.

## B.2 Randomized smoothing parameters

We use 1000 samples from the smoothing distribution to compute smoothed predictions. Abstentions, i.e. predictions for which we cannot guarantee that $p_{\boldsymbol{X},y^*} \geq 0.5$, are considered as incorrect.

For certification, we set the significance parameter $\alpha$ to $0.01$. i.e. each certificate holds with probability $99.9\%$. We use 10000 samples per confidence bound, i.e. 10000 samples to bound $p_{\boldsymbol{X},y^*}$. For the tight gray-box certificates involving rotation invariance we additionally use 10000 samples to bound the threshold $\kappa$ and 10000 samples to bound the optimal value itself (for more details, see Appendix F.5). We then use Holm-Bonferroni correction ([131]) to ensure that all three confidence bounds hold simultaneously with probability $1 - \alpha$.

Evaluating the tight certificates for $SO(3)$ and $SE(3)$ requires numerical integration over the two-dimensional parameter space $[-\frac{\pi}{2}, \frac{\pi}{2}] \times [0, 2\pi]$ (see Appendix F.4.3). We use Clenshaw-Curtis quadrature with degree 20 in each dimension.

## B.3 Computational resources

**Training and Monte Carlo sampling of smoothed predictions** was performed on a single NVIDIA A100 GPU (40 GB VRAM) with an AMD EPYC 7543 CPU @ 2.8 GHz. For EnsPointNet and AttnPointNet, 16 GB RAM were allocated. For EnsDGCNN, which requires additional memory for computation of k-nearest-neighbor graphs, 32 GB RAM were allocated. The average time for training EnsPointNet on MNIST was $42.8$ min. The average time for training EnsPointNet, AttnPoint and EnsDGCNN on ModelNet40 was $20.2$ min, $114$ min and $158$ min, respectively. The average time for obtaining 11000 samples from EnsPointNet on MNIST was $1.1$ s. The average time for obtaining 11000 samples from EnsPointNet, AttnPoint and EnsDGCNN on ModelNet40 was $1.59$ s, $1.4$ s and $24.91$ s, respectively.

**Computation of certificates** was performed on an Intel Xeon E5-2630 v4 CPU @ 2.2 GHz, with 16 GB RAM allocated to each experiment. The average time for computing a tight certificate for 2D rotation invarance (i.e. computing threshold $\kappa$ and bounding the optimal value, using 10000 samples each), was $0.05$ s. The average time for computing a tight certificate for 3D rotation invariance was $6.3$ s. The increase in computational cost is caused by the fact that we can only evaluate the worst-case classifier via numerical integration. The average time for computing the black-box randomized smoothing certificate and the tight certificate for translation invariance was in the sub-milisecond range and can thus not be accurately reported.

## B.4 Third-party assets

Our work uses the publicly available MNIST [127] and ModelNet40 [126] datasets, as well as the quadpy quadrature library [132]. Our implementation additionally uses code from a publicly available implementation of PointNet and PointNet++ [133], as well as the author's reference implementation of DGCNN [128]. Both are available under MIT license.

# C  Group definitions

In the following, we define the different groups we consider in our work, each corresponding to a specific type of spatial invariance. Recall from Section 3.2 that a group is a set $\mathbb{T}$, associated with a binary operator $\cdot : \mathbb{T} \times \mathbb{T} \to \mathbb{T}$ that is closed and associative under the operation, has an inverse $t^{-1}$ for each element $t \in \mathbb{T}$ and features an identity element $e$. Further recall that we associate each group with a group action $\circ : \mathbb{T} \times \mathbb{R}^{N \times D} \to \mathbb{R}^{N \times D}$ that describes how elements of the group modify elements of the input space.

## C.1  Translation group $T(D)$

The translation group in $D$ dimensions is the set of all $D$-dimensional continuous vectors, i.e. $T(D) = \mathbb{R}^D$. Two group elements $\boldsymbol{t}, \boldsymbol{t}' \in T(D)$ are combined via vector addition, i.e. $\boldsymbol{t} \cdot \boldsymbol{t}' = \boldsymbol{t} + \boldsymbol{t}'$. A group element $\boldsymbol{t} \in T(D)$ acts on an input $\boldsymbol{X} \in \mathbb{R}^{N \times D}$ via row-wise addition, i.e. $\boldsymbol{t} \circ \boldsymbol{X} = \boldsymbol{X} + \boldsymbol{1}_N \boldsymbol{t}^T$ with all-ones vector $\boldsymbol{1}_N \in \mathbb{R}^N$. The identity element $e$ is the all-zeros vector $\boldsymbol{0}_D$.

## C.2  Rotation group $SO(D)$

The rotation group in $D$ dimensions, $SO(D)$ (short for *special orthogonal*), is the set of all $D$-dimensional rotation matrices, i.e. $SO(D) = \left\{ \boldsymbol{R} \in \mathbb{R}^{D \times D} \mid \boldsymbol{R}^T \boldsymbol{R} = \mathbf{I}_N \wedge \det(\boldsymbol{R}) = 1 \right\}$. Two group elements $\boldsymbol{R}, \boldsymbol{R}' \in SO(D)$ are combined via matrix multiplication, i.e. $\boldsymbol{R} \cdot \boldsymbol{R}' = \boldsymbol{R}\boldsymbol{R}'$. A group element $\boldsymbol{R} \in SO(D)$ acts on an input $\boldsymbol{X} \in \mathbb{R}^{N \times D}$ via row-wise matrix-vector multiplication, i.e. $\boldsymbol{R} \circ \boldsymbol{X} = \boldsymbol{X} \boldsymbol{R}^T$. The identity element $e$ is the identity matrix $\mathbf{I}_D$.

## C.3  Orthogonal group $O(D)$

The orthogonal group in $D$ dimensions, $O(D)$, is the set of all $D$-dimensional orthogonal matrices, i.e. $O(D) = \left\{ \boldsymbol{A} \in \mathbb{R}^{D \times D} \mid \boldsymbol{R}^T \boldsymbol{R} = \mathbf{I}_N \right\}$. Note that $O(D) \supset SO(D)$. Like the rotation group, group elements are combined via matrix multiplication, i.e. $\boldsymbol{A} \cdot \boldsymbol{A}' = \boldsymbol{A}\boldsymbol{A}'$ and act on an input $\boldsymbol{X} \in \mathbb{A}^{N \times D}$ via row-wise matrix-vector multiplication, i.e. $\boldsymbol{A} \circ \boldsymbol{X} = \boldsymbol{X} \boldsymbol{A}^T$. The identity element $e$ is the identity matrix $\mathbf{I}_D$.

## C.4  Roto-translation group $SE(D)$

The roto-translation group in $D$ dimensions, $SE(D)$ (short for *special Euclidean*), is composed of all pairs of $D$-dimensional rotation matrices and translation vectors. That is, $SE(D) = SO(D) \times T(D)$. Two group elements $(\boldsymbol{R}, \boldsymbol{t}), (\boldsymbol{R}', \boldsymbol{t}') \in SE(D)$ are combined via a semidirect product, i.e. $(\boldsymbol{R}, \boldsymbol{t}) \cdot (\boldsymbol{R}', \boldsymbol{t}') = (\boldsymbol{R}\boldsymbol{R}', \boldsymbol{R}\boldsymbol{t}' + \boldsymbol{t})$. A group element $(\boldsymbol{R}, \boldsymbol{t}) \in SE(D)$ acts on an input $\boldsymbol{X} \in \mathbb{R}^{N \times D}$ via row-wise matrix-vector multiplication, followed by a row-wise addition i.e. $(\boldsymbol{R}, \boldsymbol{t}) \circ \boldsymbol{X} = \boldsymbol{X} \boldsymbol{R}^T + \boldsymbol{1}_N \boldsymbol{t}^T$. The identity element $e$ is $(\mathbf{I}_D, \boldsymbol{0}_D)$.

## C.5  Euclidean group $E(D)$

The Euclidean group in $D$ dimensions, $E(D)$, corresponds to the set of all distance-preserving functions in Euclidean space. It is composed of all pairs of $D$-dimensional orthonormal matrices and translation vectors. That is, $E(D) = O(D) \times T(D)$. Note that $E(D) \supset SE(D)$. Like the roto-translation group, group elements combined via a semidirect product, i.e. $(\boldsymbol{A}, \boldsymbol{t}) \cdot (\boldsymbol{A}', \boldsymbol{t}') = (\boldsymbol{A}\boldsymbol{A}', \boldsymbol{A}\boldsymbol{t}' + \boldsymbol{t})$ and act on an input $\boldsymbol{X} \in \mathbb{R}^{N \times D}$ via row-wise matrix-vector multiplication, followed by a row-wise addition i.e. $(\boldsymbol{A}, \boldsymbol{t}) \circ \boldsymbol{X} = \boldsymbol{X} \boldsymbol{A}^T + \boldsymbol{1}_N \boldsymbol{t}^T$. The identity element $e$ is $(\mathbf{I}_D, \boldsymbol{0}_D)$.

## C.6  Permutation group $S(N)$

The permutation group in $N$ dimensions, $S(N)$, is the set of all permutation matrices, i.e. $S(N) = \left\{ \boldsymbol{P} \in \{0, 1\}^{N \times N} \mid \boldsymbol{P}^T \boldsymbol{P} = \mathbf{I}_N \right\}$. Two group elements $\boldsymbol{P}, \boldsymbol{P}' \in S(N)$ are combined via matrix multiplication, i.e. $\boldsymbol{P} \cdot \boldsymbol{P}' = \boldsymbol{P}\boldsymbol{P}'$. A group element $\boldsymbol{P} \in S(N)$ acts on an input $\boldsymbol{X} \in \mathbb{R}^{N \times D}$ via matrix multiplication, i.e. $\boldsymbol{P} \circ \boldsymbol{X} = \boldsymbol{P}\boldsymbol{X}$. Note that the group action permutes the rows and not the columns. The identity element $e$ is the identity matrix $\mathbf{I}_N$.

# D  Proof of Theorem 1

**Theorem 1.** *Let base classifier $g : \mathbb{R}^{N \times D} \to \mathbb{Y}$ be invariant under group $\mathbb{T}$ with $\mathbb{T} \subseteq E(D)$ or $\mathbb{T} \subseteq S(N)$, where $E(D)$ is the Euclidean group and $S(N)$ is the permutation group. Then the isotropically smoothed classifier $f$, as defined in Section 3.1, is also invariant under $\mathbb{T}$.*

Recall that

$$f(\boldsymbol{X}) = \text{argmax}_{y \in \mathbb{Y}} p_{\boldsymbol{X},y}$$

$$\text{with } p_{\boldsymbol{X},y} = \Pr_{\boldsymbol{Z} \sim \mu_{\boldsymbol{X}}} [g(\boldsymbol{Z}) = y]$$

$$\text{and } \mu_{\boldsymbol{X}}(\boldsymbol{Z}) = \prod_{d=1}^{D} \mathcal{N}\left(\boldsymbol{Z}_{:,d} \mid \boldsymbol{X}_{:,d}, \sigma^2 \mathbf{I}_N\right).$$

We shall prove Theorem 1 by showing that $\forall t \in \mathbb{T} : p_{\boldsymbol{X},y} = p_{(t \circ \boldsymbol{X}),y}$, i.e. the prediction probabilities are invariant. To do so, we need the following simple lemma:

**Lemma 2.** *Consider an arbitrary invertible matrix $\boldsymbol{A} \in \mathbb{R}^{N \times N}$, vectors $\boldsymbol{z}, \boldsymbol{m} \in \mathbb{R}^N$ and scalar $\sigma > 0$. Then*

$$\left| \det\left(\boldsymbol{A}^{-1}\right) \right| \mathcal{N}\left(\boldsymbol{A}^{-1}\boldsymbol{z} \mid \boldsymbol{m}, \sigma^2 \mathbf{I}_N\right) = \mathcal{N}\left(\boldsymbol{z} \mid \boldsymbol{A}\boldsymbol{m}, \sigma^2 \boldsymbol{A}\boldsymbol{A}^T\right).$$

*Proof.* It follows from the change of variables formula for densitiy functions that the l.h.s. term is the density function of random variable $\boldsymbol{z}' = \boldsymbol{A}\boldsymbol{z}$ with $\boldsymbol{z} \sim \mathcal{N}\left(\boldsymbol{m}, \sigma^2 \mathbf{1}_N\right)$. Due to the behavior of multivariate normal distributions under affine transformation, the r.h.s. term is also the density function of $\boldsymbol{z}'$. $\qquad \square$

We begin by proving Theorem 1 for $\mathbb{T} \subseteq E(D)$, before considering $\mathbb{T} \subseteq S(N)$. In the following, let $\hat{g}(\boldsymbol{Z}) = \mathbb{1}\left[g(\boldsymbol{Z}) = y^*\right]$ indicate whether base classifier $g$ classifies input $\boldsymbol{Z}$ as $y^*$.

**Case 1:** Assume that $\mathbb{T} \subseteq E(D)$ and consider an arbitrary $t \in \mathbb{T}$. By definition of the Euclidean group and the associated group action (see Appendix C.5) we must have $t = (\boldsymbol{A}, \boldsymbol{b})$ and $t \circ \boldsymbol{X} = \boldsymbol{X}\boldsymbol{A}^T + \mathbf{1}_N \boldsymbol{b}^T$ for some orthogonal matrix $\boldsymbol{A} \in \mathbb{R}^{D \times D}$ and translation vector $\boldsymbol{b} \in \mathbb{R}^N$. By definition of the smoothing distribution, we have

$$p_{(t \circ \boldsymbol{X}),y} = \int_{\mathbb{R}^{N \times D}} \hat{g}(\boldsymbol{Z}) \mu_{(\boldsymbol{X}\boldsymbol{A}^T + \mathbf{1}_N \boldsymbol{b}^T)}(\boldsymbol{Z}) \, d\boldsymbol{Z}$$

$$= \int_{\mathbb{R}^{N \times D}} \hat{g}(\boldsymbol{Z}) \prod_{n=1}^{N} \mathcal{N}\left(\boldsymbol{Z}_n \mid \boldsymbol{A}\boldsymbol{X}_n + \boldsymbol{b}, \sigma^2 \mathbf{I}_D\right) \, d\boldsymbol{Z}$$

$$= \int_{\mathbb{R}^{N \times D}} \hat{g}(\boldsymbol{Z}) \prod_{n=1}^{N} \mathcal{N}\left(\boldsymbol{Z}_n - \boldsymbol{b} \mid \boldsymbol{A}\boldsymbol{X}_n, \sigma^2 \mathbf{I}_D\right) \, d\boldsymbol{Z},$$

where in the last equality we have used that $\mathcal{N}(z \mid m + b) = \mathcal{N}(z - b \mid m)$. We can now perform two substitutions, $\boldsymbol{V} = \boldsymbol{Z} - \mathbf{1}_N \boldsymbol{b}^T$ and $\boldsymbol{W} = \boldsymbol{V}(\boldsymbol{A}^{-1})^T$, to transform this term into an expectation w.r.t. the original smoothing distribution $\mu_{\boldsymbol{X}}$:

$$= \int_{\mathbb{R}^{N \times D}} \hat{g}(\boldsymbol{V} + \mathbf{1}_N \boldsymbol{b}^T) \prod_{n=1}^{N} \mathcal{N}\left(\boldsymbol{V}_n \mid \boldsymbol{A}\boldsymbol{X}_n, \sigma^2 \mathbf{I}_D\right) \, d\boldsymbol{V}$$

$$= \int_{\mathbb{R}^{N \times D}} \hat{g}(\boldsymbol{V} + \mathbf{1}_N \boldsymbol{b}^T) \prod_{n=1}^{N} \mathcal{N}\left(\boldsymbol{V}_n \mid \boldsymbol{A}\boldsymbol{X}_n, \sigma^2 \boldsymbol{A}\boldsymbol{A}^T\right) \, d\boldsymbol{V}$$

$$= \int_{\mathbb{R}^{N \times D}} \hat{g}(\boldsymbol{V} + \mathbf{1}_N \boldsymbol{b}^T) \prod_{n=1}^{N} \mathcal{N}\left(\boldsymbol{A}^{-1}\boldsymbol{V}_n \mid \boldsymbol{X}_n, \sigma^2 \mathbf{I}_N\right) \, d\boldsymbol{V}$$

$$= \int_{\mathbb{R}^{N \times D}} \hat{g}(\boldsymbol{W}\boldsymbol{A}^T + \mathbf{1}_N \boldsymbol{b}^T) \prod_{n=1}^{N} \mathcal{N}\left(\boldsymbol{W}_n \mid \boldsymbol{X}_n, \sigma^2 \mathbf{I}_N\right) \, d\boldsymbol{W}.$$

In the second step we have used the fact that $\boldsymbol{A}$ is orthogonal, i.e. $\boldsymbol{A}\boldsymbol{A}^T = \mathbf{I}_N$. In the third step, we have applied Lemma 2. Note that, because $|\det(\boldsymbol{A})| = 1$, we did not have to change the volume element. Finally, we can use the fact that $g$ is invariant w.r.t. group $\mathbb{T}$ to prove that

$$
\begin{aligned}
p_{(t\circ\boldsymbol{X}),y} &= \int_{\mathbb{R}^{N\times D}} \hat{g}(\boldsymbol{A}\boldsymbol{W} + \mathbf{1}_N\boldsymbol{b}^T) \prod_{n=1}^{N} \mathcal{N}\left(\boldsymbol{W}_n \mid \boldsymbol{X}_n, \sigma^2\mathbf{I}_N\right) \, d\boldsymbol{W} \\
&= \int_{\mathbb{R}^{N\times D}} \hat{g}(\boldsymbol{W}) \prod_{n=1}^{N} \mathcal{N}\left(\boldsymbol{W}_n \mid \boldsymbol{X}_n, \sigma^2\mathbf{I}_N\right) \, d\boldsymbol{W} \\
&= p_{\boldsymbol{X},y}.
\end{aligned}
$$

**Case 2:** Assume that $\mathbb{T} \subseteq S(N)$ and consider an arbitrary $t \in \mathbb{T}$. By definition of the permutation group and the associated group action (see Appendix C.6) we must have $t = \boldsymbol{P}$ and $t \circ \boldsymbol{X} = \boldsymbol{P}\boldsymbol{X}$ for some orthogonal matrix $\boldsymbol{P} \in \{0,1\}^{N\times N}$. The proof is virtually identical to that for the Euclidean group, except we now use the substitution $\boldsymbol{V} = \boldsymbol{P}\boldsymbol{Z}$:

$$
\begin{aligned}
p_{(t\circ\boldsymbol{X}),y} &= \int_{\mathbb{R}^{N\times D}} \hat{g}(\boldsymbol{Z})\mu_{(\boldsymbol{P}\boldsymbol{X})}(\boldsymbol{Z}) \, d\boldsymbol{Z} \\
&= \int_{\mathbb{R}^{N\times D}} \hat{g}(\boldsymbol{Z}) \prod_{d=1}^{D} \mathcal{N}\left(\boldsymbol{Z}_{:,d} \mid \boldsymbol{P}\boldsymbol{X}_{:,d}, \sigma^2\mathbf{I}_N\right) \, d\boldsymbol{Z} \\
&= \int_{\mathbb{R}^{N\times D}} \hat{g}(\boldsymbol{Z}) \prod_{d=1}^{D} \mathcal{N}\left(\boldsymbol{Z}_{:,d} \mid \boldsymbol{P}\boldsymbol{X}_{:,d}, \sigma^2\boldsymbol{P}\boldsymbol{P}^T\right) \, d\boldsymbol{Z} \\
&= \int_{\mathbb{R}^{N\times D}} \hat{g}(\boldsymbol{Z}) \prod_{d=1}^{D} \mathcal{N}\left(\boldsymbol{P}^{-1}\boldsymbol{Z}_{:,d} \mid \boldsymbol{X}_{:,d}, \sigma^2\mathbf{I}_N\right) \, d\boldsymbol{Z} \\
&= \int_{\mathbb{R}^{N\times D}} \hat{g}(\boldsymbol{P}\boldsymbol{V}) \prod_{d=1}^{D} \mathcal{N}\left(\boldsymbol{V}_{:,d} \mid \boldsymbol{X}_{:,d}, \sigma^2\mathbf{I}_N\right) \, d\boldsymbol{V} \\
&= \int_{\mathbb{R}^{N\times D}} \hat{g}(\boldsymbol{V}) \prod_{d=1}^{D} \mathcal{N}\left(\boldsymbol{V}_{:,d} \mid \boldsymbol{X}_{:,d}, \sigma^2\mathbf{I}_N\right) \, d\boldsymbol{V} \\
&= p_{\boldsymbol{X},y}.
\end{aligned}
$$

# E   Orbit-based gray-box certificates

In the following, we first prove the soundness of our orbit-based approach to gray-box robustness certification. We then present more explicit characterizations of the orbit-based certificates, which let us determine whether a specific perturbed input is part of the augmented certified region.

## E.1   Proof of Theorem 2

**Theorem 2.** *Let $f \in \mathbb{R}^{N \times D} \to \mathbb{Y}$ be invariant under a group $\mathbb{T}$. Let $y^* = f(X)$ be a prediction that is certifiably robust to a set of perturbed inputs $\mathbb{B} \subseteq \mathbb{R}^{N \times D}$, i.e. $\forall Z \in \mathbb{B} : f(Z) = y^*$. Let $\tilde{\mathbb{B}} = \cup_{Z \in \mathbb{B}} [Z]_{\mathbb{T}}$. Then $\forall X' \in \tilde{\mathbb{B}} : f(X') = y^*$.*

*Proof.* Consider an arbitrary $X' \in \tilde{\mathbb{B}}$. Due to the definition of $\tilde{\mathbb{B}}$, there must be a $Z \in \mathbb{B}$ and a $t \in \mathbb{T}$ such that $X' = t \circ Z$. Since $t \in \mathbb{T}$, we know that $f(t \circ Z) = f(Z)$. Since $Z \in \mathbb{B}$, we know that $f(Z) = y^*$. By the transitive property, we have $f(X') = y^*$. $\qquad\square$

## E.2   Explicit characterizations

Recall that, for randomly smoothed models, a prediction $y^* = f(X)$ is certifiably robust within a Frobenius norm ball $\mathbb{B} = \{Z \mid ||Z - X||_2 < r\}$ with $r = \sigma \Phi^{-1}(p_{X,y^*})$. For this certificate, we can determine whether a specific perturbed input $X'$ is in augmented certified region $\tilde{\mathbb{B}} = \cup_{Z \in \mathbb{B}} [Z]_{\mathbb{T}}$ by finding a transformation that minimizes the Frobenius distance between $X'$ and $X$:

**Corollary 1.** *Let $f \in \mathbb{R}^{N \times D} \to \mathbb{Y}$ be invariant under group $\mathbb{T}$. Let $y^* = f(X)$ be a prediction that is certifiably robust to a set of perturbed inputs $\mathbb{B} = \{Z \mid ||Z - X||_2 < r\}$, i.e. $\forall Z \in \mathbb{B} : f(Z) = y^*$. If $\min_{t \in \mathbb{T}} ||(t \circ X') - X||_2 < r$, then $f(X') = y^*$.*

*Proof.* Let $t^* = \operatorname{argmin}_{t \in \mathbb{T}} ||(t \circ X') - X||_2$, define $Z = t^* \circ X'$ and assume that $||Z - X||_2 < r$. By definition of $\mathbb{B}$, we have $Z \in \mathbb{B}$. Because $\mathbb{T}$ is a group, there must be an inverse element $(t^*)^{-1} \in \mathbb{T}$ with $(t^*)^{-1} \circ Z = X'$. Thus, by definition of orbits (see Definition 1), we have $X' \in [Z]_{\mathbb{T}}$. It follows from Theorem 2 that $f(X') = y^*$. $\qquad\square$

In the following, we discuss how to solve the optimization problem $\min_{\tau \in \mathbb{T}} ||\tau(X') - X||_2 < r$ for invariance under different groups $\mathbb{T}$. Before proceeding, remember that solving this optimization problem is not necessary for certifying robustness, i.e. specifying a set of inputs $\tilde{\mathbb{B}}$ such that $\forall X' \in \tilde{\mathbb{B}} : f(X') = y^*$ (see Theorem 2). It is only a way of performing membership inference, i.e. determining whether $X' \in \tilde{\mathbb{B}}$ for a specific $X' \in \mathbb{R}^{N \times D}$ – just like computing the Frobenius distance between $X'$ and $X$ can be used to determine whether $X'$ is part of the original certified region $\mathbb{B}$.

### E.2.1   Translation invariance

By definition of the translation group $T(D)$ and the associated group action (see Appendix C.1), we have $t \circ X' = X' + \mathbf{1}_N b^T$ for some translation vector $b \in \mathbb{R}^D$. Thus,

$$\min_{\tau \in \mathbb{T}} ||(\tau \circ X') - X||_2 = \min_{b \in \mathbb{R}^D} ||(X' + \mathbf{1}_N b^T) - X||_2 = ||\Delta - \mathbf{1}_N \overline{\Delta}||_2,$$

where $\Delta = X' - X$ and $\overline{\Delta} \in \mathbb{R}^{1 \times D}$ are the column-wise averages. The second equality can be shown by computing the gradients w.r.t. $b$ and setting them to zero.

### E.2.2   Rotation invariance

By definition of the rotation group $SO(D)$ and the associated group action (see Appendix C.2), we have $t \circ X' = X' R^T$ for some rotation matrix $R$. Thus,

$$\min_{\tau \in \mathbb{T}} ||(\tau \circ X') - X||_2 = \min_{R \in SO(D)} ||X' R^T - X||_2.$$

This is a special case of the orthogonal Procrustes problem, which can be solved via singular value decomposition [111]. The optimal rotation matrix $R^*$ is given by $R^* = V \hat{S} U^T$, where $U S V = (X')^T X$ and $\hat{S} = \operatorname{diag}(1, \ldots, 1, \operatorname{sign}(\det(V U^T)))$.

### E.2.3 Simultaneous rotation and reflection invariance

By definition of the orthogonal group $O(D)$ and the associated group action (see Appendix C.3), we have $t \circ \boldsymbol{X}' = \boldsymbol{X}'\boldsymbol{A}^T$ for some orthogonal matrix $\boldsymbol{A}$. Thus,

$$\min_{\tau \in \mathbb{T}} ||(\tau \circ \boldsymbol{A}') - \boldsymbol{X}||_2 = \min_{\boldsymbol{A} \in O(D)} ||\boldsymbol{X}'\boldsymbol{A}^T - \boldsymbol{X}||_2.$$

This is the orthogonal Procrustes problem, which can again be solved via singular value decomposition [111]. The optimal orthogonal matrix $\boldsymbol{A}^*$ is given by $\boldsymbol{A}^* = \boldsymbol{V}\boldsymbol{U}^T$, where $\boldsymbol{U}\boldsymbol{S}\boldsymbol{V} = (\boldsymbol{X}')^T\boldsymbol{X}$. Here, accounting for the sign of the determinant is not necessary.

### E.2.4 Simultaneous rotation and translation invariance

By definition of the special Euclidean group $SE(D)$ and the associated group action (see Appendix C.4), we have $t \circ \boldsymbol{X}' = \boldsymbol{X}'\boldsymbol{R}^T + \boldsymbol{1}_N \boldsymbol{b}^T$ for some rotation matrix $\boldsymbol{R}$ and translation vector $\boldsymbol{b} \in \mathbb{R}^D$. Thus,

$$\min_{\tau \in \mathbb{T}} ||(\tau \circ \boldsymbol{X}') - \boldsymbol{X}||_2 = \min_{\boldsymbol{R} \in O(D), \boldsymbol{c} \in \mathbb{R}^D} ||\boldsymbol{X}'\boldsymbol{R}^T + \boldsymbol{1}_N \boldsymbol{c}^T - \boldsymbol{X}||_2.$$

It can be shown that this is equivalent to centering $\boldsymbol{X}'$ and $\boldsymbol{X}$ and then solving the orthogonal Procrustes problem [112], i.e.

$$\min_{\tau \in \mathbb{T}} ||\tau(\boldsymbol{X}') - \boldsymbol{X}||_2 = \min_{\boldsymbol{R} \in SO(D)} || (\boldsymbol{X}' - \boldsymbol{1}_N\overline{\boldsymbol{X}'}) \boldsymbol{R}^T - (\boldsymbol{X} - \boldsymbol{1}_N\overline{\boldsymbol{X}}) ||_2.$$

As discussed in Appendix E.2.2, this problem can be solved via singular value decomposition.

### E.2.5 Permutation invariance

By definition of the permutation group $S(N)$ and the associated group action (see Appendix C.6), we have $t \circ \boldsymbol{X}' = \boldsymbol{P}\boldsymbol{X}'$ for some permutation matrix $\boldsymbol{P}$. Thus,

$$\min_{\tau \in \mathbb{T}} ||(\tau \circ \boldsymbol{X}') - \boldsymbol{X}||_2 = \min_{\boldsymbol{P} \in S(N)} ||\boldsymbol{P}\boldsymbol{X}' - \boldsymbol{X}||_2 = \sqrt{\min_{\boldsymbol{P} \in S(N)} ||\boldsymbol{P}\boldsymbol{X}' - \boldsymbol{X}||_2^2}.$$

The inner optimization problem is equivalent to finding an optimal matching in a bipartite graph with $2 \cdot N$ nodes whose cost matrix is given by $C_{n,m} = ||\boldsymbol{X}'_n - \boldsymbol{X}_m||_2^2$. This problem can be solved in polynomial time, for example via the Hungarian algorithm [134].

### E.2.6 Simultaneous permutation, rotation and translation invariance

By definition of the permutation group $S(N)$ and the special Euclidean group $SE(N)$, we have

$$\min_{\tau \in \mathbb{T}} ||(\tau \circ \boldsymbol{X}') - \boldsymbol{X}||_2 = \min_{\boldsymbol{P} \in S(N)\boldsymbol{R} \in O(D), \boldsymbol{c} \in \mathbb{R}^D} ||\boldsymbol{P}(\boldsymbol{X}'\boldsymbol{R}^T + \boldsymbol{1}_N \boldsymbol{c}^T) - \boldsymbol{X}||_2.$$

Different from the previously discussed problems, the above is a challenging optimization problem known as point cloud registration, which does not have an efficiently computable solution. Approximate solutions to this problem are being actively studied (for a comprehensive survey, see [135]). Note that, if some upper bound $\hat{m}$ with $\min_{\tau \in \mathbb{T}} ||(\tau \circ \boldsymbol{X}') - \boldsymbol{X}||_2 < \hat{m}$ fulfills $\hat{m} < r$, then $\min_{\tau \in \mathbb{T}} ||(\tau \circ \boldsymbol{X}') - \boldsymbol{X}||_2 < r$. Thus, any approximate solution to the point cloud registration problem can be used for certification. If $\hat{m} < r$, we provably know that $\boldsymbol{X}' \in \tilde{\mathbb{B}}$ and thus $f(\boldsymbol{X}') = y^*$. However, there may be some $\boldsymbol{X}' \in \tilde{\mathbb{B}}$ with $\hat{m} > r$, which would be incorrectly declared as potential adversarial examples. Thus, an explicit characterization based on approximate solutions to the point cloud registration problem is sound, but not optimal.

# F    Tight gray-box certificates

In the following, we first prove Lemma 1, which lets us reduce the invariance-constrained optimization problem from Section 6 to a problem that is only constrained by the classifier's clean prediction probability. Then, we restate the Neyman-Pearson lemma from statistical testing, which provides an exact solution for the worst-case classifier, given its clean prediction probability. After that, we use the two lemmeta in proving Theorem 3, which summarizes tight certificates for translation, rotation and roto-translation invariance. Next, we discuss the certificates for each type of invariance in more detail and derive the results presented in Sections 6.2 to 6.5 from Theorem 3. We conclude by showing how to use Monte Carlo sampling to obtain narrow probabilistic bounds on the certificates involving rotation invariance (Appendix F.5).

## F.1    Proof of Lemma 1

Recall from Section 6 that, in order to obtain tight gray-box certificates, we need to find the worst-case invariant classifier. That is, we need to solve $\min_{h \in \mathbb{H}_\mathbb{T}} \Pr_{\boldsymbol{Z} \sim \mu_{\boldsymbol{X}'}} [h(\boldsymbol{Z}) = y^*]$, where $\mathbb{H}_\mathbb{T}$ is the set of all classifiers that are at least as likely as base classifier $g$ to predict class $p_{\boldsymbol{X}, y^*}$ and have the same invariances. We want to prove that the invariance constraint can be eliminated via a canonical map $\gamma(\boldsymbol{Z})$, which maps each input $\boldsymbol{Z} \in \mathbb{R}^{N \times D}$ to a distinct representative of its orbit $[\boldsymbol{Z}]_\mathbb{T}$.

**Lemma 1.** *Let $g : \mathbb{R}^{N \times D} \to \mathbb{Y}$ be invariant under group $\mathbb{T}$ and let $\mathbb{H}_\mathbb{T}$ be defined as in Eq. (2). If $\gamma : \mathbb{R}^{N \times D} \to \mathbb{R}^{N \times D}$ is a canonical map for invariance under $\mathbb{T}$, then*

$$\min_{h \in \mathbb{H}_\mathbb{T}} \Pr_{\boldsymbol{Z} \sim \mu_{\boldsymbol{X}'}} [h(\boldsymbol{Z}) = y^*] = \min_{h : \mathbb{R}^{N \times D} \to \mathbb{Y}} \Pr_{\boldsymbol{Z} \sim \mu_{\boldsymbol{X}'}} [h(\gamma(\boldsymbol{Z})) = y^*] \; s.t. \; \Pr_{\boldsymbol{Z} \sim \mu_{\boldsymbol{X}}} [h(\gamma(\boldsymbol{Z})) = y^*] \geq p_{\boldsymbol{X}, y^*}.$$

For the proof, recall

**Definition 1.** *The orbit of an input $\boldsymbol{X} \in \mathbb{R}^{N \times D}$ w.r.t. a group $\mathbb{T}$ is $[\boldsymbol{X}]_\mathbb{T} = \{ t \circ \boldsymbol{X} \mid t \in \mathbb{T} \}$.*

and

**Definition 3.** *A canonical map for invariance under a group of transformations $\mathbb{T}$ is a function $\gamma : \mathbb{R}^{N \times D} \to \mathbb{R}^{N \times D}$ with*

$$\forall \boldsymbol{Z} \in \mathbb{R}^{N \times D} : \gamma(\boldsymbol{Z}) \in [\boldsymbol{Z}]_\mathbb{T}, \tag{3}$$

$$\forall \boldsymbol{Z} \in \mathbb{R}^{N \times D}, \forall \boldsymbol{Z}' \in [\boldsymbol{Z}]_\mathbb{T} : \gamma(\boldsymbol{Z}) = \gamma(\boldsymbol{Z}'). \tag{4}$$

and

$$\mathbb{H}_\mathbb{T} = \left\{ h : \mathbb{R}^{N \times D} \to \mathbb{Y} \mid \Pr_{\boldsymbol{Z} \sim \mu_{\boldsymbol{X}}} [h(\boldsymbol{Z}) = y] \geq p_{\boldsymbol{X}, y^*} \wedge \forall \boldsymbol{Z}, \forall \boldsymbol{Z}' \in [\boldsymbol{Z}]_\mathbb{T} : h(\boldsymbol{Z}) = h(\boldsymbol{Z}') \right\}.$$

We begin our proof by deriving two lemmeta.

The first lemma states that, if an input $\boldsymbol{Z}$ is not the representative of its own orbit, then it cannot be the representative of any orbit.

**Lemma 3.** *Let $\gamma$ be a canonical map for invariance under $\mathbb{T}$. Then, for all $\boldsymbol{Z} \in \mathbb{R}^{N \times D}$*

$$\boldsymbol{Z} \neq \gamma(\boldsymbol{Z}) \implies \nexists \boldsymbol{Z}' \in \mathbb{R}^{N \times D} : \gamma(\boldsymbol{Z}') = \boldsymbol{Z}.$$

*Proof.* Proof by contraposition. Assume there was some $\boldsymbol{Z}'$ with $\gamma(\boldsymbol{Z}') = \boldsymbol{Z}$. Eq. (3) from Definition 1 would imply that $\boldsymbol{Z} \in [\boldsymbol{Z}']_\mathbb{T}$. Eq. (4) from Definition 1 would then imply that $\gamma(\boldsymbol{Z}') = \gamma(\boldsymbol{Z})$. By the transitive property, we would have $\boldsymbol{Z} = \gamma(\boldsymbol{Z}') = \gamma(\boldsymbol{Z})$. □

The second lemma allows us to replace all equality constraints $\forall \boldsymbol{Z}' \in [\boldsymbol{Z}]_\mathbb{T} : h(\boldsymbol{Z}) = h(\boldsymbol{Z}')$ involving input $\boldsymbol{Z}$ and element of its equivalence class with a single equality constraint $h(\boldsymbol{Z}) = h(\gamma(\boldsymbol{Z}))$.

**Lemma 4.** *Let $\gamma$ be a canonical map for invariance under $\mathbb{T}$. Then*

$$\forall \boldsymbol{Z}, \forall \boldsymbol{Z}' \in [\boldsymbol{Z}]_\mathbb{T} : h(\boldsymbol{Z}) = h(\boldsymbol{Z}') \iff \forall \boldsymbol{Z} : h(\boldsymbol{Z}) = h(\gamma(\boldsymbol{Z})).$$

*Proof.*

$\Rightarrow$ Consider an arbitrary $\boldsymbol{Z} \in \mathbb{R}^{N \times D}$. Due to Eq. (3) from Definition 1, we know that $\gamma(\boldsymbol{Z}) \in [\boldsymbol{Z}]_{\mathbb{T}}$. Therefore, $\forall \boldsymbol{Z}' \in [\boldsymbol{Z}]_{\mathbb{T}} : h(\boldsymbol{Z}) = h(\boldsymbol{Z}')$ implies $h(\boldsymbol{Z}) = h(\gamma(\boldsymbol{Z}))$.

$\Leftarrow$ Assume that $\forall \boldsymbol{Z} : h(\boldsymbol{Z}) = h(\gamma(\boldsymbol{Z}))$. Consider an arbitrary pair of inputs $\boldsymbol{Z} \in \mathbb{R}^{N \times D}, \boldsymbol{Z}' \in [\boldsymbol{Z}]_{\mathbb{T}}$. Due to our assumption, we know that $h(\boldsymbol{Z}) = h(\gamma(\boldsymbol{Z}))$. Due to Eq. (4) from Definition 3, we know that $\gamma(\boldsymbol{Z}) = \gamma(\boldsymbol{Z}')$ and thus $h(\gamma(\boldsymbol{Z})) = h(\gamma(\boldsymbol{Z}'))$. Again, due to our assumption, we know that $h(\gamma(\boldsymbol{Z}')) = h(\boldsymbol{Z}')$. By the transitive property, we have $h(\boldsymbol{Z}) = h(\boldsymbol{Z}')$. $\qquad\square$

We can now apply the two lemmata to prove Lemma 1. Lemma 4 allows us to restate $\mathbb{H}_{\mathbb{T}}$ as follows:

$$\mathbb{H}_{\mathbb{T}} = \left\{ h : \mathbb{R}^{N \times D} \to \mathbb{Y} \mid \Pr_{\boldsymbol{Z} \sim \mu_{\boldsymbol{X}}} [h(\boldsymbol{Z}) = y] \geq p_{\boldsymbol{X}, y^*} \wedge \forall \boldsymbol{Z} : h(\boldsymbol{Z}) = h(\gamma(\boldsymbol{Z})) \right\}.$$

Thus, the optimization problem $\min_{h \in \mathbb{H}_{\mathbb{T}}} \Pr_{\boldsymbol{Z} \sim \mu_{\boldsymbol{X}'}} [h(\boldsymbol{Z}) = y^*]$ can be written as

$$\min_{h : \mathbb{R}^{N \times D} \to \mathbb{Y}} \Pr_{\boldsymbol{Z} \sim \mu_{\boldsymbol{X}'}} [h(\boldsymbol{Z}) = y^*]$$
$$\text{s.t.} \quad \Pr_{\boldsymbol{Z} \sim \mu_{\boldsymbol{X}}} [h(\boldsymbol{Z}) = y^*] \geq p_{\boldsymbol{X}, y^*} \wedge \forall \boldsymbol{Z} : h(\boldsymbol{Z}) = h(\gamma(\boldsymbol{Z})).$$

Since we already know from the second constraint that any feasible solution must fulfill $h(\boldsymbol{Z}) = h(\gamma(\boldsymbol{Z}))$, we may as well substitute $h(\gamma(\boldsymbol{Z}))$ for $h(\boldsymbol{Z})$ before solving the problem:

$$\min_{h : \mathbb{R}^{N \times D} \to \mathbb{Y}} \Pr_{\boldsymbol{Z} \sim \mu_{\boldsymbol{X}'}} [h(\gamma(\boldsymbol{Z})) = y^*]$$
$$\text{s.t.} \quad \Pr_{\boldsymbol{Z} \sim \mu_{\boldsymbol{X}}} [h(\gamma(\boldsymbol{Z})) = y^*] \geq p_{\boldsymbol{X}, y^*} \wedge \forall \boldsymbol{Z} : h(\boldsymbol{Z}) = h(\gamma(\boldsymbol{Z})). \tag{13}$$

For each $\boldsymbol{Z} \in \mathbb{R}^{N \times D}$, we can now distinguish two cases: If $\boldsymbol{Z} = \gamma(\boldsymbol{Z})$, then the constraint $h(\boldsymbol{Z}) = h(\gamma(\boldsymbol{Z}))$ is trivially fulfilled and can thus be dropped. If $\boldsymbol{Z} \neq \gamma(\boldsymbol{Z})$, then Lemma 3 shows that $h(\boldsymbol{Z})$ does not appear in the objective function or first constraint of Eq. (13), because there is no other $\boldsymbol{Z}'$ with $\gamma(\boldsymbol{Z}') = \boldsymbol{Z}$. We can thus ignore the second constraint, solve the optimization problem

$$\min_{h : \mathbb{R}^{N \times D} \to \mathbb{Y}} \Pr_{\boldsymbol{Z} \sim \mu_{\boldsymbol{X}'}} [h(\gamma(\boldsymbol{Z})) = y^*] \quad \text{s.t.} \quad \Pr_{\boldsymbol{Z} \sim \mu_{\boldsymbol{X}}} [h(\gamma(\boldsymbol{Z})) = y^*] \geq p_{\boldsymbol{X}, y^*}$$

and then let $h(\boldsymbol{Z}) \leftarrow h(\gamma(\boldsymbol{Z}))$ to obtain an optimal, feasible solution to Eq. (13).

### F.2 Neyman-Pearson Lemma

For our purposes, the Neyman-Pearson lemma [107] can be formulated as follows:

**Lemma 5** (Neyman-Pearson lower bound). *Let $\mu_{\boldsymbol{X}'}$, $\mu_{\boldsymbol{X}}$, be two continuous distributions over a measurable set $\mathbb{A}$ such that, for all $\kappa \in \mathbb{R}_+$, the set $\left\{ z \mid \frac{\mu_{\boldsymbol{X}'}(z)}{\mu_{\boldsymbol{X}}(z)} = \kappa \right\}$ has measure zero. Consider an arbitrary label set $\mathbb{Y}$, a specific class label $y \in \mathbb{Y}$ and scalar $p \in [0, 1]$. Then*

$$\left( \min_{h : \mathbb{A} \to \mathbb{Y}} \Pr_{z \sim \mu_{\boldsymbol{X}'}} [h(z) = y] \text{ s.t. } \Pr_{z \sim \mu_{\boldsymbol{X}}} [h(z) = y] \geq p \right) = \mathop{\mathbf{E}}_{z \sim \mu_{\boldsymbol{X}'}} [h^*(z)]$$
$$\text{with } h^*(z) = \mathbb{1} \left[ \frac{\mu_{\boldsymbol{X}'}(z)}{\mu_{\boldsymbol{X}}(z)} \leq \kappa \right] \text{ and } \kappa \in \mathbb{R}_+ \text{ such that } \mathop{\mathbf{E}}_{z \sim \mu_{\boldsymbol{X}}} [h^*(z)] = p.$$

Here, indicator function $h^*$ corresponds to a classifier that predicts class $y$ if and only if the likelihood ratio is below a specific threshold $\kappa$. For an application of this variant of the Neyman-Pearson lemma to black-box robustness certification, as well as a discussion of its relation to most powerful hypothesis tests, see [26]. For various other formulations of the lemma, see [136].

One can use the same approach to obtain an upper bound on the probability of predicting a specific class. This will be relevant for our discussion of multi-class certificates in Appendix H.

**Lemma 6** (Neyman-Pearson upper bound). *Let $\mu_{\boldsymbol{X}'}$, $\mu_{\boldsymbol{X}}$, be two continuous distributions over a measurable set $\mathbb{A}$ such that, for all $\kappa \in \mathbb{R}_+$, the set $\left\{ z \mid \frac{\mu_{\boldsymbol{X}'}(z)}{\mu_{\boldsymbol{X}}(z)} = \kappa \right\}$ has measure zero. Consider an arbitrary label set $\mathbb{Y}$, a specific class label $y \in \mathbb{Y}$ and scalar $p \in [0, 1]$. Then*

$$\left( \max_{h : \mathbb{A} \to \mathbb{Y}} \Pr_{z \sim \mu_{\boldsymbol{X}'}} [h(z) = y] \text{ s.t. } \Pr_{z \sim \mu_{\boldsymbol{X}}} [h(z) = y] \leq p \right) = \mathop{\mathbf{E}}_{z \sim \mu_{\boldsymbol{X}'}} [h^*(z)]$$
$$\text{with } h^*(z) = \mathbb{1} \left[ \frac{\mu_{\boldsymbol{X}'}(z)}{\mu_{\boldsymbol{X}}(z)} \geq \kappa \right] \text{ and } \kappa \in \mathbb{R}_+ \text{ such that } \mathop{\mathbf{E}}_{z \sim \mu_{\boldsymbol{X}}} [h^*(z)] = p.$$

### F.3 Proof of Theorem 3

**Theorem 3.** *Let $g : \mathbb{R}^{N \times D} \to \mathbb{Y}$ be invariant under $\mathbb{T}$ with $\mathbb{T}$ chosen from $\{T(D), SO(D), SE(D)\}$. For $SO(D)$ and $SE(D)$, let $D \in \{2, 3\}$. Let $\mathbb{H}_\mathbb{T}$ be defined as in Eq. (2) and $\eta$ be a right Haar measure on $\mathbb{T}$. Define the indicator function $h^* : \mathbb{R}^{N \times D} \to \{0, 1\}$ with*

$$h^*(\boldsymbol{Z}) = \mathbb{1}\left[\frac{\beta_{\boldsymbol{X}'}(\boldsymbol{Z})}{\beta_{\boldsymbol{X}}(\boldsymbol{Z})} \leq \kappa\right], \text{ where } \beta_{\boldsymbol{X}}(\boldsymbol{Z}) = \int_{t \in \mathbb{T}} \exp\left(\langle t \circ \boldsymbol{Z}, \boldsymbol{X}\rangle_{\mathrm{F}} / \sigma^2\right) \, d\eta(t) \tag{5}$$

$$\text{and } \kappa \in \mathbb{R} \text{ such that } \mathop{\mathbf{E}}_{\boldsymbol{Z} \sim \mu_{\boldsymbol{X}}}[h^*(\boldsymbol{Z})] = p_{\boldsymbol{X}, y^*}. \tag{6}$$

*Then*

$$\min_{h \in \mathbb{H}_\mathbb{T}} \mathop{\mathrm{Pr}}_{\boldsymbol{Z} \sim \mu_{\boldsymbol{X}'}}[h(\boldsymbol{Z}) = y^*] = \mathop{\mathbf{E}}_{\boldsymbol{Z} \sim \mu_{\boldsymbol{X}'}}[h^*(\boldsymbol{Z})]. \tag{7}$$

We begin our proof by applying Lemma 1, which lets us restate the l.h.s. optimization problem from Eq. (7) as

$$\min_{h : \mathbb{R}^{N \times D} \to \mathbb{Y}} \mathop{\mathrm{Pr}}_{\boldsymbol{Z} \sim \mu_{\boldsymbol{X}'}}[h(\gamma(\boldsymbol{Z})) = y^*] \text{ s.t. } \mathop{\mathrm{Pr}}_{\boldsymbol{Z} \sim \mu_{\boldsymbol{X}}}[h(\gamma(\boldsymbol{Z})) = y^*] \geq p_{\boldsymbol{X}, y^*} \tag{14}$$

with canonical map $\gamma : \mathbb{R}^{N \times D} \to \mathbb{R}^{N \times D}$. Note that Eq. (14) has the same form as the optimization problem solved by the Neyman Pearson lemma (Lemma 5), save for the canonical representation in the probability terms.

Our goal is to 1.) specify a canonical map $\gamma$ for invariance w.r.t. group $\mathbb{T}$ 2.) bring the two probability terms from Eq. (14) into a form that does not depend $\gamma$, so that we can solve Eq. (14) exactly via the Neyman-Pearson lemma. To this end, we make the following proposition, which we shall later verify for the different considered groups $\mathbb{T}$:

**Proposition 1.** *Let $\mathbb{T} \in \{T(D), SO(D), SE(D)\}$. For $SO(D)$ and $SE(D)$, let $D \in \{2, 3\}$. Let $e \in \mathbb{T}$ be the group's identity element and $\circ : \mathbb{T} \times \mathbb{R}^{N \times D} \to \mathbb{R}^{N \times D}$ be the group action (see Appendix C). There exist a parameter space $\Omega \subseteq \mathbb{R}^K$, a function $\tau : \Omega \to \mathbb{T}$ with $\tau(\boldsymbol{0}) = e$, as well as a parameter space $\Psi \subseteq \mathbb{R}^{N \cdot D - K}$ and a function $\xi : \psi \to \mathbb{R}^{N \times D}$ with $K \in \{1, \ldots, N \cdot D - 1\}$ such that*

$$\lambda(\boldsymbol{\omega}, \boldsymbol{\psi}) = \tau(\boldsymbol{\omega}) \circ \xi(\boldsymbol{\psi}). \tag{15}$$

*is a differentiable, surjective function from $\Omega \times \Psi$ to $\mathbb{R}^{N \times D}$, injective almost everywhere and fulfills*

$$\forall t \in \mathbb{T}, \boldsymbol{\omega} \in \Omega, \boldsymbol{\psi} \in \Psi, \exists \boldsymbol{\omega}' \in \Omega : \quad t \circ \lambda(\boldsymbol{\omega}, \boldsymbol{\psi}) = \lambda(\boldsymbol{\omega}', \boldsymbol{\psi}). \tag{16}$$

That is, the input space $\mathbb{R}^{N \times D}$ can be parameterized by a matrix $\xi(\boldsymbol{\psi}) \in \mathbb{R}^{N \times D}$ that is transformed by a group element $\tau(\boldsymbol{\omega}) \in \mathbb{T}$. As formalized in Eq. (16), this alternative parameterization lets us neatly disentagle the effect of a group action, which is a translation and/or rotation of the coordinate system, from the (group-invariant) geometry of $\boldsymbol{Z}$, i.e. $\xi(\boldsymbol{\psi})$.

**Canonical map.** Based on Proposition 1, we can define the following canonical map $\gamma : \mathbb{R}^{N \times D} \to \mathbb{R}^{N \times D}$ for $\boldsymbol{Z} = \lambda(\boldsymbol{\omega}, \boldsymbol{\psi})$:

$$\gamma(\boldsymbol{Z}) = (\tau(\boldsymbol{\omega}))^{-1} \circ \lambda(\boldsymbol{\omega}, \boldsymbol{\psi}). \tag{17}$$

Note that $(\tau(\boldsymbol{\omega}))^{-1} \in \mathbb{T}$ is the inverse of group element $\tau(\boldsymbol{\omega}) \in \mathbb{T}$ and not the inverse of the function $\tau$. Further note that by Eq. (15), it holds for arbitrary $\boldsymbol{Z} = \lambda(\boldsymbol{\omega}, \boldsymbol{\psi})$ that

$$\gamma(\boldsymbol{Z}) = (\tau(\boldsymbol{\omega}))^{-1} \circ \tau(\boldsymbol{\omega}) \circ \xi(\boldsymbol{\psi}) = e \circ \xi(\boldsymbol{\psi}) = \tau(\boldsymbol{0}) \circ \xi(\boldsymbol{\psi}) = \lambda(\boldsymbol{0}, \boldsymbol{\psi}). \tag{18}$$

We can verify that $\gamma$ is indeed a valid canonical map by testing the two criteria from Definition 3. Because $\gamma(\boldsymbol{Z})$ applies a group action, we have $\gamma(\boldsymbol{Z}) \in [\boldsymbol{Z}]_\mathbb{T}$, where $[\boldsymbol{Z}]_\mathbb{T}$ is the orbit of $\boldsymbol{Z}$ w.r.t. $\mathbb{T}$. Due to Eq. (16), we have $\forall \boldsymbol{Z}, \forall \boldsymbol{Z} \in [\boldsymbol{Z}]_\mathbb{T} : \gamma(\boldsymbol{Z}) = \gamma(\lambda(\boldsymbol{\omega}, \boldsymbol{z})) = \lambda(\boldsymbol{0}, \boldsymbol{\psi}) = \gamma(\lambda(\boldsymbol{\omega}', \boldsymbol{\psi})) = \gamma(\boldsymbol{Z}')$.

**Substitution.** Next, we use Proposition 1 and the canonical map from Eq. (17) to bring the probabilities from our optimization problem in Eq. (14) into a form that is compatible with the Neyman-Pearson lemma. To declutter the terms, we first introduce $\hat{h}(\boldsymbol{Z})$ as a shorthand for $\mathbb{1}[h(\boldsymbol{Z}) = y^*]$:

$$\mathop{\mathrm{Pr}}_{\boldsymbol{Z} \sim \mu_{\boldsymbol{X}}}[h(\gamma(\boldsymbol{Z})) = y^*] = \mathop{\mathbf{E}}_{\boldsymbol{Z} \sim \mu_{\boldsymbol{X}}}\left[\hat{h}(\gamma(\boldsymbol{Z}))\right].$$

We then perform the substitution $\boldsymbol{Z} = \lambda(\boldsymbol{\omega}, \boldsymbol{\psi})$ with $\boldsymbol{\omega} \in \Omega \subseteq \mathbb{R}^K$ and $\boldsymbol{\psi} \in \Psi \subseteq \mathbb{R}^{N \cdot D - K}$:

$$\mathbb{E}_{\boldsymbol{Z} \sim \mu_{\boldsymbol{X}}} \left[ \hat{h}(\gamma(\boldsymbol{Z})) \right] = \int_{\mathbb{R}^{N \times D}} \hat{h}(\gamma(\boldsymbol{Z})) \cdot \mu_{\boldsymbol{X}}(\boldsymbol{Z}) \, d\boldsymbol{Z} \tag{19}$$

$$= \int_{\Psi} \int_{\Omega} \hat{h}(\gamma(\lambda(\boldsymbol{\omega}, \boldsymbol{\psi}))) \cdot \mu_{\boldsymbol{X}}(\lambda(\boldsymbol{\omega}, \boldsymbol{\psi})) \cdot |\det(\boldsymbol{J}_\lambda(\boldsymbol{\omega}, \boldsymbol{\psi}))| \, d\boldsymbol{\omega} d\boldsymbol{\psi}.$$

where $\boldsymbol{J}_\lambda$ is the Jacobian of $\lambda$ after vectorizing its domain and codomain. Next, we apply the canonical map defined in Eq. (17) to eliminate the group parameters $\boldsymbol{\omega}$ from the term involving classifier $\hat{h}$. This allows us to marginalize out $\boldsymbol{\omega}$:

$$\int_{\Psi} \int_{\Omega} \hat{h}(\gamma(\lambda(\boldsymbol{\omega}, \boldsymbol{\psi}))) \cdot \mu_{\boldsymbol{X}}(\lambda(\boldsymbol{\omega}, \boldsymbol{\psi})) \cdot |\det(\boldsymbol{J}_\lambda(\boldsymbol{\omega}, \boldsymbol{\psi}))| \, d\boldsymbol{\omega} d\boldsymbol{\psi}$$

$$= \int_{\Psi} \int_{\Omega} \hat{h}(\lambda(\boldsymbol{0}, \boldsymbol{\psi})) \cdot \mu_{\boldsymbol{X}}(\lambda(\boldsymbol{\omega}, \boldsymbol{\psi})) \cdot |\det(\boldsymbol{J}_\lambda(\boldsymbol{\omega}, \boldsymbol{\psi}))| \, d\boldsymbol{\omega} d\boldsymbol{\psi}$$

$$= \int_{\Psi} \hat{h}(\lambda(\boldsymbol{0}, \boldsymbol{\psi})) \int_{\Omega} \mu_{\boldsymbol{X}}(\lambda(\boldsymbol{\omega}, \boldsymbol{\psi})) \cdot |\det(\boldsymbol{J}_\lambda(\boldsymbol{\omega}, \boldsymbol{\psi}))| \, d\boldsymbol{\omega} d\boldsymbol{\psi}$$

$$:= \mathbb{E}_{\boldsymbol{\psi} \sim \nu_{\boldsymbol{X}}} \left[ \hat{h}(\lambda(\boldsymbol{0}, \boldsymbol{\psi})) \right] \tag{20}$$

$$= \Pr_{\boldsymbol{\psi} \sim \nu_{\boldsymbol{X}}} \left[ h(\lambda(\boldsymbol{0}, \boldsymbol{\psi})) = y^* \right],$$

where $\nu_{\boldsymbol{X}}(\boldsymbol{\psi}) = \int_{\Omega} \mu_{\boldsymbol{X}}(\lambda(\boldsymbol{\omega}, \boldsymbol{\psi})) \cdot |\det(\boldsymbol{J}_\lambda(\boldsymbol{\omega}, \boldsymbol{\psi}))| \, d\boldsymbol{\omega}$ is the marginal density of parameters $\boldsymbol{\psi}$. Finally, we can insert our result into Eq. (14) to obtain the simplified optimization problem

$$\min_{h: \mathbb{R}^{N \times D} \to \mathbb{Y}} \Pr_{\boldsymbol{\psi} \sim \nu_{\boldsymbol{X}'}} \left[ h(\lambda(\boldsymbol{0}, \boldsymbol{\psi})) = y^* \right] \text{s.t.} \Pr_{\boldsymbol{\psi} \sim \nu_{\boldsymbol{X}}} \left[ h(\lambda(\boldsymbol{0}, \boldsymbol{\psi})) = y^* \right] \geq p_{\boldsymbol{X}, y^*}. \tag{21}$$

**Applying the Neyman-Pearson lemma.** Eq. (21) is almost in the form required by the Neyman-Pearson lemma. But, unlike in Lemma 5, we optimize over a function defined on $\mathbb{R}^{N \times D}$ while only evaluating it on a subset of $\mathbb{R}^{N \times D}$, namely $\lambda(\boldsymbol{0}, \Psi)$. To resolve this mismatch, it is convenient to treat $h$ as a family of variables $(h_{\boldsymbol{Z}})$ indexed by $\mathbb{R}^{N \times D}$. As specified in Proposition 1, $\lambda$ is injective almost everywhere, meaning that each tuple $(\boldsymbol{0}, \boldsymbol{\psi})$ indexes a distinct variable $h_{\lambda(\boldsymbol{0}, \boldsymbol{\psi})}$, save for sets of measure zero, which do not influence the objective and constraint in Eq. (21). Therefore, we may equivalently optimize over a family of variables $(\tilde{h}_{\boldsymbol{\psi}})$ indexed by $\Psi$:

$$\min_{\tilde{h}: \Psi \to \mathbb{Y}} \Pr_{\boldsymbol{\psi} \sim \nu_{\boldsymbol{X}'}} \left[ \tilde{h}(\boldsymbol{\psi}) = y^* \right] \text{s.t.} \Pr_{\boldsymbol{\psi} \sim \nu_{\boldsymbol{X}}} \left[ \tilde{h}(\boldsymbol{\psi}) = y^* \right] \geq p_{\boldsymbol{X}, y^*}. \tag{22}$$

According to Lemma 5, the minimizer is a function that classifies $\boldsymbol{\psi}$ as $y^*$ iff $\tilde{h}^*(\boldsymbol{\psi}) = 1$ with

$$\tilde{h}^*(\boldsymbol{\psi}) = \mathbb{1} \left[ \frac{\nu_{\boldsymbol{X}'}(\boldsymbol{\psi})}{\nu_{\boldsymbol{X}}(\boldsymbol{\psi})} \leq \kappa \right] \text{ with } \kappa \in \mathbb{R}_+ \text{ such that } \mathbb{E}_{\boldsymbol{\psi} \sim \nu_{\boldsymbol{X}}} \left[ \tilde{h}^*(\boldsymbol{\psi}) \right] = p_{\boldsymbol{X}, y^*}.$$

Consequently, the optimum of our original problem Eq. (21) is given by any function $h^*(\boldsymbol{Z})$ with $h^*(\lambda(\boldsymbol{0}, \boldsymbol{\psi})) = \tilde{h}(\boldsymbol{\psi})$, i.e. we can use an arbitrary classifier for all parts of the domain that do not appear in the probability terms of Eq. (21). We make the following proposition for our choice of worst-case classifier $h^*(\boldsymbol{Z})$:

**Proposition 2.** *Consider marginal density $\nu_{\boldsymbol{X}}(\boldsymbol{\psi}) = \int_{\Omega} \mu_{\boldsymbol{X}}(\lambda(\boldsymbol{\omega}, \boldsymbol{\psi})) \cdot |\det(\boldsymbol{J}_\lambda(\boldsymbol{\omega}, \boldsymbol{\psi}))| \, d\boldsymbol{\omega}$. Let $\eta$ be a right Haar measure on group $\mathbb{T} \in \{T(D), SO(D), SE(D)\}$. For $SO(D)$ and $SE(D)$, let $D \in \{2, 3\}$. Define the group-averaged kernel*

$$\beta_{\boldsymbol{X}}(\boldsymbol{Z}) = \int_{t \in \mathbb{T}} \exp\left( \langle t \circ \boldsymbol{Z}, \boldsymbol{X} \rangle_{\mathrm{F}} / \sigma^2 \right) \, d\eta(t) \tag{23}$$

*Then,*

$$\frac{\nu_{\boldsymbol{X}'}(\boldsymbol{\psi})}{\nu_{\boldsymbol{X}}(\boldsymbol{\psi})} \propto \frac{\beta_{\boldsymbol{X}'}(\lambda(\boldsymbol{0}, \boldsymbol{\psi}))}{\beta_{\boldsymbol{X}}(\lambda(\boldsymbol{0}, \boldsymbol{\psi}))},$$

*where $\propto$ absorbs factors that are constant in $\boldsymbol{\psi}$.*

Assuming Proposition 2 holds, we have by transitivity of the previous equalities that

$$\min_{h \in \mathbb{H}_\mathbb{T}} \Pr_{\boldsymbol{Z} \sim \mu_{\boldsymbol{X}'}} [h(\boldsymbol{Z}) = y^*] = \mathbb{E}_{\boldsymbol{\psi} \sim \nu_{\boldsymbol{X}'}} [h^*(\lambda(\boldsymbol{0}, \boldsymbol{\psi}))] \tag{24}$$

with

$$h^*(\boldsymbol{Z}) = \mathbb{1}\left[\frac{\beta_{\boldsymbol{X}'}(\boldsymbol{Z})}{\beta_{\boldsymbol{X}}(\boldsymbol{Z})} \leq \kappa\right] \text{ and } \kappa \in \mathbb{R}_+ \text{ such that } \mathbb{E}_{\boldsymbol{\psi} \sim \nu_{\boldsymbol{X}}} [h^*(\lambda(\boldsymbol{0}, \boldsymbol{\psi}))] = p_{\boldsymbol{X}, y^*}. \tag{25}$$

**Resubstitution.** Next, we need to transform the expectations w.r.t. marginal distribution $\nu_{\boldsymbol{X}}$ over $\Psi \subseteq \mathbb{R}^{N \cdot D - K}$ back into an expectation w.r.t. our original smoothing distribution $\mu_{\boldsymbol{X}}$ over $\mathbb{R}^{N \times D}$. Applying the steps from Eq. (19) to Eq. (20) in reverse order shows that

$$\mathbb{E}_{\boldsymbol{\psi} \sim \nu_{\boldsymbol{X}}} [h^*(\lambda(\boldsymbol{0}, \boldsymbol{\psi}))] = \mathbb{E}_{\boldsymbol{Z} \sim \mu_{\boldsymbol{X}}} [h^*(\gamma(\boldsymbol{Z}))]. \tag{26}$$

**Exploiting invariance.** Finally, we use the fact that our worst-case classifier $h^*$ is invariant under group $\mathbb{T}$ to eliminate the canonical map $\gamma$ from the expectation. Recall from Eq. (17) that, for any $\boldsymbol{Z} = \lambda(\boldsymbol{\omega}, \boldsymbol{\psi})$, we defined $\gamma(\boldsymbol{Z}) = (\tau(\boldsymbol{\omega}))^{-1} \circ \boldsymbol{Z}$, i.e. the canonical map lets a group element act on $\boldsymbol{Z}$. Since $\eta$ is a right Haar measure, we have

$$\begin{aligned}
\beta_{\boldsymbol{X}}((\tau(\boldsymbol{\omega}))^{-1} \circ \boldsymbol{Z}) &= \int_{t \in \mathbb{T}} \exp\left(\langle t \circ (\tau(\boldsymbol{\omega}))^{-1} \circ \boldsymbol{Z}, \boldsymbol{X}\rangle_{\mathrm{F}} / \sigma^2\right) \, d\eta(t) \\
&= \int_{t \in \mathbb{T}} \exp\left(\langle (t \cdot (\tau(\boldsymbol{\omega}))^{-1}) \circ \boldsymbol{Z}, \boldsymbol{X}\rangle_{\mathrm{F}} / \sigma^2\right) \, d\eta(t) \\
&= \int_{t \in \mathbb{T}} \exp\left(\langle u \circ \boldsymbol{Z}, \boldsymbol{X}\rangle_{\mathrm{F}} / \sigma^2\right) \, d\eta(u) \\
&= \beta_{\boldsymbol{X}}(\boldsymbol{Z}),
\end{aligned}$$

where the second equality follows from the fact that $\circ$ is a group action The third equality holds because $\eta$ is a right Haar measure, meaning we can make the substitution $u = t \cdot (\tau(\boldsymbol{\omega}))^{-1})$ without having to change the measure. Thus, we have

$$h^*(\gamma(\boldsymbol{Z})) = \mathbb{1}\left[\frac{\beta_{\boldsymbol{X}'}(\gamma(\boldsymbol{Z}))}{\beta_{\boldsymbol{X}}(\gamma(\boldsymbol{Z}))} \leq \kappa\right] = \mathbb{1}\left[\frac{\beta_{\boldsymbol{X}'}(\boldsymbol{Z})}{\beta_{\boldsymbol{X}}(\boldsymbol{Z})} \leq \kappa\right] = h^*(\boldsymbol{Z}).$$

Combined with Eq. (26) and Eqs. (24) and (25), this proves that the optimal value of the original variance-constrained problem is

$$\min_{h \in \mathbb{H}_\mathbb{T}} \Pr_{\boldsymbol{Z} \sim \mu_{\boldsymbol{X}'}} [h(\boldsymbol{Z}) = y^*] = \mathbb{E}_{\boldsymbol{Z} \sim \mu_{\boldsymbol{X}'}} [h^*(\boldsymbol{Z})] \tag{27}$$

with

$$h^*(\boldsymbol{Z}) = \mathbb{1}\left[\frac{\beta_{\boldsymbol{X}'}(\boldsymbol{Z})}{\beta_{\boldsymbol{X}}(\boldsymbol{Z})} \leq \kappa\right] \text{ and } \kappa \in \mathbb{R}_+ \text{ such that } \mathbb{E}_{\boldsymbol{Z} \sim \mu_{\boldsymbol{X}}} [h^*(\boldsymbol{Z})] = p_{\boldsymbol{X}, y^*} = p_{\boldsymbol{X}, y^*}.$$

The last thing we need to do in order to conclude our proof is verify that Propositions 1 and 2 hold for each of the considered groups $\mathbb{T}$.

### F.3.1 Verifying Propositions 1 and 2 for translation invariance

For $\mathbb{T} = T(D)$, we define $\lambda(\boldsymbol{\omega}, \boldsymbol{\psi}) = \tau(\boldsymbol{\omega}) \circ \xi(\boldsymbol{\psi})$ with $\tau : \mathbb{R}^D \to T(D), \xi : \mathbb{R}^{(N-1)D} \to \mathbb{R}^{N \times D}$ and

$$\tau(\boldsymbol{\omega}) = \boldsymbol{\omega}$$
$$\xi(\boldsymbol{\psi}) = \begin{bmatrix} \boldsymbol{0}_D^T \\ \mathrm{vec}^{-1}(\boldsymbol{\psi}) \end{bmatrix},$$

where $\mathrm{vec}^{-1} : \mathbb{R}^{(N-1)D} \to \mathbb{R}^{(N-1) \times D}$ reshapes an input vector into a matrix. Due to the definition of group action $\circ$ from Appendix C.1, we have

$$\lambda(\boldsymbol{\omega}, \boldsymbol{\psi}) = \begin{bmatrix} \boldsymbol{0}_D^T \\ \mathrm{vec}^{-1}(\boldsymbol{\psi}) \end{bmatrix} + \boldsymbol{1}_N \boldsymbol{\omega}^T,$$

with all-ones vector $\mathbf{1}_N \in \mathbb{R}^N$. In other words: We represent each element of $\mathbb{R}^{N \times D}$ as a matrix whose first row is zero, translated by a vector $\boldsymbol{\omega}$.

The function $\lambda$ is evidently a differentiable bijection with inverse

$$\lambda^{-1}(\boldsymbol{Z}) = \left(\boldsymbol{Z}_1, \operatorname{vec}\left(\boldsymbol{Z}_{2:} - \mathbf{1}_D(\boldsymbol{Z}_1)^T\right)\right),$$

meaning it is surjective and injective. Furthermore, we have $\tau(\mathbf{0}_D) = \mathbf{0}_D$, with $\mathbf{0}_D$ being the identity element of $T(D)$. Lastly, we have for all $\boldsymbol{t} \in T(D)$, $\boldsymbol{\omega} \in \mathbb{R}^D$, $\boldsymbol{\psi} \in \mathbb{R}^{(N-1)D}$ that

$$\boldsymbol{t} \circ \lambda(\boldsymbol{\omega}, \boldsymbol{\psi}) = \lambda(\boldsymbol{t} + \boldsymbol{\omega}, \boldsymbol{\psi}).$$

Thus, all conditions from Proposition 1 are fulfilled.

To verify Proposition 2, we need to show that

$$\frac{\nu_{\boldsymbol{X}'}(\boldsymbol{\psi})}{\nu_{\boldsymbol{X}}(\boldsymbol{\psi})} \propto \frac{\beta_{\boldsymbol{X}'}(\lambda(\mathbf{0}, \boldsymbol{\psi}))}{\beta_{\boldsymbol{X}}(\lambda(\mathbf{0}, \boldsymbol{\psi}))},$$

with marginal density $\nu_{\boldsymbol{X}}(\boldsymbol{\psi}) = \int_{\mathbb{R}^D} \mu_{\boldsymbol{X}}(\lambda(\boldsymbol{\omega}, \boldsymbol{\psi})) \cdot |\det(\boldsymbol{J}_\lambda(\boldsymbol{\omega}, \boldsymbol{\psi}))| \; d\boldsymbol{\omega}$ and Haar integral $\beta_{\boldsymbol{X}}(\boldsymbol{Z}) = \int_{t \in \mathbb{R}^D} \exp\left(\langle t \circ \boldsymbol{Z}, \boldsymbol{X} \rangle_{\mathrm{F}} / \sigma^2\right) \; d\eta(t)$, where $\eta$ is an arbitrary right Haar measure on translation group $T(D) = \mathbb{R}^D$. Firstly, we see that $|\det(\boldsymbol{J}_\lambda(\boldsymbol{\omega}, \boldsymbol{\psi}))| = 1$, since we are only performing translations. Thus,

$$
\begin{aligned}
\nu_{\boldsymbol{X}}(\boldsymbol{\psi}) &= \int_{\mathbb{R}^D} \mu_{\boldsymbol{X}}(\lambda(\boldsymbol{\omega}, \boldsymbol{\psi})) \; d\boldsymbol{\omega} \\
&\propto \int_{\mathbb{R}^D} \prod_{n=1}^{N} \exp\left(-\frac{1}{2\sigma^2}(\lambda(\boldsymbol{\omega}, \boldsymbol{\psi})_n - \boldsymbol{X}_n)^T \lambda(\boldsymbol{\omega}, \boldsymbol{\psi})_n - \boldsymbol{X}_n\right) \; d\boldsymbol{\omega} \\
&\propto \int_{\mathbb{R}^D} \prod_{n=1}^{N} \exp\left(\frac{1}{\sigma^2}(\lambda(\boldsymbol{\omega}, \boldsymbol{\psi})_n)^T \lambda(\boldsymbol{\omega}, \boldsymbol{\psi})_n\right) \; d\boldsymbol{\omega} \\
&= \int_{\mathbb{R}^D} \exp\left(\langle \lambda(\boldsymbol{\omega}, \boldsymbol{\psi}), \boldsymbol{X} \rangle_{\mathrm{F}} / \sigma^2\right) \; d\boldsymbol{\omega} \\
&= \int_{\mathbb{R}^D} \exp\left(\langle \boldsymbol{\omega} \circ \lambda(\mathbf{0}_D, \boldsymbol{\psi}), \boldsymbol{X} \rangle_{\mathrm{F}} / \sigma^2\right) \; d\boldsymbol{\omega},
\end{aligned}
$$

where $\propto$ absorbs factors that are constant in $\boldsymbol{\omega}$. In the above equalities we have first inserted the definition of our isotropic matrix normal smoothing distribution, then removed constant terms, then expressed the product of exponential functions more compactly using the Frobenius inner product and finally used that, by definition, $\lambda(\boldsymbol{\omega}, \boldsymbol{\psi}) = \tau(\boldsymbol{\omega}) \circ \xi(\boldsymbol{\psi}) = \boldsymbol{\omega} \circ \lambda(\mathbf{0}_D, \boldsymbol{\psi})$. Finally, we note that the Lebesgue measure is translation-invariant, i.e. $\int_{\mathbb{R}^D} h(\boldsymbol{\omega} + \boldsymbol{c}) \; d\boldsymbol{\omega} = \int_{\mathbb{R}^D} h(\boldsymbol{\omega}) \; d\boldsymbol{\omega}$ for arbitrary functions $h$, meaning it is a Haar measure of the translation group. Since the translation group is a Lie group and thus locally compact, the Haar measure is unique up to a multiplicative constant [109, 110]. Thus, Proposition 2 holds.

### F.3.2 Verifying Propositions 1 and 2 for rotation invariance in 2D

In this section, let

$$\boldsymbol{R}(\omega) = \begin{bmatrix} \cos(\omega) & -\sin(\omega) \\ \sin(\omega) & \cos(\omega) \end{bmatrix} \tag{28}$$

be the matrix that rotates counter-clockwise by angle $\omega$. Note that $SO(2) = \{\boldsymbol{R}(\omega) \mid \omega \in [0, 2\pi]\}$. To verify our propositions for $\mathbb{T} = SO(2)$, we define $\lambda(\boldsymbol{\omega}, \boldsymbol{\psi}) = \tau(\omega) \circ \xi(\boldsymbol{\psi})$ with $\tau : [0, 2\pi] \to SO(2)$ and $\xi : \mathbb{R}_+ \times \mathbb{R}^{2(N-1)} \to \mathbb{R}^{N \times 2}$ (note that the first argument is non-negative) with

$$\tau(\omega) = \boldsymbol{R}(\omega)$$

$$\xi(\boldsymbol{\psi}) = \begin{bmatrix} \psi_1 & 0 \\ \operatorname{vec}^{-1}(\boldsymbol{\psi}_{2:}) \end{bmatrix},$$

where $\operatorname{vec}^{-1} : \mathbb{R}^{2(N-1)} \to \mathbb{R}^{(N-1) \times 2}$ reshapes an input vector into a matrix. Due to the definition of group action $\circ$ from Appendix C.2, we have

$$\lambda(\omega, \boldsymbol{\psi}) = \begin{bmatrix} \psi_1 & 0 \\ \operatorname{vec}^{-1}(\boldsymbol{\psi}_{2:}) \end{bmatrix} \boldsymbol{R}(\omega)^T$$

In other words: We represent each element of $\mathbb{R}^{N\times 2}$ as a matrix whose first row is aligned with the $x$-axis, rotated counter-clockwise by an angle $\omega$.

The function $\lambda$ is evidently surjective. Any $\boldsymbol{Z} \in \mathbb{R}^{N\times 2}$ can be represented via

$$\boldsymbol{Z} = \begin{bmatrix} ||\boldsymbol{Z}||_2 & 0 \\ \boldsymbol{Z}_{2:}\boldsymbol{R}(-\omega^*)^T \end{bmatrix} \boldsymbol{R}(\omega^*)^T$$

with $\omega^* = \arctan2(Z_{1,2}, Z_{1,1})$. It is also injective, save for the set $\{\boldsymbol{Z} \in \mathbb{R}^{N\times 2} \mid \boldsymbol{Z}_1 = \boldsymbol{0}\}$, which has measure zero. The first row $\boldsymbol{Z}_1$ uniquely defines $\psi_1 = ||\boldsymbol{Z}_1||_2$ and $\omega = \arctan2(Z_{1,2}, Z_{1,1})$, because polar coordinates for non-zero vectors are unique. The parameter vector $\boldsymbol{z}_{2:}$ must fulfill

$$\mathrm{vec}^{-1}(\boldsymbol{\psi}_{2:})\boldsymbol{R}(\omega) = \boldsymbol{Z}_{2:},$$

which has a unique solution because rotation matrix $\boldsymbol{R}(\omega)$ is invertible. In addition to surjectivity and injectivity almost everywhere, $\lambda$ is differentiable and we have $\tau(0) = \mathbf{I}_N$, with identity matrix $\mathbf{I}_N$ being the identity element of $SO(2)$. Furthermore, we have for all $\boldsymbol{R}(\alpha), \boldsymbol{\omega}, \boldsymbol{\psi}$ that

$$\boldsymbol{R}(\alpha) \circ \lambda(\omega, \boldsymbol{\psi}) = \lambda(\alpha + \omega, \boldsymbol{\psi}).$$

Thus, all conditions from Proposition 1 are fulfilled.

To verify Proposition 2, we need to show that

$$\frac{\nu_{\boldsymbol{X}'}(\boldsymbol{\psi})}{\nu_{\boldsymbol{X}}(\boldsymbol{\psi})} \propto \frac{\beta_{\boldsymbol{X}'}(\lambda(\boldsymbol{0}, \boldsymbol{\psi}))}{\beta_{\boldsymbol{X}}(\lambda(\boldsymbol{0}, \boldsymbol{\psi}))},$$

with marginal density $\nu_{\boldsymbol{X}}(\boldsymbol{\psi}) = \int_{[0,2\pi]} \mu_{\boldsymbol{X}}(\lambda(\boldsymbol{\omega}, \boldsymbol{\psi})) \cdot |\det(\boldsymbol{J}_\lambda(\boldsymbol{\omega}, \boldsymbol{\psi}))| \ d\boldsymbol{\omega}$ and Haar integral $\beta_{\boldsymbol{X}}(\boldsymbol{Z}) = \int_{\boldsymbol{R}\in SO(2)} \exp\left(\langle t \circ \boldsymbol{Z}, \boldsymbol{X}\rangle_{\mathrm{F}} / \sigma^2\right) d\eta(t)$. We begin by calculating the Jacobian

$$\boldsymbol{J}_\lambda(\omega, \boldsymbol{\psi}) = \begin{bmatrix} \frac{\partial \mathrm{vec}(\lambda)}{\partial \omega} & \frac{\partial \mathrm{vec}(\lambda)}{\partial \psi_1} & \frac{\partial \mathrm{vec}(\lambda)}{\partial \psi_2} & \cdots & \frac{\partial \mathrm{vec}(\lambda)}{\partial \psi_{2N-1}} \end{bmatrix}$$

$$= \begin{bmatrix} -\psi_1 \sin(\omega) & \cos(\omega) & \boldsymbol{0} \\ \psi_1 \cos(\omega) & \sin(\omega) & \boldsymbol{0} \\ \boldsymbol{a} & \boldsymbol{0} & \boldsymbol{B} \end{bmatrix},$$

with some vector $\boldsymbol{a} \in \mathbb{R}^{2(N-1)}$ and block-diagonal matrix $\boldsymbol{B} \in \mathbb{R}^{2(N-1)\times 2(N-1)}$ with

$$\boldsymbol{B} = \begin{bmatrix} \boldsymbol{R}(\omega) & & \boldsymbol{0} \\ & \ddots & \\ \boldsymbol{0} & & \boldsymbol{R}(\omega) \end{bmatrix}.$$

Due to the block structure, we have

$$|\det(\boldsymbol{J}_\lambda(\omega, \boldsymbol{\psi}))| = \left|\det\left(\begin{bmatrix} -\psi_1 \sin(\omega) & \cos(\omega) \\ \psi_1 \cos(\omega) & \sin(\omega) \end{bmatrix}\right)\right| \prod_{n=2}^{N} |\det(\boldsymbol{R}(\omega))| = |\psi_1|.$$

Thus, our marginal density is

$$\nu_{\boldsymbol{X}}(\boldsymbol{\psi}) = \int_{[0,2\pi]} |\psi_1| \cdot \mu_{\boldsymbol{X}}(\lambda(\omega, \boldsymbol{\psi})) \ d\boldsymbol{\omega}$$

$$\propto \int_{[0,2\pi]} \prod_{n=1}^{N} \exp\left(-\frac{1}{2\sigma^2}(\lambda(\omega, \boldsymbol{\psi})_n - \boldsymbol{X}_n)^T \lambda(\omega, \boldsymbol{\psi})_n - \boldsymbol{X}_n\right) d\omega$$

$$\propto \int_{[0,2\pi]} \prod_{n=1}^{N} \exp\left(\frac{1}{\sigma^2}(\lambda(\omega, \boldsymbol{\psi})_n)^T \lambda(\omega, \boldsymbol{\psi})_n\right) d\omega$$

$$= \int_{[0,2\pi]} \exp\left(\langle \lambda(\omega, \boldsymbol{\psi}), \boldsymbol{X}\rangle_{\mathrm{F}} / \sigma^2\right) d\omega$$

$$= \int_{[0,2\pi]} \exp\left(\langle \boldsymbol{R}(\omega) \circ \lambda(\boldsymbol{0}_D, \boldsymbol{\psi}), \boldsymbol{X}\rangle_{\mathrm{F}} / \sigma^2\right) d\omega,$$

where $\propto$ absorbs factors that are constant in $\omega$. Because the composition of two rotations corresponds to a translation of rotation angles, the translation-invariant Lebesgue measure is an invariant measure for group $SO(2)$ (in this angle-based parameterization): $\int_{[0,2\pi]} h(\boldsymbol{R}(\omega) \cdot \boldsymbol{R}(\omega')) \ d\omega = \int_{[0,2\pi]} h(\boldsymbol{R}(\omega + \omega')) \ d\omega = \int_{[0,2\pi]} h(\boldsymbol{R}(\omega)) \ d\omega$ for arbitrary functions $h$. Since $SO(2)$ is a Lie group and thus locally compact, the Haar measure is unique up to a multiplicative constant [109, 110]. Thus, Proposition 2 holds.

### F.3.3 Verifying Propositions 1 and 2 for rotation invariance in 3D

In this section, let

$$\boldsymbol{R}(\boldsymbol{\omega}) = \begin{bmatrix} \cos(\omega_1) & -\sin(\omega_1) & 0 \\ \sin(\omega_1) & \cos(\omega_1) & 0 \\ 0 & 0 & 1 \end{bmatrix} \cdot \begin{bmatrix} \cos(\omega_2) & 0 & \sin(\omega_2) \\ 0 & 1 & 0 \\ -\sin(\omega_2) & 0 & \cos(\omega_2) \end{bmatrix} \cdot \begin{bmatrix} 1 & 0 & 0 \\ 0 & \cos(\omega_3) & -\sin(\omega_3) \\ 0 & \sin(\omega_3) & \cos(\omega_3) \end{bmatrix}$$

be the matrix that performs an *intrinsic* rotation around the $z$-axis by angle $\omega_1$, followed by a rotation around the new $y$-axis by angle $\omega_2$ and then a rotation around the new $x$-axis by angle $\omega_3$.

Note that $SO(3) = \{\boldsymbol{R}(\boldsymbol{\omega}) \mid \boldsymbol{\omega} \in \Omega\}$ with $\Omega = [0, 2\pi] \times [-\frac{\pi}{2}, \frac{\pi}{2}] \times [0, 2\pi]$. To verify our propositions for $\mathbb{T} = SO(3)$, we define $\lambda(\boldsymbol{\omega}, \boldsymbol{\psi}) = \tau(\omega) \circ \xi(\boldsymbol{\psi})$ with $\tau : \Omega \to SO(3)$ and $\xi : \mathbb{R}_+ \times \mathbb{R} \times \mathbb{R}_+ \times \mathbb{R}^{3(N-2)} \to \mathbb{R}^{N \times 3}$ (note that the first and third argument are non-negative) with

$$\tau(\boldsymbol{\omega}) = \boldsymbol{R}(\boldsymbol{\omega})$$

$$\xi(\boldsymbol{\psi}) = \begin{bmatrix} \psi_1 & 0 & 0 \\ \psi_2 & \psi_3 & 0 \\ \mathrm{vec}^{-1}(\boldsymbol{\psi}_{4:}) \end{bmatrix},$$

where $\mathrm{vec}^{-1} : \mathbb{R}^{3(N-1)} \to \mathbb{R}^{(N-1) \times 3}$ reshapes an input vector into a matrix. Due to the definition of group action $\circ$ from Appendix C.2, we have

$$\lambda(\omega, \boldsymbol{\psi}) = \begin{bmatrix} \psi_1 & 0 & 0 \\ \psi_2 & \psi_3 & 0 \\ \mathrm{vec}^{-1}(\boldsymbol{\psi}_{4:}) \end{bmatrix} \boldsymbol{R}(\boldsymbol{\omega})^T$$

In other words: We represent each element of $\mathbb{R}^{N \times 3}$ as a matrix whose first row is aligned with the $x$-axis, and whose second row is in the first or second quadrant of the $x$-$y$-plane (because $\psi_3$ is non-negative) which is then intrinsically rotated by $z$-$y$-$x$ angles $\omega_1, \omega_2, \omega_3$.

The function $\lambda(\boldsymbol{\omega}, \boldsymbol{\psi})$ is injective and surjective, save for the set $\{\boldsymbol{Z} \in \mathbb{R}^{N \times 3} \mid \boldsymbol{Z}_1 = \boldsymbol{0} \vee \exists c \in \mathbb{R} : \boldsymbol{Z}_2 = c \cdot \boldsymbol{Z}_1\}$, which has measure zero. That is, for any $\boldsymbol{Z}$ outside this set, $\boldsymbol{Z} = \lambda(\boldsymbol{\omega}, \boldsymbol{\psi})$ has a unique solution. Firstly, $(\psi_1, \omega_1, \omega_2)$ are spherical coordinates of $\boldsymbol{Z}_1$, which are unique for $\boldsymbol{Z}_1 \neq \boldsymbol{0}$. Secondly, the unique angles $\omega_1$ and $\omega_2$ constrain the $x$-axis after the intrinsic rotation to be co-linear with $\boldsymbol{Z}_1$. Thus, and because we assume that $\boldsymbol{Z}_1$ and $\boldsymbol{Z}_2$ are not co-linear, there must be unique angle $\omega_3$ for rotation around the $x$-axis that ensures that $\boldsymbol{Z}_2$ is in the first or second quadrant of the new $x$-$y$-plane. Finally, with the rotation angles $\boldsymbol{\omega}$ and parameter $\psi_1$ fixed, the remaining parameter values are determined by

$$\begin{bmatrix} \psi_2 & \psi_3 & 0 \\ \mathrm{vec}^{-1}(\boldsymbol{\psi}_{4:}) \end{bmatrix} \boldsymbol{R}(\boldsymbol{\omega})^T = \boldsymbol{Z}_{2:} \iff \begin{bmatrix} \psi_2 & \psi_3 & 0 \\ \mathrm{vec}^{-1}(\boldsymbol{\psi}_{4:}) \end{bmatrix} = \boldsymbol{Z}_{2:}(\boldsymbol{R}(\boldsymbol{\omega})^{-1})^T.$$

The function is also surjective on the entirety of $\mathbb{R}^{N \times 3}$. If $\boldsymbol{Z}_1 = \boldsymbol{0}$, we have $\psi_1 = 0$ and $\omega_1, \omega_2$ can be chosen arbitrarily. If $\exists c : \boldsymbol{Z}_2 = c \cdot \boldsymbol{Z}_1$, then $\omega_3$ can be chosen arbitrarily. The remaining parameters can be chosen using the procedure described above. Finally, the function is differentiable, we have $\tau(\boldsymbol{0}) = \mathbf{I}_N$ and $\tau$ is a surjective function into $SO(3)$, meaning

$$\tau(\boldsymbol{\omega}') \circ \lambda(\boldsymbol{\omega}, \boldsymbol{\psi}) = (\boldsymbol{R}(\boldsymbol{\omega}') \cdot \boldsymbol{R}(\boldsymbol{\omega})) \circ \xi(\boldsymbol{v}) = \lambda(\boldsymbol{\omega}'', \boldsymbol{\psi})$$

for some $\boldsymbol{\omega}'' \in \Omega$. Thus, all criteria from Proposition 1 are fulfilled.

Like in previous sections, we verify Proposition 2 by first calculating the Jacobian of $\lambda$. In the following, we use the shorthands $s_i = \sin(\omega_i)$ and $c_i = \cos(\omega_i)$. We have

$$\boldsymbol{J}_\lambda(\omega, \boldsymbol{\psi}) = \begin{bmatrix} \frac{\partial \mathrm{vec}(\lambda)}{\partial \psi_1} & \frac{\partial \mathrm{vec}(\lambda)}{\partial \omega_1} & \frac{\partial \mathrm{vec}(\lambda)}{\partial \omega_2} & \frac{\partial \mathrm{vec}(\lambda)}{\partial \omega_3} & \frac{\partial \mathrm{vec}(\lambda)}{\partial \psi_2} & \cdots & \frac{\partial \mathrm{vec}(\lambda)}{\partial \psi_{(3(N-1))}} \end{bmatrix}$$

$$= \begin{bmatrix} \boldsymbol{A} & \boldsymbol{0} & \boldsymbol{0} \\ \boldsymbol{D} & \boldsymbol{B} & \boldsymbol{0} \\ \boldsymbol{E} & \boldsymbol{F} & \boldsymbol{C} \end{bmatrix},$$

with

$$\boldsymbol{A} = \begin{bmatrix} c_1 c_2 & -\psi_1 c_2 s_1 & -\psi_1 c_1 s_2 \\ c_2 s_1 & \psi_1 c_1 c_2 & -\psi_1 s_1 s_2 \\ -s_2 & 0 & -\psi_1 c_2 \end{bmatrix},$$

$$\boldsymbol{B} = \begin{bmatrix} \psi_3 \left( c_1 c_\gamma s_2 + s_1 s_\gamma \right) & c_1 c_2 & c_1 s_2 s_\gamma - c_\gamma s_1 \\ \psi_3 \left( -c_1 s_\gamma + c_\gamma s_1 s_2 \right) & c_2 s_1 & c_1 c_\gamma + s_1 s_2 s_\gamma \\ \psi_3 \left( c_2 c_\gamma \right) & -s_2 & c_2 s_\gamma, \end{bmatrix}$$

$$\boldsymbol{C} = \begin{bmatrix} \boldsymbol{R}(\boldsymbol{\omega}) & & \boldsymbol{0} \\ & \ddots & \\ \boldsymbol{0} & & \boldsymbol{R}(\boldsymbol{\omega}) \end{bmatrix}.$$

Due to the block structure and because $\omega_2 \in [-\frac{\pi}{2}, \frac{\pi}{2}]$ and $\psi_3 \in \mathbb{R}_+$, we have

$$\begin{aligned} |\det\left(\boldsymbol{J}_\lambda(\omega, \boldsymbol{\psi})\right)| &= \det\left(\boldsymbol{A}\right) \cdot \det\left(\boldsymbol{B}\right) \cdot \det\left(\boldsymbol{C}\right) \\ &= |\psi_1^2 \cos(\omega_2)| \cdot |\psi_3| \cdot |1| \\ &= \psi_1^2 \cdot \psi_3 \cdot \cos(\omega_2). \end{aligned}$$

Thus, our marginal density $\nu_{\boldsymbol{X}}(\boldsymbol{\psi})$ is

$$\begin{aligned} \nu_{\boldsymbol{X}}(\boldsymbol{\psi}) &= \int_\Omega \psi_1^2 \cdot \psi_3 \cdot \cos(\omega_2) \cdot \mu_{\boldsymbol{X}}(\lambda(\boldsymbol{\omega}, \boldsymbol{\psi})) \, d\boldsymbol{\omega} \\ &\propto \int_\Omega \cos(\omega_2) \cdot \prod_{n=1}^N \exp\left( -\frac{1}{2\sigma^2} (\lambda(\boldsymbol{\omega}, \boldsymbol{\psi})_n - \boldsymbol{X}_n)^T \lambda(\boldsymbol{\omega}, \boldsymbol{\psi})_n - \boldsymbol{X}_n \right) \, d\boldsymbol{\omega} \\ &\propto \int_\Omega \cos(\omega_2) \cdot \prod_{n=1}^N \exp\left( \frac{1}{\sigma^2} (\lambda(\boldsymbol{\omega}, \boldsymbol{\psi})_n)^T \lambda(\boldsymbol{\omega}, \boldsymbol{\psi})_n \right) \, d\boldsymbol{\omega} \\ &= \int_\Omega \cos(\omega_2) \cdot \exp\left( \langle \lambda(\boldsymbol{\omega}, \boldsymbol{\psi}), \boldsymbol{X} \rangle_{\mathrm{F}} / \sigma^2 \right) \, d\boldsymbol{\omega} \\ &= \int_\Omega \cos(\omega_2) \cdot \exp\left( \langle \boldsymbol{R}(\boldsymbol{\omega}) \circ \lambda(\boldsymbol{0}_D, \boldsymbol{\psi}), \boldsymbol{X} \rangle_{\mathrm{F}} / \sigma^2 \right) \, d\boldsymbol{\omega}. \end{aligned}$$

This is, up to a multiplicative constant, the unique Haar integral for this angle-based parameterization of $SO(D)$ (see, for instance [137, Chapter 1]).[3] Thus, Proposition 2 holds.

### F.3.4  Verifying Propositions 1 and 2 for roto-translation invariance in 2D and 3D

Finally, we prove that the propositions hold for $\mathbb{T} = SE(D)$ with $D \in \{2, 3\}$, which amounts to combining the results from the previous sections. In the following, let $\tau_{\mathrm{rot}} : \Omega_{\mathrm{rot}} \to SO(D)$ with $\tau_{\mathrm{rot}}(\boldsymbol{\omega}) = \boldsymbol{R}(\boldsymbol{\omega})$ be the the parameterization of $SO(D)$ defined in Appendix F.3.2 or Appendix F.3.3. Further let $\xi_{\mathrm{rot}} : \Psi_{\mathrm{rot}} \to \mathbb{R}^{(N-1) \times D}$ be the same function as in Appendix F.3.2 or Appendix F.3.3, but for matrices with $N - 1$ instead of $N$ rows.

We begin by defining $\tau : \Phi_{\mathrm{rot}} \times \mathbb{R}^D :\to SE(D)$ and $\xi : \Psi_{\mathrm{rot}} \to \mathbb{R}^{N \times D}$ as follows:

$$\tau(\boldsymbol{\omega}, \boldsymbol{b}) = (\tau_{\mathrm{rot}}(\boldsymbol{\omega}), \boldsymbol{b}) = (\boldsymbol{R}(\boldsymbol{\omega}), \boldsymbol{b})$$

$$\xi(\boldsymbol{\psi}) = \begin{bmatrix} \boldsymbol{0}_D^T \\ \xi(\boldsymbol{\psi}) \end{bmatrix},$$

Due to the definition of group action $\circ$ from Appendix C.4, we have

$$\lambda((\boldsymbol{\omega}, \boldsymbol{b}), \boldsymbol{\psi}) = \tau(\boldsymbol{\omega}, \boldsymbol{b}) \circ \xi(\boldsymbol{\psi}) = \begin{bmatrix} \boldsymbol{0}_D^T \\ \xi(\boldsymbol{\psi}) \end{bmatrix} \boldsymbol{R}(\boldsymbol{\omega})^T + \boldsymbol{1}_N \boldsymbol{b}^T.$$

In other words: We represent each element of $\mathbb{R}^{N \times D}$ as a matrix whose first row is zero and whose second row is aligned with the $x$-axis which is then rotated and finally translated. For $D = 3$ we

---

[3]Note that they have a factor $\sin$ instead of $\cos$, because they parameterize $SO(D)$ via $z$-$x$-$z$ Euler angles. The proof for intrinsic $z$-$y$-$x$ rotation is analogous.

additionally constrain the third row to be in the first or second quadrant of the $x$-$y$ plane (for more details, see Appendix F.3.3).

The function $\lambda((\boldsymbol{\omega}, \boldsymbol{b}), \boldsymbol{\psi})$ is surjective and injective, save for

- $\{\boldsymbol{Z} \in \mathbb{R}^{N \times 2} \mid \boldsymbol{Z}_2 - \boldsymbol{Z}_1 = 0\}$ (if $D = 2$),
- $\{\boldsymbol{Z} \in \mathbb{R}^{N \times 3} \mid \boldsymbol{Z}_2 - \boldsymbol{Z}_1 = 0 \vee \exists c \in \mathbb{R} : \boldsymbol{Z}_2 - \boldsymbol{Z}_1 = c(\boldsymbol{Z}_3 - \boldsymbol{Z}_1)\}$ (if $D = 3$),

which have measure zero. That is, given an $\boldsymbol{Z} \in \mathbb{R}^{N \times D}$ the solution to $\boldsymbol{Z} = \lambda((\boldsymbol{\omega}, \boldsymbol{b}), \boldsymbol{\psi})$ is unique. Evidently, we must have $\boldsymbol{b} = \boldsymbol{Z}_1$. With this parameter fixed, the remaining parameters can be found by solving the equation $\boldsymbol{Z}_{2:} - \mathbf{1}_{N-1}\boldsymbol{Z}_1^T = \tau_{\mathrm{rot}}(\boldsymbol{\omega}) \circ \xi_{\mathrm{rot}}(\boldsymbol{\psi})$, whose unique solution we discussed in Appendix F.3.2 and Appendix F.3.3. The function is also surjective on the entirety of $\mathbb{R}^{N \times D}$ for the reasons presented in Appendix F.3.2 and Appendix F.3.3. Further note that the function $\lambda((\boldsymbol{\omega}, \boldsymbol{b}), \boldsymbol{\psi})$ is differentiable and that $\tau(\mathbf{0}, \mathbf{0}) = (\mathbf{I}_D, \mathbf{0}_D)$, which is the identity element of $SE(D)$ (see Appendix C.4). Finally, we see that

$$
\begin{aligned}
(\boldsymbol{R}(\boldsymbol{\omega}'), \boldsymbol{b}') \circ \lambda((\boldsymbol{\omega}, \boldsymbol{b}), \boldsymbol{\psi}) &= ((\boldsymbol{R}(\boldsymbol{\omega}'), \boldsymbol{b}') \cdot (\boldsymbol{R}(\boldsymbol{\omega}), \boldsymbol{b}) \circ \xi(\boldsymbol{\psi}) \\
&= (\boldsymbol{R}(\boldsymbol{\omega}')\boldsymbol{R}(\boldsymbol{\omega}), \boldsymbol{R}(\boldsymbol{\omega})\boldsymbol{b} + \boldsymbol{b}') \circ \xi(\boldsymbol{\psi}) \\
&= \lambda((\boldsymbol{\omega}'', \boldsymbol{b}''), \boldsymbol{\psi})
\end{aligned}
$$

for some $\boldsymbol{\omega}'' \in \Omega, \boldsymbol{b}'' \in \mathbb{R}^D$. Thus, all criteria from Proposition 1 are fulfilled.

Next, we verify Proposition 2 by again showing that the marginal density $\nu_{\boldsymbol{X}}(\boldsymbol{\psi})$ is a Haar integral. The function $\lambda$ is a composition of two functions: The function $\lambda_{\mathrm{rot}}$ from the previous sections and a translation. Thus, the Jacobian determinant is the product of the two corresponding Jacobians. As discussed in Appendix F.3.1, the Jacobian determinant for translation is 1. Let $\boldsymbol{J}_{\lambda_{\mathrm{rot}}}(\boldsymbol{\omega}, \boldsymbol{\psi})$ be the Jacobian of $\lambda_{\mathrm{rot}}$. Then

$$
\begin{aligned}
\nu_{\boldsymbol{X}}(\boldsymbol{\psi}) &= \int_{\Omega_{\mathrm{rot}}} |\det(\boldsymbol{J}_{\lambda_{\mathrm{rot}}}(\boldsymbol{\omega}, \boldsymbol{\psi}))| \cdot \int_{\mathbb{R}^D} \mu_{\boldsymbol{X}}(\lambda((\boldsymbol{\omega}, \boldsymbol{b}), \boldsymbol{\psi})) \, d\boldsymbol{b} \, d\boldsymbol{\omega} \\
&\propto \int_{\Omega_{\mathrm{rot}}} |\det(\boldsymbol{J}_{\lambda_{\mathrm{rot}}}(\boldsymbol{\omega}, \boldsymbol{\psi}))| \cdot \int_{\mathbb{R}^D} \exp\left(\langle \lambda((\boldsymbol{\omega}, \boldsymbol{b}), \boldsymbol{\psi}), \boldsymbol{X} \rangle_{\mathrm{F}} / \sigma^2 \right) \, d\boldsymbol{b} d\boldsymbol{\omega} \\
&= \int_{\Omega_{\mathrm{rot}}} |\det(\boldsymbol{J}_{\lambda_{\mathrm{rot}}}(\boldsymbol{\omega}, \boldsymbol{\psi}))| \cdot \int_{\mathbb{R}^D} \exp\left(\langle (\boldsymbol{R}(\boldsymbol{\omega}), \boldsymbol{b}) \circ \lambda((\mathbf{0}_D, \mathbf{0}_D), \boldsymbol{\psi}), \boldsymbol{X} \rangle_{\mathrm{F}} / \sigma^2 \right) \, d\boldsymbol{b} \, d\boldsymbol{\omega},
\end{aligned}
$$

where $\propto$ absorbs factors that are constant in $\boldsymbol{\omega}$ and $\boldsymbol{b}$. This is a Haar integral, because the inner and outer integral are Haar integrals for $T(D)$ and $SO(D)$, respectively. Let us verify this by considering an arbitrary $(\boldsymbol{R}(\boldsymbol{\omega}'), \boldsymbol{b}') \in SE(D)$. To avoid clutter, define the shorthand $f(\boldsymbol{Z}) = \exp\left(\langle \boldsymbol{Z}, \boldsymbol{X} \rangle_{\mathrm{F}} / \sigma^2 \right)$. We have

$$
\begin{aligned}
&\int_{\Omega_{\mathrm{rot}}} |\det(\boldsymbol{J}_{\lambda_{\mathrm{rot}}}(\boldsymbol{\omega}, \boldsymbol{\psi}))| \cdot \int_{\mathbb{R}^D} f\left(((\boldsymbol{R}(\boldsymbol{\omega}), \boldsymbol{b}) \cdot (\boldsymbol{R}(\boldsymbol{\omega}'), \boldsymbol{b}')) \circ \lambda((\mathbf{0}_D, zeros_D), \boldsymbol{\psi})\right) \, d\boldsymbol{b} \, d\boldsymbol{\omega} \\
={}& \int_{\Omega_{\mathrm{rot}}} |\det(\boldsymbol{J}_{\lambda_{\mathrm{rot}}}(\boldsymbol{\omega}, \boldsymbol{\psi}))| \cdot \int_{\mathbb{R}^D} f\left(\lambda(\mathbf{0}_D, \boldsymbol{\psi})(\boldsymbol{R}(\omega)\boldsymbol{R}(\boldsymbol{\omega}')) + \mathbf{1}_N (\boldsymbol{R}(\omega)\boldsymbol{b}' + \boldsymbol{b})^T\right) \, d\boldsymbol{b} \, d\boldsymbol{\omega} \\
={}& \int_{\Omega_{\mathrm{rot}}} |\det(\boldsymbol{J}_{\lambda_{\mathrm{rot}}}(\boldsymbol{\omega}, \boldsymbol{\psi}))| \cdot \int_{\mathbb{R}^D} f\left(\lambda(\mathbf{0}_D, \boldsymbol{\psi})(\boldsymbol{R}(\omega)\boldsymbol{R}(\boldsymbol{\omega}')) + \mathbf{1}_N \boldsymbol{b}^T\right) \, d\boldsymbol{b} \, d\boldsymbol{\omega} \\
={}& \int_{\Omega_{\mathrm{rot}}} |\det(\boldsymbol{J}_{\lambda_{\mathrm{rot}}}(\boldsymbol{\omega}, \boldsymbol{\psi}))| \cdot \int_{\mathbb{R}^D} f\left(\lambda(\mathbf{0}_D, \boldsymbol{\psi})\boldsymbol{R}(\omega) + \mathbf{1}_N \boldsymbol{b}^T\right) \, d\boldsymbol{b} \, d\boldsymbol{\omega} \\
={}& \int_{\Omega_{\mathrm{rot}}} |\det(\boldsymbol{J}_{\lambda_{\mathrm{rot}}}(\boldsymbol{\omega}, \boldsymbol{\psi}))| \cdot \int_{\mathbb{R}^D} f\left((\boldsymbol{R}(\boldsymbol{\omega}), \boldsymbol{b}) \circ \lambda((\mathbf{0}_D, \mathbf{0}_D), \boldsymbol{\psi})\right) \, d\boldsymbol{b} \, d\boldsymbol{\omega}.
\end{aligned}
$$

Here, we have first applied the definition of the group action and group operator from Appendix C.4 and then used the fact that we are integrating over Haar measures for $T(D)$ and then $SO(D)$. Since $SE(D)$ is a Lie group and thus locally compact, the Haar measure is unique up to a multiplicative constant [109, 110]. This confirms that Proposition 2 holds.

## F.4 Group-specific certificates

In this section, we prove the results for specific invariances from Sections 6.2 to 6.5, i.e. translation invariance (Appendix F.4.1), rotation invariance in 2D (Appendix F.4.2 and 3D (Appendix F.4.3, as well as roto-translation invariance in 2D and 3D (Appendix F.4.4).

### F.4.1 Translation invariance

**Theorem 4.** *Let $g : \mathbb{R}^{N \times D} \to \mathbb{Y}$ be invariant under $\mathbb{T} = T(D)$ and $\mathbb{H}_{\mathbb{T}}$ be defined as in Eq. (2). Then*

$$\min_{h \in \mathbb{H}_{\mathbb{T}}} \Pr_{\mathbf{Z} \sim \mu_{\mathbf{X}'}} [h(\mathbf{Z}) = y^*] = \Phi \left( \Phi^{-1} \left( p_{\mathbf{X}, y^*} \right) - \frac{1}{\sigma} \left\| \mathbf{\Delta} - \mathbf{1}_N \overline{\mathbf{\Delta}} \right\|_2 \right),$$

*where $\overline{\mathbf{\Delta}} \in \mathbb{R}^{1 \times D}$ are the column-wise averages of $\Delta = \mathbf{X}' - \mathbf{X}$ and $\sigma$ is the standard deviation of the isotropic matrix normal smoothing distribution $\mu_{\mathbf{X}}$.*

During our proof of Theorem 3 in Appendices F.3 and F.3.1, we have shown that

$$\min_{h \in \mathbb{H}_{\mathbb{T}}} \Pr_{\mathbf{Z} \sim \mu_{\mathbf{X}'}} [h^*(\mathbf{Z}) = y^*] = \underset{\boldsymbol{\psi} \sim \nu_{\mathbf{X}'}}{\mathbf{E}} \left[ \tilde{h}^*(\boldsymbol{\psi}) \right] \tag{29}$$

with

$$\tilde{h}^*(\boldsymbol{\psi}) = \mathbb{1} \left[ \frac{\nu_{\mathbf{X}'}(\boldsymbol{\psi})}{\nu_{\mathbf{X}}(\boldsymbol{\psi})} \leq \kappa \right] \text{ and } \kappa \in \mathbb{R}_+ \text{ such that } \underset{\boldsymbol{\psi} \sim \nu_{\mathbf{X}}}{\mathbf{E}} \left[ \tilde{h}^*(\boldsymbol{\psi}) \right] = p_{\mathbf{X}, y^*} \tag{30}$$

with $\boldsymbol{\psi} \in \mathbb{R}^{(N-1) \cdot D}$ and marginal distribution

$$\nu_{\mathbf{X}}(\boldsymbol{\psi}) = \int_{\mathbb{R}^D} \mu_{\mathbf{X}} \left( \begin{bmatrix} \mathbf{0}_D^T \\ \text{vec}^{-1}(\boldsymbol{\psi}) \end{bmatrix} + \mathbf{1}_N \boldsymbol{\omega}^T \right) d\boldsymbol{\omega}$$

$$= \int_{\mathbb{R}^D} \prod_{d=1}^{D} \mathcal{N} \left( \begin{bmatrix} 0 \\ \text{vec}^{-1}(\boldsymbol{\psi})_{:,d} \end{bmatrix} + \mathbf{1}_N \boldsymbol{\omega}_d \mid \mathbf{X}_{:,d}, \sigma^2 \mathbf{I}_N \right) d\boldsymbol{\omega},$$

where we have inserted the definition of our isotropic matrix normal smoothing distribution for the second equality. Using the matrix $\mathbf{A} = \begin{bmatrix} 1 & \mathbf{0}_{N-1}^T \\ -\mathbf{1}_{N-1} & \mathbf{I}_{N-1} \end{bmatrix} \in \mathbb{R}^{N \times N}$ with inverse $\mathbf{A}^{-1} = \begin{bmatrix} 1 & \mathbf{0}_{N-1}^T \\ \mathbf{1}_{N-1} & \mathbf{I}_{N-1} \end{bmatrix}$ and all-zeros vector $\mathbf{0}_{N-1} \in \mathbb{R}^{N-1}$, all-ones vector $\mathbf{1}_{N-1} \in \mathbb{R}^{N-1}$ and identity matrix $\mathbf{I}_{N-1}$, the marginal density can equivalently be written as

$$\nu_{\mathbf{X}}(\boldsymbol{\psi}) = \int_{\mathbb{R}^D} \prod_{d=1}^{D} \mathcal{N} \left( \mathbf{A}^{-1} \begin{bmatrix} \omega_d \\ \text{vec}^{-1}(\boldsymbol{\psi})_{:,d} \end{bmatrix} \mid \mathbf{X}_{:,d}, \sigma^2 \mathbf{I}_N \right) d\boldsymbol{\omega},$$

$$= \int_{\mathbb{R}^D} \prod_{d=1}^{D} \frac{1}{|\det(\mathbf{A})|} \mathcal{N} \left( \begin{bmatrix} \omega_d \\ \text{vec}^{-1}(\boldsymbol{\psi})_{:,d} \end{bmatrix} \mid \mathbf{A} \mathbf{X}_{:,d}, \sigma^2 \mathbf{A} \mathbf{A}^T \right) d\boldsymbol{\omega},$$

$$= \int_{\mathbb{R}^D} \prod_{d=1}^{D} \mathcal{N} \left( \begin{bmatrix} \omega_d \\ \text{vec}^{-1}(\boldsymbol{\psi})_{:,d} \end{bmatrix} \mid \mathbf{A} \mathbf{X}_{:,d}, \sigma^2 \mathbf{A} \mathbf{A}^T \right) d\boldsymbol{\omega},$$

where the second equality follows from the change of variable formula for densities (see also Lemma 2) and the third equality is due to $\det(\mathbf{A}) = 1$. Evidently, we are marginalizing out the first dimension of each of the $D$ densities. For normal distributions, this is equivalent to dropping the first row of the mean as well as the first row and column of the covariance matrix, i.e.

$$\nu_{\mathbf{X}}(\boldsymbol{\psi}) = \prod_{d=1}^{D} \mathcal{N} \left( \text{vec}^{-1}(\boldsymbol{\psi})_{:,d} \mid \mathbf{A}_{2:} \mathbf{X}_{:,d}, \sigma^2 \mathbf{A}_{2:} (\mathbf{A}_{2:})^T \right)$$

$$= \mathcal{N} (\boldsymbol{\psi} \mid \mathbf{m}_{\mathbf{X}}, \mathbf{\Sigma})$$

with

$$m_X = \text{vec}\left(A_{2:}X\right),$$

$$\Sigma = \begin{bmatrix} B & 0 & \dots & 0 \\ 0 & B & \dots & 0 \\ \vdots & \vdots & \ddots & 0 \\ 0 & 0 & \dots & B \end{bmatrix}$$

and $B = \sigma^2 A_{2:}\left(A_{2:}\right)^T$.

Inserting back into Eqs. (29) and (30), we see that the optimal value of our variance-constrained optimization problem $\min_{h \in \mathbb{H}_{\mathbb{T}}} \Pr_{Z \sim \mu_{X'}}\left[h^*(Z) = y^*\right]$ equals the optimal value of the black-box certification problem for anisotropic normal smoothing distribution $\mathcal{N}\left(m_X, \Sigma\right)$. This optimal value has been derived in prior work (see [138] and Appendix A of [94]) and is

$$\Phi\left(\Phi^{-1}\left(p_{X,y^*}\right) - \sqrt{\left(m_{X'} - m_X\right)^T \Sigma^{-1}\left(m_{X'} - m_X\right)}\right)$$

We can conclude our proof by showing that

$$\sqrt{\left(m_{X'} - m_X\right)^T \Sigma^{-1}\left(m_{X'} - m_X\right)} = \frac{1}{\sigma}\left|\left|\Delta - \mathbf{1}_N \overline{\Delta}\right|\right|_2 \tag{31}$$

To this end, we need the following result:

**Lemma 7.** *Consider an arbitrary vector $x \in \mathbb{R}^D$ and let $\overline{x} = \frac{1}{D}\sum_{d=1}^D x_d$ be its average. Then*

$$x^T\left(x - \mathbf{1}_D \overline{x}\right) = \left(x - \mathbf{1}_D \overline{x}\right)^T\left(x - \mathbf{1}_D \overline{x}\right).$$

*Proof.* We substract the left-hand side from the right-hand side

$$x^T\left(x - \mathbf{1}_D \overline{x}\right) = \left(x - \mathbf{1}_D \overline{x}\right)^T\left(x - \mathbf{1}_D \overline{x}\right)$$

$$\Longleftrightarrow 0 = \left(-\mathbf{1}_D \overline{x}\right)^T\left(x - \mathbf{1}_D \overline{x}\right)$$

$$\Longleftrightarrow 0 = \overline{x}\sum_{d=1}^D \overline{x} - \overline{x}\sum_{d=1}^D x_d$$

$$\Longleftrightarrow 0 = \overline{x}D\overline{x} - \overline{x}D\overline{x},$$

where the last equality follows from the fact that $\sum_{d=1}^D x_d = D\overline{x}$. $\qquad\square$

Now, we can proceed by using the fact that the inverse of a block-diagonal matrix is also a block-diagonal matrix and inserting the definitions of $m_{X'}, m_X, \Sigma$ and $B$ to show that:

$$\left(m_{X'} - m_X\right)^T \Sigma^{-1}\left(m_{X'} - m_X\right) \tag{32}$$

$$= \left(\text{vec}\left(A_{2:}X'\right) - \text{vec}\left(A_{2:}X\right)\right)^T \Sigma^{-1}\left(\text{vec}\left(A_{2:}X'\right) - \text{vec}\left(A_{2:}X\right)\right) \tag{33}$$

$$= \sum_{d=1}^D \left(A_{2:}X'_{:,d} - A_{2:}X_{:,d}\right)^T B^{-1}\left(A_{2:}X'_{:,d} - A_{2:}X_{:,d}\right) \tag{34}$$

$$= \sum_{d=1}^D \left(A_{2:}X'_{:,d} - A_{2:}X_{:,d}\right)^T \frac{1}{\sigma^2}\left(A_{2:}\left(A_{2:}\right)^T\right)^{-1}\left(A_{2:}X'_{:,d} - A_{2:}X_{:,d}\right) \tag{35}$$

$$= \frac{1}{\sigma^2}\sum_{d=1}^D \left(A_{2:}\Delta_{:,d}\right)^T\left(A_{2:}\left(A_{2:}\right)^T\right)^{-1}\left(A_{2:}\Delta_{:,d}\right), \tag{36}$$

$$= \frac{1}{\sigma^2}\sum_{d=1}^D \left(\Delta_{:,d}\right)^T\left(A_{2:}\right)^T\left(A_{2:}\left(A_{2:}\right)^T\right)^{-1} A_{2:}\Delta_{:,d}, \tag{37}$$

where for the second to last last equality we have used that $X' = X + \Delta$.

Our next step is to compute $(\boldsymbol{A}_{2:})^T \left( \boldsymbol{A}_{2:} (\boldsymbol{A}_{2:})^T \right)^{-1} \boldsymbol{A}_{2:}$. Recall that $\boldsymbol{A}_{2:} \in \mathbb{R}^{(N-1) \times D}$ and $\boldsymbol{A}_{2:} = [-\mathbf{1}_{N-1} \quad \mathbf{I}_{N-1}]$. Matrix multiplication shows that

$$\boldsymbol{A}_{2:} (\boldsymbol{A}_{2:})^T = [-\mathbf{1}_{N-1} \quad \mathbf{I}_{N-1}] [-\mathbf{1}_{N-1} \quad \mathbf{I}_{N-1}]^T = \mathbf{1}_{N-1,N-1} + \mathbf{I}_{N-1} = \begin{bmatrix} 2 & 1 & \cdots & 1 \\ 1 & 2 & \cdots & 1 \\ \vdots & \vdots & \ddots & 1 \\ 1 & 1 & \cdots & 2 \end{bmatrix}.$$

The matrix is sufficiently simple to be inverted by inspection:

$$\left( \boldsymbol{A}_{2:} (\boldsymbol{A}_{2:})^T \right)^{-1} = \mathbf{I}_{N-1} - \frac{1}{N} \mathbf{1}_{N-1,N-1} = \begin{bmatrix} 1 - \frac{1}{N} & -\frac{1}{N} & \cdots & -\frac{1}{N} \\ -\frac{1}{N} & 1 - \frac{1}{N} & \cdots & -\frac{1}{N} \\ \vdots & \vdots & \ddots & \frac{1}{N} \\ -\frac{1}{N} & -\frac{1}{N} & \cdots & 1 - \frac{1}{N} \end{bmatrix}.$$

Finally, matrix multiplication shows that

$$(\boldsymbol{A}_{2:})^T \left( \boldsymbol{A}_{2:} (\boldsymbol{A}_{2:})^T \right)^{-1} \boldsymbol{A}_{2:} = \mathbf{I}_N - \frac{1}{N} \mathbf{1}_{N,N} = \begin{bmatrix} 1 - \frac{1}{N} & -\frac{1}{N} & \cdots & -\frac{1}{N} \\ -\frac{1}{N} & 1 - \frac{1}{N} & \cdots & -\frac{1}{N} \\ \vdots & \vdots & \ddots & \frac{1}{N} \\ -\frac{1}{N} & -\frac{1}{N} & \cdots & 1 - \frac{1}{N} \end{bmatrix}. \tag{38}$$

Note that the matrix in Eq. (38) transforms a vector by subtracting its average from all entries. Inserting into Eq. (37) shows that

$$(\boldsymbol{m}_{X'} - \boldsymbol{m}_X)^T \boldsymbol{\Sigma}^{-1} (\boldsymbol{m}_{X'} - \boldsymbol{m}_X)$$

$$= \frac{1}{\sigma^2} \sum_{d=1}^{D} (\boldsymbol{\Delta}_{:,d})^T \left( \boldsymbol{\Delta}_{:,d} - \mathbf{1}_D \overline{\boldsymbol{\Delta}_{:,d}} \right)$$

$$= \frac{1}{\sigma^2} \sum_{d=1}^{D} \left( \boldsymbol{\Delta}_{:,d} - \mathbf{1}_D \overline{\boldsymbol{\Delta}_{:,d}} \right)^T \left( \boldsymbol{\Delta}_{:,d} - \mathbf{1}_D \overline{\boldsymbol{\Delta}_{:,d}} \right)$$

$$= \frac{1}{\sigma^2} \sum_{n=1}^{N} \sum_{d=1}^{D} \left( \boldsymbol{\Delta} - \mathbf{1}_N \overline{\boldsymbol{\Delta}} \right)_{n,d}^2 ,$$

where $\overline{\boldsymbol{\Delta}_{:,d}} \in \mathbb{R}$ is the average of column $\boldsymbol{\Delta}_{:,d}$, $\overline{\boldsymbol{\Delta}} \in \mathbb{R}^{1 \times D}$ are the column-wise averages of matrix $\boldsymbol{\Delta}$, the second equality is due to Lemma 7 and the third equality uses the definition of inner products.

Taking the square root and using the definition of the Frobenius norm yields our desired result:

$$\sqrt{(\boldsymbol{m}_{X'} - \boldsymbol{m}_X)^T \boldsymbol{\Sigma}^{-1} (\boldsymbol{m}_{X'} - \boldsymbol{m}_X)} = \sqrt{\frac{1}{\sigma^2} \sum_{n=1}^{N} \sum_{d=1}^{D} \left( \boldsymbol{\Delta} - \mathbf{1}_N \overline{\boldsymbol{\Delta}} \right)_{n,d}^2} = \frac{1}{\sigma} \left\| \boldsymbol{\Delta} - \mathbf{1}_N \overline{\boldsymbol{\Delta}} \right\|_2 .$$

### F.4.2   Rotation invariance in 2D

**Theorem 6.** *Let $g : \mathbb{R}^{N \times 2} \to \mathbb{Y}$ be invariant under $\mathbb{T} = SO(2)$ and $\mathbb{H}_{\mathbb{T}}$ be defined as in Eq. (2). Define the indicator function $\rho : \mathbb{R}^4 \to \{0,1\}$ with*

$$\rho(\boldsymbol{q}) = \mathbb{1}\left[ \mathcal{I}_0 \left( \sqrt{q_1^2 + q_2^2} \right) / \mathcal{I}_0 \left( \sqrt{q_3^2 + q_4^2} \right) \leq \kappa \right],$$

$$\text{with } \kappa \in \mathbb{R} \text{ such that} \quad \underset{\boldsymbol{q} \sim \mathcal{N}\left( \boldsymbol{m}^{(2)}, \boldsymbol{\Sigma} \right)}{\mathbf{E}} [\rho(\boldsymbol{q})] = p_{\boldsymbol{X}, y^*}.$$

*Then*

$$\min_{h \in \mathbb{H}_{\mathbb{T}}} \underset{\boldsymbol{Z} \sim \mu_{\boldsymbol{X}'}}{\Pr} [h(\boldsymbol{Z}) = y^*] = \underset{\boldsymbol{q} \sim \mathcal{N}\left( \boldsymbol{m}^{(1)}, \boldsymbol{\Sigma} \right)}{\mathbf{E}} [\rho(\boldsymbol{q})],$$

*where*

$$\boldsymbol{m}^{(1)} = \frac{1}{\sigma^2} \begin{bmatrix} 2\epsilon_1 + ||\boldsymbol{X}||_2^2 + ||\boldsymbol{\Delta}||_2^2 \\ 0 \\ \epsilon_1 + ||\boldsymbol{X}||_2^2 \\ \epsilon_2 \end{bmatrix}, \quad \boldsymbol{m}^{(2)} = \frac{1}{\sigma^2} \begin{bmatrix} \epsilon_1 + ||\boldsymbol{X}||_2^2 \\ -\epsilon_2 \\ ||\boldsymbol{X}||_2^2 \\ 0 \end{bmatrix},$$

$$\boldsymbol{\Sigma} = \frac{1}{\sigma^2} \begin{bmatrix} 2\epsilon_1 + ||\boldsymbol{X}||_2^2 + ||\boldsymbol{\Delta}||_2^2 & 0 & \epsilon_1 + ||\boldsymbol{X}||_2^2 & \epsilon_2 \\ 0 & 2\epsilon_1 + ||\boldsymbol{X}||_2^2 + ||\boldsymbol{\Delta}||_2^2 & -\epsilon_2 & \epsilon_1 + ||\boldsymbol{X}||_2^2 \\ \epsilon_1 + ||\boldsymbol{X}||_2^2 & -\epsilon_2 & ||\boldsymbol{X}||_2^2 & 0 \\ \epsilon_2 & \epsilon_1 + ||\boldsymbol{X}||_2^2 & 0 & ||\boldsymbol{X}||_2^2. \end{bmatrix},$$

*with clean data norm* $||\boldsymbol{X}||_2$, *perturbation norm* $||\boldsymbol{\Delta}||_2$ *and parameters* $\epsilon_1 = \langle \boldsymbol{X}, \boldsymbol{\Delta} \rangle_{\mathrm{F}}$, $\epsilon_2 = \langle \boldsymbol{X}\boldsymbol{R}(-\pi/2)^T, \boldsymbol{\Delta} \rangle_{\mathrm{F}}$.

We know from Theorem 3 and our derivations in Appendix F.3.2 that

$$\min_{h \in \mathbb{H}_{\mathbb{T}}} \Pr_{\boldsymbol{Z} \sim \mu_{\boldsymbol{X}'}} [h(\boldsymbol{Z}) = y^*] = \mathbf{E}_{\boldsymbol{Z} \sim \mu_{\boldsymbol{X}'}} [h^*(\boldsymbol{Z})], \tag{39}$$

with

$$h^*(\boldsymbol{Z}) = \mathbb{1}\left[\frac{\beta_{\boldsymbol{X}'}(\boldsymbol{Z})}{\beta_{\boldsymbol{X}}(\boldsymbol{Z})} \leq \kappa\right] \text{ and } \kappa \in \mathbb{R}_+ \text{ such that } \mathbf{E}_{\boldsymbol{Z} \sim \mu_{\boldsymbol{X}}} [h^*(\boldsymbol{Z})] = p_{\boldsymbol{X}, y^*}, \tag{40}$$

$$\beta_{\boldsymbol{X}}(\boldsymbol{Z}) = \int_{[0,2\pi]} \exp\left(\langle \boldsymbol{Z}\boldsymbol{R}(\omega)^T, \boldsymbol{X} \rangle_{\mathrm{F}} / \sigma^2\right) d\omega,$$

$$\boldsymbol{R}(\omega) = \begin{bmatrix} \cos(\omega) & -\sin(\omega) \\ \sin(\omega) & \cos(\omega) \end{bmatrix}.$$

To prove Theorem 6, we shall first calculate the integral $\beta_{\boldsymbol{X}}(\boldsymbol{Z})$ characterizing our worst-case classifier and then use the fact that it applies an affine transformation to our random input data sampled from $\mu_{\boldsymbol{X}}$ and $\mu_{\boldsymbol{X}'}$.

We begin by factoring out $\cos(\omega)$ and $\sin(\omega)$ from the exponent:

$$\langle \boldsymbol{Z}\boldsymbol{R}(\omega)^T, \boldsymbol{X} \rangle_{\mathrm{F}} = \sum_{n=1}^{N} (\boldsymbol{R}(\omega)\boldsymbol{Z}_n)(\boldsymbol{X}_n)^T$$

$$= \sum_{n=1}^{N} \cos(\omega)\boldsymbol{Z}_{n,1}\boldsymbol{X}_{n,1} - \sin(\omega)\boldsymbol{Z}_{n,2}\boldsymbol{X}_{n,1} + \sin(\omega)\boldsymbol{Z}_{n,1}\boldsymbol{X}_{n,2} + \cos(\omega)\boldsymbol{Z}_{n,2}\boldsymbol{X}_{n,2}$$

$$= \cos(\omega)\begin{bmatrix} \boldsymbol{X}_{:,1} \\ \boldsymbol{X}_{:,2} \end{bmatrix}^T \begin{bmatrix} \boldsymbol{Z}_{:,1} \\ \boldsymbol{Z}_{:,2} \end{bmatrix} + \sin(\omega)\begin{bmatrix} \boldsymbol{X}_{:,2} \\ -\boldsymbol{X}_{:,1} \end{bmatrix}^T \begin{bmatrix} \boldsymbol{Z}_{:,1} \\ \boldsymbol{Z}_{:,2} \end{bmatrix}$$

$$= \cos(\omega) \cdot \mathrm{vec}(\boldsymbol{X})^T \mathrm{vec}(\boldsymbol{Z}) + \sin(\omega) \cdot \mathrm{vec}\left(\boldsymbol{X}\boldsymbol{R}(-\pi/2)^T\right)^T \mathrm{vec}(\boldsymbol{Z}).$$

We can then show that

$$\beta_{\boldsymbol{X}}(\boldsymbol{Z})$$
$$= \int_{[0,2\pi]} \exp\left(\frac{1}{\sigma^2}\left(\cos(\omega) \cdot \mathrm{vec}(\boldsymbol{X})^T \mathrm{vec}(\boldsymbol{Z}) + \sin(\omega) \cdot \mathrm{vec}\left(\boldsymbol{X}\boldsymbol{R}(-\pi/2)^T\right)^T \mathrm{vec}(\boldsymbol{Z})\right)\right) d\omega$$

$$= \int_{[0,2\pi]} \exp\left(\cos(\omega) \cdot \sqrt{(\mathrm{vec}(\boldsymbol{X})^T \mathrm{vec}(\boldsymbol{Z})/\sigma^2)^2 + \left(\mathrm{vec}\left(\boldsymbol{X}\boldsymbol{R}(-\pi/2)^T\right)^T \mathrm{vec}(\boldsymbol{Z})/\sigma^2\right)^2}\right) d\omega$$

$$= 2\pi \cdot \mathcal{I}_0\left(\sqrt{(\mathrm{vec}(\boldsymbol{X})^T \mathrm{vec}(\boldsymbol{Z})/\sigma^2)^2 + \left(\mathrm{vec}\left(\boldsymbol{X}\boldsymbol{R}(-\pi/2)^T\right)^T \mathrm{vec}(\boldsymbol{Z})/\sigma^2\right)^2}\right),$$

where the second equality follows from the fact that $\cos(\omega)\eta_1 + \sin(\omega)\eta_2 = \cos(\omega + \alpha)\sqrt{\eta_1^2 + \eta_2^2}$ for some $\alpha \in [0, 2\pi]$ (see also Eq. 29 of [139]) and the third equality is due to the integral representation

of the modified Bessel function of the first kind with order 0 (see Eq. 10.32.1 of [140]):

$$\mathcal{I}_0(z) = \frac{1}{\pi} \int\limits_{[0,\pi]} \exp\left(\cos(\alpha)z\right) \, d\alpha.$$

We can now insert this expression for the worst-case classifier into the expectations w.r.t. clean and perturbed smoothing distributions $\mu_{\boldsymbol{X}}, \mu'_{\boldsymbol{X}}$ from Eqs. (39) and (40) . Note that, by definition, we have $\text{vec}(\boldsymbol{Z}) \sim \mathcal{N}(\text{vec}(\boldsymbol{X}), \sigma^2 \cdot \mathbf{I}_{2N})$ for $\boldsymbol{Z} \sim \mu_{\boldsymbol{X}}$. Thus:

$$\underset{\boldsymbol{Z} \sim \mu_{\boldsymbol{X}'}}{\mathbf{E}} [h^*(\boldsymbol{Z})] = \underset{\boldsymbol{q} \sim \mathcal{N}(\boldsymbol{m}^{(1)}, \boldsymbol{\Sigma})}{\mathbf{E}} \left[ \mathbb{1}\left[\mathcal{I}_0\left(\sqrt{q_1^2 + q_2^2}\right) / \mathcal{I}_0\left(\sqrt{q_3^2 + q_4^2}\right) \leq \kappa\right] \right],$$

$$\underset{\boldsymbol{Z} \sim \mu_{\boldsymbol{X}}}{\mathbf{E}} [h^*(\boldsymbol{Z})] = \underset{\boldsymbol{q} \sim \mathcal{N}(\boldsymbol{m}^{(2)}, \boldsymbol{\Sigma})}{\mathbf{E}} \left[ \mathbb{1}\left[\mathcal{I}_0\left(\sqrt{q_1^2 + q_2^2}\right) / \mathcal{I}_0\left(\sqrt{q_3^2 + q_4^2}\right) \leq \kappa\right] \right],$$

with means $\boldsymbol{m}^{(1)} = \boldsymbol{W}\text{vec}(\boldsymbol{X}')$, $\boldsymbol{m}^{(2)} = \boldsymbol{W}\text{vec}(\boldsymbol{X})$ and covariance matrix $\boldsymbol{\Sigma} = \sigma^2 \boldsymbol{W}\boldsymbol{W}^T$, where

$$\boldsymbol{W} = \frac{1}{\sigma^2} \cdot \begin{bmatrix} \text{vec}(\boldsymbol{X}')^T \\ \text{vec}\left(\boldsymbol{X}'\boldsymbol{R}\left(-\pi/2\right)^T\right) \\ \text{vec}(\boldsymbol{X})^T \\ \text{vec}\left(\boldsymbol{X}\boldsymbol{R}\left(-\pi/2\right)^T\right) \end{bmatrix}.$$

Calculating the matrix-vector and matrix-matrix products yields the values from Theorem 6, thus proving that the certificate is valid.

### F.4.3   Rotation invariance in 3D

As discussed in Section 6.4, we can apply Theorem 3 to $\mathbb{T} = SO(3)$, but do not have a closed-form expression for the Haar integral $\beta_{\boldsymbol{X}}(\boldsymbol{Z})$ characterizing our worst-case classifier. We can however evaluate it using numerical integration. In the following, we first show how we can reduce the number of integration variables to facilitate numerical integration. We then show that, similar to the 2D case, the Monte Carlo certification procedure presented in Section 6.3 only requires sampling from a 16-dimensional normal distribution, which allows us to obtain tight bounds via a large number of samples at little computational cost.

**Reducing the number of integration variables.** Recall from our proof of Theorem 3 in Appendices F.3 and F.3.3 that

$$\beta_{\boldsymbol{X}}(\boldsymbol{Z}) = \int_\Omega \cos(\omega_2) \cdot \exp\left(\langle \boldsymbol{Z}\boldsymbol{R}(\boldsymbol{\omega})^T, \boldsymbol{X}\rangle_{\mathrm{F}} / \sigma^2\right) \, d\boldsymbol{\omega}, \tag{41}$$

with $\Omega = [0, 2\pi] \times [-\frac{\pi}{2}, \frac{\pi}{2}] \times [0, 2\pi]$ and rotation matrix

$$\begin{aligned}
\boldsymbol{R}(\boldsymbol{\omega}) &= \begin{bmatrix} \cos(\omega_1) & -\sin(\omega_1) & 0 \\ \sin(\omega_1) & \cos(\omega_1) & 0 \\ 0 & 0 & 1 \end{bmatrix} \cdot \begin{bmatrix} \cos(\omega_2) & 0 & \sin(\omega_2) \\ 0 & 1 & 0 \\ -\sin(\omega_2) & 0 & \cos(\omega_2) \end{bmatrix} \cdot \begin{bmatrix} 1 & 0 & 0 \\ 0 & \cos(\omega_3) & -\sin(\omega_3) \\ 0 & \sin(\omega_3) & \cos(\omega_3) \end{bmatrix} \\
&= \begin{bmatrix} \cos(\omega_1) & -\sin(\omega_1) & 0 \\ \sin(\omega_1) & \cos(\omega_1) & 0 \\ 0 & 0 & 1 \end{bmatrix} \cdot \begin{bmatrix} \cos(\omega_2) & \sin(\omega_2)\sin(\omega_3) & \cos(\omega_3)\sin(\omega_2) \\ 0 & \cos(\omega_3) & -\sin(\omega_3) \\ -\sin(\omega_2) & \cos(\omega_2)\sin(\omega_3) & \cos(\omega_2)\cos(\omega_3) \end{bmatrix} \\
&:= \begin{bmatrix} \cos(\omega_1) & -\sin(\omega_1) & 0 \\ \sin(\omega_1) & \cos(\omega_1) & 0 \\ 0 & 0 & 1 \end{bmatrix} \cdot \tilde{\boldsymbol{R}}(\boldsymbol{\omega}_{2:}),
\end{aligned}$$

with $\tilde{\boldsymbol{R}}(\boldsymbol{\omega}_{2:})$ being the matrix that rotates around the $y$- and $z$-axis by angles $\omega_2$ and $\omega_3$, respectively. While we cannot evaluate this term analytically, we can calculate the inner integral over $\omega_1$ in order

to reduce it to a double integral. We begin by factoring out $\cos(\omega_1)$ and $\sin(\omega_1)$ from the exponent:

$$\left\langle \boldsymbol{Z}\boldsymbol{R}(\boldsymbol{\omega})^T, \boldsymbol{X}\right\rangle_{\mathrm{F}} / \sigma^2$$

$$= \sum_{n=1}^{N} (\boldsymbol{X}_n)^T \boldsymbol{R}(\boldsymbol{\omega})\boldsymbol{Z}_n / \sigma^2$$

$$= \cos(\omega_1)\left(\sum_{n=1}^{N} \boldsymbol{X}_{n,1}\left(\tilde{\boldsymbol{R}}(\boldsymbol{\omega}_{2:})\boldsymbol{Z}_n\right)_1 + \boldsymbol{X}_{n,2}\left(\tilde{\boldsymbol{R}}(\boldsymbol{\omega}_{2:})\boldsymbol{Z}_n\right)_2\right) / \sigma^2$$

$$+ \sin(\omega_1)\left(\sum_{n=1}^{N} \boldsymbol{X}_{n,2}\left(\tilde{\boldsymbol{R}}(\boldsymbol{\omega}_{2:})\boldsymbol{Z}_n\right)_1 - \boldsymbol{X}_{n,1}\left(\tilde{\boldsymbol{R}}(\boldsymbol{\omega}_{2:})\boldsymbol{Z}_n\right)_2\right) / \sigma^2$$

$$+ \sum_{n=1}^{N} \boldsymbol{X}_{n,3}\left(\tilde{\boldsymbol{R}}(\boldsymbol{\omega}_{2:})\boldsymbol{Z}_n\right)_3 / \sigma^2.$$

$$= \cos(\omega_1)\cdot\chi_1(\boldsymbol{\omega}_{2:}, \boldsymbol{X}^T\boldsymbol{Z}) + \sin(\omega_1)\cdot\chi_2(\boldsymbol{\omega}_{2:}, \boldsymbol{X}^T\boldsymbol{Z}) + \chi_3(\boldsymbol{\omega}_{2:}, \boldsymbol{X}^T\boldsymbol{Z}),$$

where $\chi_1, \chi_2, \chi_3$ are shorthands we introduce to avoid clutter in the following derivations. Due to the entries of $\tilde{\boldsymbol{R}}(\omega_2 :)$, the shorthand $\chi_1 : \mathbb{R}^2 \times \mathbb{R}^{3\times 3} \to \mathbb{R}$ is defined as

$$\chi_1(\boldsymbol{\omega}_{2:}, \boldsymbol{X}^T\boldsymbol{Z}))$$

$$:= \frac{1}{\sigma^2}\sum_{n=1}^{N}\Big(\boldsymbol{X}_{n,1}\left(\cos(\omega_2)\boldsymbol{Z}_{n,1} + \sin(\omega_2)\sin(\omega_3)\boldsymbol{Z}_{n,2} + \cos(\omega_3)\sin(\omega_2)\boldsymbol{Z}_{n,3}\right)$$

$$+ \boldsymbol{X}_{n,2}\left(\cos(\omega_3)\boldsymbol{Z}_{n,2} - \sin(\omega_3)\boldsymbol{Z}_{n,3}\right)\Big)$$

$$= \cos(\omega_2)(\boldsymbol{X}_{:,1})^T\boldsymbol{Z}_{:,1} + \sin(\omega_2)\sin(\omega_3)(\boldsymbol{X}_{:,1})^T\boldsymbol{Z}_{:,2} + \cos(\omega_3)\sin(\omega_2)(\boldsymbol{X}_{:,1})^T\boldsymbol{Z}_{:,3}$$

$$+ \cos(\omega_3)(\boldsymbol{X}_{:,2})^T\boldsymbol{Z}_{:,2} - \sin(\omega_3)(\boldsymbol{X}_{:,2})^T\boldsymbol{Z}_{:,3}$$

$$= \cos(\omega_2)(\boldsymbol{X}^T\boldsymbol{Z})_{1,1} + \sin(\omega_2)\sin(\omega_3)(\boldsymbol{X}^T\boldsymbol{Z})_{1,2} + \cos(\omega_3)\sin(\omega_2)(\boldsymbol{X}^T\boldsymbol{Z})_{1,3}$$

$$+ \cos(\omega_3)(\boldsymbol{X}^T\boldsymbol{Z})_{2,2} - \sin(\omega_3)(\boldsymbol{X}^T\boldsymbol{Z})_{2,3}$$

Similarly, the shorthand $\chi_2 : \mathbb{R}^2 \times \mathbb{R}^{3\times 3} \to \mathbb{R}$ is defined as

$$\chi_2(\boldsymbol{\omega}_{2:,}, \boldsymbol{X}^T\boldsymbol{Z})$$

$$:= \cos(\omega_2)(\boldsymbol{X}^T\boldsymbol{Z})_{2,1} + \sin(\omega_2)\sin(\omega_3)(\boldsymbol{X}^T\boldsymbol{Z})_{2,2} + \cos(\omega_3)\sin(\omega_2)(\boldsymbol{X}^T\boldsymbol{Z})_{2,3}$$

$$- \cos(\omega_3)(\boldsymbol{X}^T\boldsymbol{Z})_{1,2} + \sin(\omega_3)(\boldsymbol{X}^T\boldsymbol{Z})_{1,3}.$$

The shorthand $\chi_3 : \mathbb{R}^2 \times \mathbb{R}^{3\times 3} \to \mathbb{R}$ is defined as

$$\chi_3(\boldsymbol{\omega}_{2:,}, \boldsymbol{X}^T\boldsymbol{Z})$$

$$:= -\sin(\omega_2)(\boldsymbol{X}^T\boldsymbol{Z})_{3,1} + \cos(\omega_2)\sin(\omega_3)(\boldsymbol{X}^T\boldsymbol{Z})_{3,2} + \cos(\omega_2)\cos(\omega_3)(\boldsymbol{X}^T\boldsymbol{Z})_{3,3}.$$

Just like in Appendix F.4.2, we can use the integral formula for the modified Bessel function of the first kind and order zero to eliminate the integral w.r.t. $\omega_1$:

$$\beta_{\boldsymbol{X}}(\boldsymbol{Z})$$

$$= \int_{\Omega} \cos(\omega_2)\cdot\exp\left(\chi_3(\boldsymbol{\omega}_{2:}, \boldsymbol{X}^T\boldsymbol{Z})\right)\cdot\exp\left(\cos(\omega_1)\cdot\chi_1(\boldsymbol{\omega}_{2:}, \boldsymbol{X}, \boldsymbol{Z}) + \sin(\omega_1)\cdot\chi_2(\boldsymbol{\omega}_{2:}, \boldsymbol{X}^T\boldsymbol{Z})\right)\,d\boldsymbol{\omega}$$

$$= 2\pi\cdot\int_{[-\frac{\pi}{2},\frac{\pi}{2}]\times[0,2\pi]} \cos(\omega_2)\cdot\exp\left(\chi_3(\boldsymbol{\omega}_{2:}, \boldsymbol{X}^T\boldsymbol{Z}))\right)\cdot\mathcal{I}_0\left(\sqrt{\chi_1(\boldsymbol{\omega}_{2:}, \boldsymbol{X}, \boldsymbol{Z})^2 + \chi_2(\boldsymbol{\omega}_{2:}, \boldsymbol{X}, \boldsymbol{Z})^2}\right)\,d\boldsymbol{\omega}_{2:}$$

$$:= \hat{\beta}(\boldsymbol{X}^T\boldsymbol{Z}).$$

We introduce $\hat{\beta}(\boldsymbol{X}^T\boldsymbol{Z})$ to avoid clutter in the next equations and to highlight that this function characterizing our worst-case classifier only depends on $\boldsymbol{X}^T\boldsymbol{Z} \in \mathbb{R}^{3\times 3}$.

**Efficient Monte Carlo certification.** Just like in Appendix F.4.2, we can now use the fact that our worst-case classifier only depends on a small number of variables that are the result of linearly

transforming our randomized inputs $\boldsymbol{Z} \sim \mu_{\boldsymbol{X}}$, namely the 9 entires of $\boldsymbol{X}^T \boldsymbol{Z}$. Let $\zeta : \mathbb{R}^8 \to \mathbb{R}^{3\times3}$ with

$$\zeta(\boldsymbol{q}) = \begin{bmatrix} q_1 & q_3 & q_6 \\ 0 & q_4 & q_7 \\ q_2 & q_5 & q_8 \end{bmatrix}$$

be a function that zero-pads its 8-dimensional input vector before devectorizing into shape $3 \times 3$. Since $\operatorname{vec}(\boldsymbol{Z}) \sim \mathcal{N}\left(\operatorname{vec}(\boldsymbol{X}), \sigma^2 \mathbf{I}_{3N}\right)$ if $\boldsymbol{Z} \sim \mu_{\boldsymbol{X}}$, we have by definition of $\hat{\beta}$ that

$$\mathop{\mathbf{E}}_{\boldsymbol{Z} \sim \mu_{\boldsymbol{X}}} \left[ \frac{\beta_{\boldsymbol{X}'}(\boldsymbol{Z})}{\beta_{\boldsymbol{X}}(\boldsymbol{Z})} \leq \kappa \right] = \mathop{\mathbf{E}}_{\boldsymbol{q} \sim \mathcal{N}(\boldsymbol{m}_{\boldsymbol{X}}, \boldsymbol{\Sigma})} \left[ \frac{\hat{\beta}(\zeta(\boldsymbol{q}_{1:8}))}{\hat{\beta}(\zeta(\boldsymbol{q}_{9:16}))} \leq \kappa \right],$$

with $\boldsymbol{m}_{\boldsymbol{X}} = \boldsymbol{W} \operatorname{vec}(\boldsymbol{X})$, $\boldsymbol{\Sigma} = \sigma^2 \boldsymbol{W} \boldsymbol{W}^T$ and

$$\boldsymbol{W} = \begin{bmatrix} \boldsymbol{X}'_{:,1} & \boldsymbol{X}'_{:,3} & \boldsymbol{0} & \boldsymbol{0} & \boldsymbol{X}_{:,1} & \boldsymbol{X}_{:,3} & \boldsymbol{0} & \boldsymbol{0} \\ \boldsymbol{0} & \boldsymbol{0} & \boldsymbol{X}' & \boldsymbol{0} & \boldsymbol{0} & \boldsymbol{0} & \boldsymbol{X} & \boldsymbol{0} \\ \boldsymbol{0} & \boldsymbol{0} & \boldsymbol{0} & \boldsymbol{X}' & \boldsymbol{0} & \boldsymbol{0} & \boldsymbol{0} & \boldsymbol{X} \end{bmatrix}^T.$$

Thus, the expectation that provides the optimal value of our variance-constrained optimization problem can be probabilistically bounded via Monte Carlo sampling from a 16-dimensional normal distribution, regardless of the data dimensionality.

### F.4.4 Roto-translation invariance in 2D and 3D

For the roto-granslation group $SE(D)$, we prove that evaluating the tight certificate is equivalent to evaluating the tight certificate for rotation group $SO(D)$ after centering the clean and perturbed data $\boldsymbol{X}, \boldsymbol{X}'$ by substracting their averages.

**Theorem 7.** *Let $\tilde{\boldsymbol{X}} = \boldsymbol{X} - \mathbf{1}_N \overline{\boldsymbol{X}}$ with column-wise averages $\overline{\boldsymbol{X}}, \overline{\boldsymbol{X}'} \in \mathbb{R}^{1\times D}$. Further let $\eta$ be a right Haar measure on $SO(3)$ and $\mu_{\boldsymbol{X}}$ our isotropic matrix normal smoothing distribution. Let*

$$\beta_{\boldsymbol{X}}^{SE(D)} = \int_{SE(D)} \int_{\mathbb{R}^D} \exp\left(\langle \boldsymbol{Z}\boldsymbol{R}^T + \mathbf{1}_N \boldsymbol{b}^T, \boldsymbol{X} \rangle_{\mathrm{F}} / \sigma^2\right) \, d\boldsymbol{b} \, d\eta(\boldsymbol{R})$$

*be the Haar integral characterizing the worst-case classifier for roto-translation invariance (see also Appendix F.3.4) and*

$$\beta_{\boldsymbol{X}}^{SO(D)} = \int_{SO(D)} \int_{\mathbb{R}^D} \exp\left(\langle \boldsymbol{Z}\boldsymbol{R}^T, \boldsymbol{X} \rangle_{\mathrm{F}} / \sigma^2\right) \, d\eta(\boldsymbol{R})$$

*be the Haar integral characterizing the worst-case classifier for rotation invariance (see also Appendices F.3.2 and F.3.3). Then, for all $\boldsymbol{V} \in \mathbb{R}^{N\times D}$ and $\kappa \in \mathbb{R}_+$,*

$$\mathop{\mathbf{E}}_{\boldsymbol{Z} \sim \mu_{\boldsymbol{V}}} \left[ \mathbb{1}\left[ \frac{\beta_{\boldsymbol{X}'}^{SE(D)}(\boldsymbol{Z})}{\beta_{\boldsymbol{X}}^{SE(D)}(\boldsymbol{Z})} \leq \kappa \right] \right] = \mathop{\mathbf{E}}_{\boldsymbol{Z} \sim \mu_{\tilde{\boldsymbol{V}}}} \left[ \mathbb{1}\left[ \frac{\beta_{\tilde{\boldsymbol{X}}'}^{SO(D)}(\boldsymbol{Z})}{\beta_{\tilde{\boldsymbol{X}}}^{SO(D)}(\boldsymbol{Z})} \leq \kappa \right] \right].$$

Note that these (with $\boldsymbol{V} = \boldsymbol{X}$ and $\boldsymbol{V} = \boldsymbol{X}'$) are exactly the expectations determining the optimal value of our variance-constrained optimization problem (see Theorem 3).

To prove this theorem, we first simplify the integrand of $\beta_{\boldsymbol{X}}^{SE(D)}(\boldsymbol{Z})$ by moving the rotation matrix into the second argument of the Frobenius inner product (note that, since $\boldsymbol{R}$ is a rotation matrix, we have $\boldsymbol{R}^{-1} = \boldsymbol{R}^T$):

$$\begin{aligned} \langle \boldsymbol{Z}\boldsymbol{R}^T + \mathbf{1}_N \boldsymbol{b}^T, \boldsymbol{X} \rangle_{\mathrm{F}} &= \langle \boldsymbol{Z}\boldsymbol{R}^T + \mathbf{1}_N \boldsymbol{b}^T \boldsymbol{R}\boldsymbol{R}^T, \boldsymbol{X} \rangle_{\mathrm{F}} \\ &= \langle \left(\boldsymbol{Z} + \mathbf{1}_N \boldsymbol{b}^T \boldsymbol{R}\right) \boldsymbol{R}^T, \boldsymbol{X} \rangle_{\mathrm{F}} \\ &= \langle \boldsymbol{Z} + \mathbf{1}_N \boldsymbol{b}^T \boldsymbol{R}, \boldsymbol{X}\boldsymbol{R} \rangle_{\mathrm{F}} \\ &= \langle \boldsymbol{Z} + \mathbf{1}_N (\boldsymbol{R}^T \boldsymbol{b})^T, \boldsymbol{X}\boldsymbol{R} \rangle_{\mathrm{F}} \end{aligned}$$

Next, we bring our integral into the same form as in our derivation of the certificate for translation invariance (see Appendix F.4.1), so that we can easily integrate out the translation vector $\boldsymbol{b}$. First, we eliminate the rotation matrix $\boldsymbol{R}^T$ via the substitution $\boldsymbol{c} = \boldsymbol{R}^T \boldsymbol{b}$:

$$\beta_{\boldsymbol{X}}^{SE(D)}(\boldsymbol{Z}) = \int_{SO(D)} \int_{\mathbb{R}^D} \exp\left(\left\langle \boldsymbol{Z} + \mathbf{1}_N (\boldsymbol{R}^T \boldsymbol{b})^T, \boldsymbol{X}\boldsymbol{R} \right\rangle_{\mathrm{F}} / \sigma^2\right) \, d\boldsymbol{b} \, d\eta(\boldsymbol{R})$$

$$= \int_{SO(D)} \int_{\mathbb{R}^D} \exp\left(\left\langle \boldsymbol{Z} + \mathbf{1}_N \boldsymbol{c}^T, \boldsymbol{X}\boldsymbol{R} \right\rangle_{\mathrm{F}} / \sigma^2\right) \, d\boldsymbol{c} \, d\eta(\boldsymbol{R})$$

For the next step, let $\boldsymbol{A} = \begin{bmatrix} 1 & \mathbf{0}_{N-1}^T \\ -\mathbf{1}_{N-1} & \mathbf{I}_{N-1} \end{bmatrix} \in \mathbb{R}^{N \times N}$ with inverse $\boldsymbol{A}^{-1} = \begin{bmatrix} 1 & \mathbf{0}_{N-1}^T \\ \mathbf{1}_{N-1} & \mathbf{I}_{N-1} \end{bmatrix}$ and all-zeros vector $\mathbf{0}_{N-1} \in \mathbb{R}^{N-1}$, all-ones vector $\mathbf{1}_{N-1} \in \mathbb{R}^{N-1}$ and identity matrix $\mathbf{I}_{N-1}$. Multiplying a vector or matrix with $\boldsymbol{A}$ is equivalent to subtracting the first row from all other rows. We introduce the redundant factor $\boldsymbol{A}^{-1}\boldsymbol{A}$ and make the substitution $\boldsymbol{u} = \boldsymbol{c} + \boldsymbol{Z}_1$ to show that

$$\beta_{\boldsymbol{X}}^{SE(D)}(\boldsymbol{Z}) = \int_{SO(D)} \int_{\mathbb{R}^D} \exp\left(\left\langle \boldsymbol{A}^{-1}\boldsymbol{A}\left(\boldsymbol{Z} + \mathbf{1}_N \boldsymbol{c}^T\right), \boldsymbol{X}\boldsymbol{R} \right\rangle_{\mathrm{F}} / \sigma^2\right) \, d\boldsymbol{c} \, d\eta(\boldsymbol{R})$$

$$= \int_{SO(D)} \int_{\mathbb{R}^D} \exp\left( \left\langle \boldsymbol{A}^{-1} \begin{bmatrix} \boldsymbol{Z}_1^T + \boldsymbol{c}^T \\ (\boldsymbol{Z}_2 - \boldsymbol{Z}_1)^T \\ \cdots \\ (\boldsymbol{Z}_N - \boldsymbol{Z}_1)^T \end{bmatrix}, \boldsymbol{X}\boldsymbol{R} \right\rangle_{\mathrm{F}} / \sigma^2 \right) \, d\boldsymbol{c} \, d\eta(\boldsymbol{R})$$

$$= \int_{SO(D)} \int_{\mathbb{R}^D} \exp\left( \left\langle \boldsymbol{A}^{-1} \begin{bmatrix} \boldsymbol{u}^T \\ (\boldsymbol{Z}_2 - \boldsymbol{Z}_1)^T \\ \cdots \\ (\boldsymbol{Z}_N - \boldsymbol{Z}_1)^T \end{bmatrix}, \boldsymbol{X}\boldsymbol{R} \right\rangle_{\mathrm{F}} / \sigma^2 \right) \, d\boldsymbol{u} \, d\eta(\boldsymbol{R})$$

$$\propto \int_{SO(D)} \int_{\mathbb{R}^D} \prod_{d=1}^D \mathcal{N}\left( \boldsymbol{A}^{-1} \begin{bmatrix} \boldsymbol{u}_d \\ \boldsymbol{Z}_{2,d} - \boldsymbol{Z}_{1,d} \\ \cdots \\ \boldsymbol{Z}_{N,d} - \boldsymbol{Z}_{1,d} \end{bmatrix} \mid \boldsymbol{X}\boldsymbol{R}_{:,d}, \sigma^2 \mathbf{I}_N \right) \, d\boldsymbol{u} \, d\eta(\boldsymbol{R}).$$

Next, we use the behavior of multivariate normal densities under affine transformation (see also Lemma 2), as well as the fact that marginalizing out one dimension of a multivariate normal density is equivalent to dropping the correspond row of the mean and corresponding row and column of the adjacency matrix, to show that

$$\beta_{\boldsymbol{X}}^{SE(D)}(\boldsymbol{Z}) \propto \int_{SO(D)} \int_{\mathbb{R}^D} \prod_{d=1}^D \mathcal{N}\left( \begin{bmatrix} \boldsymbol{u}_d \\ \boldsymbol{Z}_{2,d} - \boldsymbol{Z}_{1,d} \\ \cdots \\ \boldsymbol{Z}_{N,d} - \boldsymbol{Z}_{1,d} \end{bmatrix} \mid \boldsymbol{A}\boldsymbol{X}\boldsymbol{R}_{:,d}, \sigma^2 \boldsymbol{A}\boldsymbol{A}^T \right) \, d\boldsymbol{u} \, d\eta(\boldsymbol{R})$$

$$= \int_{SO(D)} \prod_{d=1}^D \mathcal{N}\left( \begin{bmatrix} \boldsymbol{Z}_{2,d} - \boldsymbol{Z}_{1,d} \\ \cdots \\ \boldsymbol{Z}_{N,d} - \boldsymbol{Z}_{1,d} \end{bmatrix} \mid \boldsymbol{A}_{2:}\boldsymbol{X}\boldsymbol{R}_{:,d}, \sigma^2 \boldsymbol{A}_{2:}(\boldsymbol{A}_{2:})^T \right) \, d\eta(\boldsymbol{R})$$

$$= \int_{SO(D)} \prod_{d=1}^D \mathcal{N}\left( \boldsymbol{A}_{2:}\boldsymbol{Z}_{:,d} \mid \boldsymbol{A}_{2:}\boldsymbol{X}\boldsymbol{R}_{:,d}, \sigma^2 \boldsymbol{A}_{2:}(\boldsymbol{A}_{2:})^T \right) \, d\eta(\boldsymbol{R})$$

$$\propto \int_{SO(D)} \prod_{d=1}^D \exp\left( \frac{1}{\sigma^2} \boldsymbol{Z}_{:,d}^T (\boldsymbol{A}_{2:})^T \left( \boldsymbol{A}_{2:}(\boldsymbol{A}_{2:})^T \right)^{-1} \boldsymbol{A}_{2:}\boldsymbol{X}\boldsymbol{R}_{:,d} \right) \, d\eta(\boldsymbol{R})$$

Finally, reusing the fact that $(\boldsymbol{A}_{2:})^T \left( \boldsymbol{A}_{2:}(\boldsymbol{A}_{2:})^T \right)^{-1} \boldsymbol{A}_{2:}\boldsymbol{X} = \tilde{\boldsymbol{X}}$ with $\tilde{\boldsymbol{X}} = \boldsymbol{X} - \mathbf{1}_N \overline{\boldsymbol{X}}$ from Appendix F.4.1 proves that

$$\beta_{\boldsymbol{X}}^{SE(D)}(\boldsymbol{Z}) \propto \int_{SO(D)} \prod_{d=1}^D \exp\left( \frac{1}{\sigma^2} \boldsymbol{Z}_{:,d}^T \tilde{\boldsymbol{X}}\boldsymbol{R}_{:,d} \right) \, d\eta(\boldsymbol{R})$$

$$= \int_{SO(D)} \exp\left( \left\langle \boldsymbol{Z}\boldsymbol{R}^T, \tilde{\boldsymbol{X}} \right\rangle / \sigma^2 \right) \, d\eta(\boldsymbol{R}).$$

The last term is the Haar integral $\beta_{\tilde{X}}^{SO(D)}(\mathbf{Z})$ characterizing the worst-case classifier for $SO(D)$, after centering $\mathbf{X}$. This means that we have

$$\frac{\beta_{\mathbf{X}'}^{SE(D)}(\mathbf{Z})}{\beta_{\mathbf{X}}^{SE(D)}(\mathbf{Z})} = \frac{\beta_{\tilde{\mathbf{X}}'}^{SO(D)}(\mathbf{Z})}{\beta_{\tilde{\mathbf{X}}}^{SO(D)}(\mathbf{Z})}.$$

Note that, when using $\propto$ in the previous steps, we have only removed constant factors that appear in the enumerator and denominator and thus cancel out.

The last thing we need to do is verify that not only the classifiers, but also their expectations under $\mathbf{V}$ and $\tilde{\mathbf{V}}$ are identical (see Theorem 7). We use the fact that the worst-case classifier only depends on a low-dimensional linear projection of our random input variable $\mathbf{Z}$ (similar to our derivations for Monte Carlo evaluation in Appendices F.4.2 and F.4.3):

$$\beta_{\mathbf{X}}^{SE(D)}(\mathbf{Z}) \propto \int_{SO(D)} \prod_{d=1}^{D} \exp\left(\frac{1}{\sigma^2} \mathbf{Z}_{:,d}^T \tilde{\mathbf{X}} \mathbf{R}_{:,d}\right) d\eta(\mathbf{R})$$

$$= \int_{SO(D)} \prod_{d=1}^{D} \exp\left(\frac{1}{\sigma^2} (\tilde{\mathbf{X}}^T \mathbf{Z}_{:,d})^T \mathbf{R}_{:,d}\right) d\eta(\mathbf{R})$$

$$= \int_{SO(D)} \exp\left(\left\langle \tilde{\mathbf{X}}^T \mathbf{Z}, \mathbf{R}\right\rangle / \sigma^2\right) d\eta(\mathbf{R})$$

By definition of our smoothing distribution, we have $\mathrm{vec}(\mathbf{Z}) \sim \mathcal{N}(\mathrm{vec}(\mathbf{V}) \mid \mathrm{vec}(\mathbf{V}), \sigma^2 \mathbf{I}_{D \cdot N})$ for $\mathbf{Z} \sim \mu_{\mathbf{V}}$. Because our worst-case classifier for $SE(D)$ performs a linear transformation of $\mathrm{vec}(\mathbf{Z})$, we have

$$\mathop{\mathbf{E}}_{\mathbf{Z} \sim \mu_{\mathbf{V}}} \left[ \mathbb{1}\left[\frac{\beta_{\mathbf{X}'}^{SE(D)}(\mathbf{Z})}{\beta_{\mathbf{X}}^{SE(D)}(\mathbf{Z})} \le \kappa\right] \right]$$

$$= \mathop{\mathbf{E}}_{\mathbf{Z} \sim \mu_{\mathbf{V}}} \left[ \mathbb{1}\left[\frac{\int_{SO(D)} \exp\left(\left\langle \tilde{\mathbf{X}'}^T \mathbf{Z}, \mathbf{R}\right\rangle / \sigma^2\right) d\eta(\mathbf{R})}{\int_{SO(D)} \exp\left(\left\langle \tilde{\mathbf{X}}^T \mathbf{Z}, \mathbf{R}\right\rangle / \sigma^2\right) d\eta(\mathbf{R})} \le \kappa\right] \right]$$

$$= \mathop{\mathbf{E}}_{\mathbf{q} \sim \mathcal{N}(\mathbf{m}_{\mathbf{V}}, \mathbf{\Sigma})} \left[ \mathbb{1}\left[\frac{\int_{SO(D)} \exp\left(\left\langle \mathrm{vec}^{-1}(\mathbf{q}_{1:9}), \mathbf{R}\right\rangle / \sigma^2\right) d\eta(\mathbf{R})}{\int_{SO(D)} \exp\left(\left\langle \mathrm{vec}^{-1}(\mathbf{q}_{10:18}), \mathbf{R}\right\rangle / \sigma^2\right) d\eta(\mathbf{R})} \le \kappa\right] \right],$$

with $\mathbf{m}_{\mathbf{V}} = \mathbf{W} \mathrm{vec}(\mathbf{V})$ and $\mathbf{\Sigma} = \mathbf{W}\mathbf{W}^T$, where

$$\mathbf{W} = \begin{bmatrix} \tilde{\mathbf{X}}' & \mathbf{0} & \mathbf{0} & \tilde{\mathbf{X}} & \mathbf{0} & \mathbf{0} \\ \mathbf{0} & \tilde{\mathbf{X}}' & \mathbf{0} & \mathbf{0} & \tilde{\mathbf{X}} & \mathbf{0} \\ \mathbf{0} & \mathbf{0} & \tilde{\mathbf{X}}' & \mathbf{0} & \mathbf{0} & \tilde{\mathbf{X}} \end{bmatrix}^T.$$

Similarly, we have for $SO(3)$ (after centering $\mathbf{V}$), that

$$\mathop{\mathbf{E}}_{\mathbf{Z} \sim \mu_{\tilde{\mathbf{V}}}} \left[ \mathbb{1}\left[\frac{\beta_{\tilde{\mathbf{X}}'}^{SO(D)}(\mathbf{Z})}{\beta_{\tilde{\mathbf{X}}}^{SO(D)}(\mathbf{Z})} \le \kappa\right] \right]$$

$$= \mathop{\mathbf{E}}_{\mathbf{q} \sim \mathcal{N}(\mathbf{m}_{\tilde{\mathbf{V}}}, \mathbf{\Sigma})} \left[ \mathbb{1}\left[\frac{\int_{SO(D)} \exp\left(\left\langle \mathrm{vec}^{-1}(\mathbf{q}_{1:9}), \mathbf{R}\right\rangle / \sigma^2\right) d\eta(\mathbf{R})}{\int_{SO(D)} \exp\left(\left\langle \mathrm{vec}^{-1}(\mathbf{q}_{10:18}), \mathbf{R}\right\rangle / \sigma^2\right) d\eta(\mathbf{R})} \le \kappa\right] \right],$$

with $\mathbf{m}_{\tilde{\mathbf{V}}} = \mathbf{W} \mathrm{vec}(\tilde{\mathbf{V}})$ and $\mathbf{\Sigma} = \mathbf{W}\mathbf{W}^T$. Finally, due to the fact (see Lemma 7) that calculating an inner product between a centered and an uncentered vector (here: columns of $\tilde{\mathbf{X}}$ and $\mathbf{V}$) is equivalent to centering both vectors before calculating the inner product (here: columns of $\tilde{\mathbf{X}}$ and $\tilde{\mathbf{V}}$), we have $\mathbf{m}_{\mathbf{V}} = \mathbf{m}_{\tilde{\mathbf{V}}}$. Thus, both expectations are equal, which concludes our proof.

## F.5 Monte Carlo certification

All of the discussed tight certificates, save for the one for translation invariance, are of the form

$$\min_{h \in \mathbb{H}_T} \Pr_{\mathbf{Z} \sim \mu_{\mathbf{X}'}} [h(\mathbf{Z}) = y^*]] = \Pr_{V \sim \upsilon^{(1)}} [\rho(V) \leq \kappa]$$

$$\text{with } \kappa \in \mathbb{R} \text{ such that } \Pr_{V \sim \upsilon^{(2)}} [\rho(V) \leq \kappa] = p_{\mathbf{X}, y^*}$$

$$\text{and } p_{\mathbf{X}, y^*} := \Pr_{\mathbf{Z} \sim \mu_{\mathbf{X}}} [g(\mathbf{Z}) = y^*],$$

where $g : \mathbb{R}^{N \times D} \to \mathbb{Y}$ is our base classifier, $\mu_{\mathbf{X}}, \mu_{\mathbf{X}'}$ are the clean and perturbed smoothing distribution, $\upsilon^{(1)}$ and $\upsilon^{(2)}$ are distributions over some set $\mathbb{S}$, and $\rho : \mathbb{S} \to \mathbb{R}$ is some arbitrary scalar-valued function.

As discussed in Section 6.3, we propose to compute a narrow probabilistic lower bound on $\min_{h \in \mathbb{H}_T} \Pr_{\mathbf{Z} \sim \mu_{\mathbf{X}'}} [h(\mathbf{Z}) = y^*]]$ by combining three confidence bounds.

Before computing these bounds, we have to inspect the monotonicity of our certificate. Evidently, $\Pr_{V \sim \upsilon^{(1)}} [\rho(V) \leq \kappa]$ is monotonically decreasing in $\kappa$. Furthermore, $\kappa$ is monotonically decreasing in $p_{\mathbf{X}, y^*}$: If $p_{\mathbf{X}, y^*}$ decreases, then a smaller $\kappa$ is sufficient for fulfilling the constraint $\Pr_{V \sim \upsilon^{(2)}} [\rho(V) \leq \kappa] = p_{\mathbf{X}, y^*}$. Thus, we can compute a lower bound on our certificate by

1. lower-bounding $p_{\mathbf{X}, y^*}$, i.e. $\underline{p_{\mathbf{X}, y^*}} \leq p_{\mathbf{X}, y^*}$,

2. lower-bounding the $\kappa \in \mathbb{R}$ that fulfills $\Pr_{V \sim \upsilon^{(2)}} [\rho(V) \leq \kappa] = \underline{p_{\mathbf{X}, y^*}}$, i.e. $\underline{\kappa} \leq \kappa$

3. and then lower-bounding $\Pr_{V \sim \upsilon^{(1)}} [\rho(V) \leq \underline{\kappa}]$.

The random variable $\mathbb{1}[g(\mathbf{Z}) = y^*]$ is a Bernoulli random variable. Just like other randomized smoothing methods (e.g. [26]), we can lower-bound $p_{\mathbf{X}, y^*}$ by evaluating $g$ on $N_1$ samples from $\mu_{\mathbf{X}}$ and computing a binomial proportion confidence bound, such as the the Clopper-Pearson confidence bound [141].

Lower-bounding $\kappa$ requires computing a lower bound on $F^{-1}(\underline{p_{\mathbf{X}, y^*}})$, where $F^{-1}$ is the quantile function of $\rho(V)$ with $V \sim \upsilon^{(2)}$. A non-parametric lower confidence bound on this quantile can be constructed by evaluating $\rho(V)$ on $N_2$ samples from $\upsilon^{(2)}$ and returning the largest order statistic $R^{(n)}$ such that $[\rho(V) \leq R^{(n)}] \leq \underline{p_{\mathbf{X}, y^*}}$ holds with high probability (see Section 5.2.1. of [142] and Algorithm 1 below).

The random variable $[\rho(V) \leq \underline{\kappa}]$ is another Bernoulli random variable and can thus be lower-bounded by evaluating $\rho(V)$ on $N_3$ samples from $\mu_{\mathbf{X}'}$ and computing a binomial proportion confidence bound.

We want all three confidence bounds to simultaneously hold with high probability $1 - \alpha$. We ensure this by using Holm-Bonferroni [131] correction to account for the multiple comparisons problem.[4] In our case, this corresponds to computing the first bound with significance $\alpha$, the second one with significance $\alpha/2$ and the last one with significance $\alpha/3$.

Algorithm 1 summarizes our certification procedure. LOWERCOUNFBOUND refers to the Clopper-Pearson lower confidence bound. BINPVALUE$(n, N_2, \geq, \underline{p_{\mathbf{X}, y^*}})$ refers to the $p$-value of a Binomial test with the null-hypothesis that the success probability is greater or equal $\underline{p_{\mathbf{X}, y^*}}$. Note that only the first confidence bound requires evaluating the base classifier $g$. For the other two confidence bounds a large number of samples can be evaluated at little computational cost.

---

[4]Holm-Bonferroni correction has already been used in the context of randomized smoothing, see [91].

---

**Algorithm 1** Monte Carlo certification procedure

---

**function** PROBCERTIFY($y^*$, $g$, $\mu_{\boldsymbol{X}}$, $\upsilon^{(1)}$, $\upsilon^{(2)}$, $N_1$, $N_2$, $N_3$, $\alpha$)

$\boldsymbol{Z}^{(1)}, \ldots, \boldsymbol{Z}^{(N_1)} \leftarrow$ SAMPLE($\mu_{\boldsymbol{X}}$, $N_1$)

$\text{count}_1 \leftarrow \sum_{n=1}^{N_1} \mathbb{1}\left[g(\boldsymbol{Z}^{(n)}) = y^*\right]$

$\underline{p_{\boldsymbol{X},y^*}} \leftarrow$ LOWERCONFBOUND($\text{count}_1$, $N_1$, $1 - \alpha$)  $\qquad\qquad\qquad$ ▷ First bound

$n^* \leftarrow \max\left\{n \mid \text{BINPVALUE}(n, N_2, \geq, \underline{p_{\boldsymbol{X},y^*}}) < \frac{\alpha}{2}\right\}$

$V^{(1)}, \ldots, V^{(N_2)} \leftarrow$ SAMPLE($\upsilon^{(2)}$, $N_2$)

$R^{(1)}, \ldots, R^{(N_2)} \leftarrow$ SORTASCENDING($\rho(V^{(1)}), \ldots, \rho(V^{(N_2)})$)

$\underline{\kappa} \leftarrow R^{(n^*)}$  $\qquad\qquad\qquad\qquad\qquad\qquad\qquad\qquad\qquad\qquad$ ▷ Second bound

$V^{(1)}, \ldots, V^{(N_3)} \leftarrow$ SAMPLE($\upsilon^{(1)}$, $N_3$)

$\text{count}_2 \leftarrow \sum_{n=1}^{N_3} \mathbb{1}\left[\rho(V) \leq \underline{\kappa}\right]$

**return** LOWERCONFBOUND($\text{count}_2$, $N_3$, $1 - \frac{\alpha}{3}$)  $\qquad\qquad\qquad$ ▷ Third bound

---

# G  Proof of Theorem 5

Next, we prove that the post-processing-based certificate for rotation-invariance ( $\mathbb{T} = SO(D)$ ) is not tight, which we formalized in Section 6.3 as follows:

**Theorem 5.** *Let $g : \mathbb{R}^{N \times D} \to \mathbb{Y}$ be invariant under $\mathbb{T} = SO(D)$ and $\mathbb{H}_{\mathbb{T}}$ be defined as in Eq. (2). Assume that perturbed input $\boldsymbol{X}'$ is not obtained via rotation of $\boldsymbol{X}$, i.e. $\nexists \boldsymbol{R} \in SO(D) : \boldsymbol{X}' = \boldsymbol{X} \boldsymbol{R}^T$. Further assume that $p_{\boldsymbol{X}, y^*} \in (0, 1)$. Then, for all $\boldsymbol{R} \in SO(D)$:*

$$\min_{h \in \mathbb{H}_{\mathbb{T}}} \Pr_{\boldsymbol{Z} \sim \mu_{\boldsymbol{X}'}} [h(\boldsymbol{Z}) = y^*] > \Phi \left( \Phi^{-1} (p_{\boldsymbol{X}, y^*}) - \frac{1}{\sigma} \left\| \boldsymbol{X}' \boldsymbol{R}^T - \boldsymbol{X} \right\|_2 \right). \tag{8}$$

The right-hand side term is the optimal value of the black-box optimization problem evaluated for perturbed input $\boldsymbol{X}' \boldsymbol{R}^T$, i.e.

$$\min_{h \in \mathbb{H}} \Pr_{\boldsymbol{Z} \sim \mu_{\boldsymbol{X}' \boldsymbol{R}^T}} [h(\boldsymbol{Z}) = y^*],$$

with $\mathbb{H} = \left\{ h : \mathbb{R}^{N \times D} \to \mathbb{Y} \mid \Pr_{\boldsymbol{Z} \sim \mu_{\boldsymbol{X}}} [h(\boldsymbol{Z}) = y^*] \geq p_{\boldsymbol{X}, y^*} \right\}$. The left-hand side term is the optimal value of the gray-box optimization problem evaluated for perturbed input $\boldsymbol{X}'$. Note that $\mathbb{H}_{\mathbb{T}}$ is a set of rotation invariant classifiers. In Appendix D we have proven that rotation of the smoothing distribution's mean does not have an effect on the prediction probabilities of such classifiers, i.e.

$$\min_{h \in \mathbb{H}_{\mathbb{T}}} \Pr_{\boldsymbol{Z} \sim \mu_{\boldsymbol{X}'}} [h(\boldsymbol{Z}) = y^*] = \min_{h \in \mathbb{H}_{\mathbb{T}}} \Pr_{\boldsymbol{Z} \sim \mu_{\boldsymbol{X}' \boldsymbol{R}^T}} [h(\boldsymbol{Z}) = y^*].$$

Thus, we can prove Theorem 5 by proving the following, more general statement:

**Lemma 8.** *Let $g : \mathbb{R}^{N \times D} \to \mathbb{Y}$ be invariant under $\mathbb{T} = SO(D)$ and $\mathbb{H}_{\mathbb{T}}$ be defined as in Eq. (2). Further assume that $\boldsymbol{X}' \neq \boldsymbol{X}$ and that $p_{\boldsymbol{X}, y^*} \in (0, 1)$. Then:*

$$\min_{h \in \mathbb{H}_{\mathbb{T}}} \Pr_{\boldsymbol{Z} \sim \mu_{\boldsymbol{X}'}} [h(\boldsymbol{Z}) = y^*] > \min_{h \in \mathbb{H}} \Pr_{\boldsymbol{Z} \sim \mu_{\boldsymbol{X}'}} [h(\boldsymbol{Z}) = y^*] \tag{42}$$

We can do so by showing that no optimal solution to the r.h.s. black-box problem from Eq. (42) is a feasible solution to the l.h.s. gray-box problem. More formally, let $\mathbb{H}^* \subseteq \mathbb{H}$ be the set of all classifiers minimizing the r.h.s. black box problem. We have $\mathbb{H}^* \subseteq \mathbb{H}$ and $\mathbb{H}_{\mathbb{T}} \subseteq \mathbb{H}$. By showing that $\mathbb{H}^* \cap \mathbb{H}_{\mathbb{T}} = \emptyset$, we prove that $\mathbb{H}_{\mathbb{T}} \subseteq \mathbb{H} \setminus \mathbb{H}^*$, i.e. all rotation invariant classifiers from $\mathbb{H}_{\mathbb{T}}$ yield strictly larger optimal values.

Recall that *an* optimal solution to the r.h.s. black-box problem in Eq. (42) is given by the Neyman-Pearson lemma [107]. Later work on hypothesis testing has shown that the Neyman-Pearson lemma is not only a sufficient, but a necessary condition for optimality[143]: Every most powerful test must fulfill the likelihood ratio inequalities, save for sets of zero measure. For our formulation of the Neyman-Pearson lemma (see Lemma 5), this means that any classifier $h$ that is an optimal solution to the r.h.s. black-box problem from Eq. (42) must fulfill

$$h(\boldsymbol{Z}) \in \begin{cases} \{y^*\} & \text{if } \frac{\mu_{\boldsymbol{X}'}(\boldsymbol{Z})}{\mu_{\boldsymbol{X}}(\boldsymbol{Z})} < \kappa \\ \mathbb{Y} \setminus \{y^*\} & \text{if } \frac{\mu_{\boldsymbol{X}'}(\boldsymbol{Z})}{\mu_{\boldsymbol{X}}(\boldsymbol{Z})} > \kappa \end{cases}$$

for some $\kappa \in \mathbb{R}$, save for sets of zero measure. Cohen et al. [26] have shown that if $\boldsymbol{X}' \neq \boldsymbol{X}$ and $p \in (0, 1)$ and $\mu_{\boldsymbol{X}}, \mu_{\boldsymbol{X}'}$ are isotropic Gaussian distributions, then these likelihood ratio inequalities correspond to a linear decision boundary, i.e. any optimal solution most fulfill

$$h(\boldsymbol{Z}) \in \begin{cases} \{y^*\} & \text{if } \langle \boldsymbol{W}, \boldsymbol{Z} \rangle_{\mathrm{F}} + b < 0 \\ \mathbb{Y} \setminus \{y^*\} & \text{if } \langle \boldsymbol{W}, \boldsymbol{Z} \rangle_{\mathrm{F}} + b > 0 \end{cases} \tag{43}$$

for some $b \in \mathbb{R}$ and $\boldsymbol{W} \in \mathbb{R}^{N \times D}$ with $\boldsymbol{w} \neq \boldsymbol{0}_{N,D}$, save for sets of zero measure. Here, $\langle \boldsymbol{A}, \boldsymbol{B} \rangle_{\mathrm{F}} = \mathrm{vec}(\boldsymbol{W})^T \mathrm{vec}(\boldsymbol{Z})$ is the Frobenius inner product.

Therefore, we can prove Lemma 8 by showing that classifiers complying with Eq. (43) (save for sets of zero measure) are not rotation invariant, i.e. they are not feasible solutions to the gray-box optimization problem.

We first do this for the case that $D$ is even and then for the case that $D$ is odd.

**Lemma 9.** *Assume that $D = 2k$ for some $k \in \mathbb{N}$. Let $h : \mathbb{R}^{N \times D} \to \mathbb{Y}$ be a classifier that fulfills*

$$h(\boldsymbol{Z}) \in \begin{cases} \{y^*\} & \text{if } \langle \boldsymbol{W}, \boldsymbol{Z} \rangle_{\mathrm{F}} + b < 0 \\ \mathbb{Y} \setminus \{y^*\} & \text{if } \langle \boldsymbol{W}, \boldsymbol{Z} \rangle_{\mathrm{F}} + b > 0 \end{cases} \tag{44}$$

*for some $b \in \mathbb{R}$ and $\boldsymbol{w} \in \mathbb{R}^{N \times D}$ with $\boldsymbol{w} \neq \boldsymbol{0}_{N \cdot D}$, save for sets of zero measure. Then, there is an input $\hat{\boldsymbol{Z}}$ and a rotation matrix $\boldsymbol{R} \in SO(D)$ such that $h(\hat{\boldsymbol{Z}}) \neq h(\hat{\boldsymbol{Z}} \boldsymbol{R}^T)$.*

*Proof.* **Case 1.** Assume that $b \geq 0$.

Consider the set $\mathbb{S} = \{ \boldsymbol{Z} \mid \langle \boldsymbol{W}, \boldsymbol{Z} \rangle_{\mathrm{F}} < b \}$. By construction, we have $\forall \boldsymbol{Z} \in \mathbb{S} : \langle \boldsymbol{W}, \boldsymbol{Z} \rangle_{\mathrm{F}} + b < 0$ and $\forall \boldsymbol{Z} \in \mathbb{S} : \langle \boldsymbol{W}, -\boldsymbol{Z} \rangle_{\mathrm{F}} + b > 0$.

There must be at least one $\hat{\boldsymbol{Z}} \in \mathbb{S}$ with $h(\hat{\boldsymbol{Z}}) = y^*$ and $h(\hat{\boldsymbol{Z}}) \neq y^*$. Otherwise, all points $\boldsymbol{Z} \in \mathbb{S}$ would need to fulfill $h(\boldsymbol{Z}) \neq y^*$ or $h(-\boldsymbol{Z}) = y^*$, in which case we would have sets of non-zero measure violating the likelihood ratio inequalities from Eq. (44).

Note that $-\hat{\boldsymbol{Z}} = \hat{\boldsymbol{Z}}(-\mathbf{I}_N)^T$ and $-\mathbf{I}_N \in SO(D)$, because $-\mathbf{I}_N$ is orthonormal and $\det(-\mathbf{I}_N) = 1$, due to $D$ being even. Thus, we have found an input $\hat{\boldsymbol{Z}}$ and a rotation matrix $\boldsymbol{R} \in SO(D)$ such that $h(\boldsymbol{Z}) \neq h(\boldsymbol{Z} \boldsymbol{R}^T)$.

**Case 2.** Assume that $b < 0$. This case follows analogously by constructing the set $\mathbb{S} = \{ \boldsymbol{Z} \mid \langle \boldsymbol{W}, \boldsymbol{Z} \rangle_{\mathrm{F}} > b \}$ and considering a point $\hat{\boldsymbol{Z}} \in \mathbb{S}$ with $h(\hat{\boldsymbol{Z}}) \neq y^*$ and $h(-\hat{\boldsymbol{Z}}) = y^*$ ◻

Now, we consider the case that $D$ is odd. Here, $-\mathbf{I}_N \notin SO(D)$, because $\det(-\mathbf{I}_N) = -1$. Therefore, we will need to use slightly more complicated constructions.

**Lemma 10.** *Assume that $D = 2k - 1$ for some $k \in \mathbb{N}$. Let $h : \mathbb{R}^{N \times D} \to \mathbb{Y}$ be a classifier that fulfills*

$$h(\boldsymbol{Z}) \in \begin{cases} \{y^*\} & \text{if } \langle \boldsymbol{W}, \boldsymbol{Z} \rangle_{\mathrm{F}} + b < 0 \\ \mathbb{Y} \setminus \{y^*\} & \text{if } \langle \boldsymbol{W}, \boldsymbol{Z} \rangle_{\mathrm{F}} + b > 0 \end{cases} \tag{45}$$

*for some $b \in \mathbb{R}$ and $\boldsymbol{w} \in \mathbb{R}^{N \times D}$ with $\boldsymbol{w} \neq \boldsymbol{0}_{N \cdot D}$, save for sets of zero measure. Then, there is an input $\hat{\boldsymbol{Z}}$ and a rotation matrix $\boldsymbol{R} \in SO(D)$ such that $h(\hat{\boldsymbol{Z}}) \neq h(\hat{\boldsymbol{Z}} \boldsymbol{R}^T)$.*

*Proof.* In the following, let $\boldsymbol{A} = \begin{bmatrix} 1 & \boldsymbol{0}_D^T \\ \boldsymbol{0}_{D-1} & -\mathbf{I}_{D-1} \end{bmatrix}$. We have $\boldsymbol{A} \in SO(D)$, because $\boldsymbol{A}$ is orthonormal and $\det(\boldsymbol{A}) = 1$, due to $D$ being odd.

**Case 1.** Assume that $b \geq 0$.

We know that $\boldsymbol{W} \neq \boldsymbol{0}_{N,D}$. Without loss of generality, assume that $\boldsymbol{W}_{:,2:} \neq \boldsymbol{0}_{N,(D-1)}$, i.e. at least one of the the last $D - 1$ columns is non-zero.

Consider the set $\mathbb{S} = \{ \boldsymbol{Z} \mid \boldsymbol{W}_{:,1}^T \boldsymbol{Z}_{:,1} \in [0, 1] \wedge \langle \boldsymbol{W}_{:,2:}, \boldsymbol{Z}_{:,2:} \rangle_{\mathrm{F}} < b - 1 \}$. By construction, we have $\forall \boldsymbol{Z} \in \mathbb{S} : \langle \boldsymbol{W}, \boldsymbol{Z} \rangle_{\mathrm{F}} + b < 0$ and $\forall \boldsymbol{Z} \in \mathbb{S} : \langle \boldsymbol{W}, \boldsymbol{Z} \boldsymbol{A}^T \rangle_{\mathrm{F}} + b > 0$.

There must be at least one $\hat{\boldsymbol{Z}} \in \mathbb{S}$ with $h(\hat{\boldsymbol{Z}}) = y^*$ and $h(\hat{\boldsymbol{Z}} \boldsymbol{A}^T) \neq y^*$. Otherwise, all points $\boldsymbol{Z} \in \mathbb{S}$ would need to fulfill $h(\boldsymbol{Z}) \neq y^*$ or $h(\boldsymbol{Z} \boldsymbol{A}^T) = y^*$, in which case we would have sets of non-zero measure violating Eq. (44).

**Case 2.** Assume that $b < 0$. This case follows analogously by constructing the set $\mathbb{S} = \{ \boldsymbol{Z} \mid \boldsymbol{W}_{:,1}^T \boldsymbol{Z}_{:,1} \in [-1, 0] \wedge \langle \boldsymbol{W}_{:,2:}, \boldsymbol{Z}_{:,2:} \rangle_{\mathrm{F}} > b + 1 \}$ and considering a point $\hat{\boldsymbol{Z}} \in \mathbb{S}$ with $h(\hat{\boldsymbol{Z}}) \neq y^*$ and $h(\boldsymbol{A} \hat{\boldsymbol{Z}}) = y^*$. ◻

Lemmas 9 and 10 combined prove Lemma 8, which in turn proves Theorem 5.

# H  Multi-class certificates

In our discussions in Sections 3.1, 5 and 6, we have only considered binary certificates[5]. That is, prediction $y^*$ of smoothed classifier is certifiably robust to a perturbed input $\boldsymbol{X}'$ if the probability of base classifier $g$ predicting $y^*$ under perturbed smoothing distribution $\mu_{\boldsymbol{X}'}$ is greater than the probability of all other classes combined. That is, $p_{\boldsymbol{X}',y^*} > \frac{1}{2}$. We have then, for different invariances, computed lower bounds $\underline{p_{\boldsymbol{X}',y^*}} \leq p_{\boldsymbol{X}',y^*}$. If $\underline{p_{\boldsymbol{X}',y^*}} > \frac{1}{2}$, then $p_{\boldsymbol{X}',y^*} > \frac{1}{2}$ and the prediction is certifiably robust.

Alternatively, one can certify robustness by showing that $y^*$ remains more likely than the second most likely class under the perturbed smoothing distribution [26], i.e. $p_{\boldsymbol{X}',y^*} > \max_{y \neq y^*} p_{\boldsymbol{X}',y}$. To this end, one can compute a lower bound $\underline{p_{\boldsymbol{X}',y^*}} \leq p_{\boldsymbol{X}',y^*}$ and an upper-bound $\overline{\max_{y \neq y^*} p_{\boldsymbol{X}',y}} \geq \max_{y \neq y^*} p_{\boldsymbol{X}',y}$. If $\underline{p_{\boldsymbol{X}',y^*}} > \overline{\max_{y \neq y^*} p_{\boldsymbol{X}',y}}$, then robustness is guaranteed. Due to the monotonicity of randomized smoothing certificates w.r.t. to the prediction probabilities, this can be achieved by showing that $\underline{p_{\boldsymbol{X}',y^*}} > \overline{p_{\boldsymbol{X}',y'}}$, where $y' = \mathrm{argmax}_{y \neq y^*} p_{\boldsymbol{X},y}$ is the second most likely class under the clean smoothing distribution. This second most likely class $y'$ can be found via a binomial test, see [26, 144].

The lower bound $\underline{p_{\boldsymbol{X}',y^*}}$ can be obtained using the same formulae discussed throughout the main text (e.g. $\underline{p_{\boldsymbol{X}',y^*}} = \Phi\left(\Phi^{-1}(p_{\boldsymbol{X},y'}) - \frac{||\Delta||_2}{\sigma}\right)$ for black-box randomized smoothing). The upper bound $\overline{p_{\boldsymbol{X}',y'}}$ can be found via the Neyman-Pearson upper bound discussed in Appendix F.2:

**Lemma 6** (Neyman-Pearson upper bound). *Let $\mu_{\boldsymbol{X}'}$, $\mu_{\boldsymbol{X}}$, be two continuous distributions over a measurable set $\mathbb{A}$ such that, for all $\kappa \in \mathbb{R}_+$, the set $\left\{z \mid \frac{\mu_{\boldsymbol{X}'}(z)}{\mu_{\boldsymbol{X}}(z)} = \kappa\right\}$ has measure zero. Consider an arbitrary label set $\mathbb{Y}$, a specific class label $y \in \mathbb{Y}$ and scalar $p \in [0,1]$. Then*

$$\left(\max_{h:\mathbb{A} \to \mathbb{Y}} \Pr_{z \sim \mu_{\boldsymbol{X}'}}[h(z) = y] \ s.t. \ \Pr_{z \sim \mu_{\boldsymbol{X}}}[h(z) = y] \leq p\right) = \underset{z \sim \mu_{\boldsymbol{X}'}}{\mathbb{E}}[h^*(z)]$$

$$\text{with } h^*(z) = \mathbb{1}\left[\frac{\mu_{\boldsymbol{X}'}(z)}{\mu_{\boldsymbol{X}}(z)} \geq \kappa\right] \text{ and } \kappa \in \mathbb{R}_+ \text{ such that } \underset{z \sim \mu_{\boldsymbol{X}}}{\mathbb{E}}[h^*(z)] = p.$$

Note that it is identical to the lower bound, save for replacing "$\leq$" with "$\geq$" and vice-versa.

The resulting black-box upper bound is $\overline{p_{\boldsymbol{X}',y'}} = \Phi\left(\Phi^{-1}(p_{\boldsymbol{X},y'}) + \frac{||\Delta||_2}{\sigma}\right)$. It is identical to the black-box lower bound, save for replacing "$-$" with "$+$". Substituting both bounds into $\underline{p_{\boldsymbol{X}',y^*}} > \overline{\max_{y \neq y^*} p_{\boldsymbol{X}',y}}$ and applying the inverse-normal CDF $\Phi^{-1}$ to both sides shows that the classifier is certifiably robust if $||\Delta|| \leq \frac{\sigma}{2}\left(\Phi^{-1}(p_{\boldsymbol{X}',y^*}) - \Phi^{-1}(p_{\boldsymbol{X}',y'})\right)$ [26].

Evidently, the orbit-based versions of the multi-class certificates are identical to their binary counterparts, except that the radius of the underlying black-box certificate changes.

For the tight gray-box approach, upper bounds can be derived by going through the same derivations as in Appendix F, but using the Neyman-Pearson upper bounds from Lemma 6, i.e. replacing "$\leq$" with "$\geq$" and vice-versa. (see Lemma 6). In the case of translation invariance, this results in upper bound $\Phi\left(\Phi^{-1}(p_{\boldsymbol{X},y'}) + \frac{||\Delta - \mathbf{1}_N\overline{\Delta}||_2}{\sigma}\right)$). In the cases involving rotation invariance, this results in bounds of the form

$$\Pr_{V \sim v^{(1)}}[\rho(V) \geq \kappa]$$

$$\text{with } \kappa \in \mathbb{R} \text{ such that } \Pr_{V \sim v^{(2)}}[\rho(V) \geq \kappa] = p_{\boldsymbol{X},y'}$$

$$\text{and } p_{\boldsymbol{X},y^*} := \Pr_{\boldsymbol{Z} \sim \mu_{\boldsymbol{X}}}[g(\boldsymbol{Z}) = y'],$$

i.e. the same certificates, but with "$\geq \kappa$" instead of "$\leq \kappa$".

Our Monte Carlo certification procedure proposed in Appendix F.5 can also easily be adapted to computing an upper bound by replacing "$\leq$" with "$\geq$" and vice-versa.:

---

[5]Not to be confused with certificates that are limited to binary classifiers.

---

**Algorithm 2** Monte Carlo certification procedure (Upper bound)

---

    **function** PROBCERTIFYUPPER($y^*, g, \mu_{\boldsymbol{X}}, \upsilon^{(1)}, \upsilon^{(2)}, N_1, N_2, N_3, \alpha$)

        $\boldsymbol{Z}^{(1)}, \ldots, \boldsymbol{Z}^{(N_1)} \leftarrow$ SAMPLE($\mu_{\boldsymbol{X}}, N_1$)

        $\text{count}_1 \leftarrow \sum_{n=1}^{N_1} \mathbb{1}\left[g(\boldsymbol{Z}^{(n)}) = y^*\right]$

        $\overline{p_{\boldsymbol{X},y'}} \leftarrow$ UPPERCONFBOUND($\text{count}_1, N_1, 1-\alpha$)                                       $\triangleright$ First bound

        $n^* \leftarrow \max\left\{n \mid \text{BINPVALUE}(n, N_2, \leq, 1-\overline{p_{\boldsymbol{X},y'}}) < \frac{\alpha}{2}\right\}$

        $V^{(1)}, \ldots, V^{(N_2)} \leftarrow$ SAMPLE($\upsilon^{(2)}, N_2$)

        $R^{(1)}, \ldots, R^{(N_2)} \leftarrow$ SORTASCENDING($\rho(V^{(1)}), \ldots, \rho(V^{(N_2)})$)

        $\underline{\kappa} \leftarrow R^{(n^*)}$                                                      $\triangleright$ Second bound

        $\overline{V}^{(1)}, \ldots, V^{(N_3)} \leftarrow$ SAMPLE($\upsilon^{(1)}, N_3$)

        $\text{count}_2 \leftarrow \sum_{n=1}^{N_3} \mathbb{1}\left[\rho(V) \geq \underline{\kappa}\right]$

        **return** UPPERCONFBOUND($\text{count}_2, N_3, 1-\frac{\alpha}{3}$)                            $\triangleright$ Third bound

---

Note that we still need to lower-bound threshold $\kappa$, because this increases the probability $\Pr_{V \sim \upsilon^{(1)}}[\rho(V) \geq t]$ and thus leads to more pessimistic, sound multi-class certificates.

Finally note that, because the multi-class certificates are just a combination of two binary certificates, our evaluation of binary invariance-aware certificates in in Section 8 is a good indicator of the performance of invariance-aware multi-class certificates.

# I   Inverse certificates

In Section 8.1, we evaluate inverse certificates, i.e. compute the smallest prediction probability $p_{\min}$ for which robustness can still be certified, given the remaining certificate parameters such as $||\mathbf{\Delta}||_2$ or $||\mathbf{X}||_2$.

In the case of the black-box randomized smoothing baseline, $p_{\min}$ can be calculated by solving $\Phi\left(\Phi^{(-1)}(p) - \frac{||\mathbf{\Delta}||_2}{\sigma}\right) = \frac{1}{2}$ for $p$, which results in $p_{\min} = \Phi\left(\frac{||\mathbf{\Delta}||_2}{\sigma}\right)$.

Similarly, the gray-box certificates for translation invariance has $p_{\min} = \Phi\left(\frac{||\mathbf{\Delta} - \mathbf{1}_N \overline{\mathbf{\Delta}}||_2}{\sigma}\right)$, where $\overline{\mathbf{\Delta}} \in \mathbb{R}^{1 \times D}$ are the column-wise averages.

The tight gray-box certificates involving rotation invariance do not have closed-form analytic expressions. Instead, they are of the form

$$\min_{h \in \mathbb{H}_{\mathbb{T}}} \Pr_{\mathbf{Z} \sim \mu_{\mathbf{X}'}} \left[ h(\mathbf{Z}) = y^* ] \right] = \Pr_{V \sim \upsilon^{(1)}} \left[ \rho(V) \leq \kappa \right]$$

$$\text{with } \kappa \in \mathbb{R} \text{ such that } \Pr_{V \sim \upsilon^{(2)}} \left[ \rho(V) \leq \kappa \right] = p_{\mathbf{X}, y^*}.$$

Here, an inverse certificate can be computed by first finding a threshold $\kappa$ such that $\Pr_{V \sim \upsilon^{(1)}} \left[ \rho(V) \leq \kappa \right] = \frac{1}{2}$ and then evaluating $\Pr_{V \sim \upsilon^{(2)}} \left[ \rho(V) \leq \kappa \right]$. That is,

$$p_{\min} = \Pr_{V \sim \upsilon^{(2)}} \left[ \rho(V) \leq \kappa \right]$$

$$\text{with } \kappa \in \mathbb{R} \text{ such that } \Pr_{V \sim \upsilon^{(1)}} \left[ \rho(V) \leq \kappa \right] = \frac{1}{2}.$$

To ensure a fair comparison with our baselines, we modify the Monte Carlo certification procedure proposed in Appendix F.5 to compute a probabilistic upper bound on $p_{\min}$ that holds with high probability $1 - \alpha$.

---

**Algorithm 3** Monte Carlo inverse certification procedure

---

   **function** INVERSEPROBCERTIFY($\upsilon^{(1)}$, $\upsilon^{(2)}$, $N_1$, $N_2$, $\alpha$)
      $n^* \leftarrow \min\{n \mid \text{BINPVALUE}(n, N_1, \leq, 0.5) < \alpha\}$
      $V^{(1)}, \ldots, V^{(N_2)} \leftarrow \text{SAMPLE}(\upsilon^{(1)}, N_1)$
      $R^{(1)}, \ldots, R^{(N_2)} \leftarrow \text{SORTASCENDING}(\rho(V^{(1)}), \ldots, \rho(V^{(N_2)}))$
      $\overline{\kappa} \leftarrow R^{(n^*)}$                                             ▷ First bound
      $V^{(1)}, \ldots, V^{(N_2)} \leftarrow \text{SAMPLE}(\upsilon^{(2)}, N_3)$
      $\text{count}_2 \leftarrow \sum_{n=1}^{N_2} \mathbb{1}\left[\rho(V) \leq \overline{\kappa}\right]$
   **return** UPPERCONFBOUND($\text{count}_2$, $N_2$, $1 - \frac{\alpha}{2}$)              ▷ Second bound

---

Note that, different from Appendix F.5 we compute a probabilistic upper bound on threshold $\kappa$ by finding the smallest order statistic $R^{(n)}$ such that $\left[\rho(V) \leq R^{(n)}\right] \geq \frac{1}{2}$ holds with high probability.

# J  Parameter space of the tight certificate for rotation invariance in 2D

Our tight certificates for 2D rotation invariance depend on $||\boldsymbol{X}||_2$, $||\boldsymbol{\Delta}||_2$, as well as parameters $\epsilon_1 = \langle \boldsymbol{X}, \boldsymbol{\Delta} \rangle_{\mathrm{F}}$, $\epsilon_2 = \left\langle \boldsymbol{X}\boldsymbol{R}\left(-\frac{\pi}{2}\right)^T, \boldsymbol{\Delta} \right\rangle_{\mathrm{F}}$, which capture the orientation of the perturbed pointcloud, relative to the clean point cloud.

In this section, we determine the feasible range of $\epsilon_1$ and $\epsilon_2$ and calculate the two pairs of values corresponding to adversarial rotations, which is relevant for our experiments in Section 8.1.

## J.1  Feasible parameter range

Let $\hat{\boldsymbol{X}} = \frac{\boldsymbol{X}}{||\boldsymbol{X}||_2}$. Then, parameters $\epsilon_1$ and $\epsilon_2$ can be equivalently stated as follows:

$$\epsilon_1 = \langle \boldsymbol{X}, \boldsymbol{\Delta} \rangle_{\mathrm{F}} = \mathrm{vec}(\boldsymbol{X})^T \mathrm{vec}(\boldsymbol{\Delta}) = \mathrm{vec}(\hat{\boldsymbol{X}})^T \mathrm{vec}(||\boldsymbol{X}||_2 \boldsymbol{\Delta}),$$

$$\epsilon_2 = \left\langle \boldsymbol{X}\boldsymbol{R}\left(-\frac{\pi}{2}\right)^T, \boldsymbol{\Delta} \right\rangle_{\mathrm{F}} = \mathrm{vec}\left(\hat{\boldsymbol{X}}\boldsymbol{R}\left(-\frac{\pi}{2}\right)^T\right)^T \mathrm{vec}(||\boldsymbol{X}||_2 \boldsymbol{\Delta}).$$

Note that $\mathrm{vec}(\hat{\boldsymbol{X}})$ and $\mathrm{vec}\left(\hat{\boldsymbol{X}}\boldsymbol{R}\left(-\frac{\pi}{2}\right)^T\right)$ have norm 1 and are orthogonal to each other. Thus, they are the first two elements of an orthonormal basis of $\mathbb{R}^{2N}$. Parameters $\epsilon_1$ and $\epsilon_2$ are projections of $\mathrm{vec}(||\boldsymbol{X}||_2 \boldsymbol{\Delta})$ onto the first two elements of this basis. Basis changes preserve the norm of vectors. Thus, the values $\epsilon_1, \epsilon_2$ must fulfill $\sqrt{\epsilon_1^2 + \epsilon_2^2} \leq ||\boldsymbol{X}||_2 \cdot ||\boldsymbol{\Delta}||_2$.

## J.2  Adversarial rotations

**Known rotation angle.** Consider an arbitrary $\boldsymbol{X} \in \mathbb{R}^{N \times 2}$. Further consider an adversarially perturbed $\boldsymbol{X}' = \boldsymbol{X}\boldsymbol{R}(\theta)^T$ that is the result of rotating all rows of $\boldsymbol{X}$ by angle $\theta$. Let $\boldsymbol{\Delta} = \boldsymbol{X}' - \boldsymbol{X}$. Recall that, for any vector $\boldsymbol{a}, \boldsymbol{b} \in \mathbb{R}^D$, $\boldsymbol{a}^T \boldsymbol{b}^T = ||\boldsymbol{a}||_2 ||\boldsymbol{b}||_2 \cos(\angle \boldsymbol{a}\boldsymbol{b})$. Therefore

$$\epsilon_1 = \langle \boldsymbol{X}, \boldsymbol{\Delta} \rangle_{\mathrm{F}} = \sum_{n=1}^N \boldsymbol{X}_n^T (\boldsymbol{X}_n' - \boldsymbol{X}_n) = \sum_{n=1}^N ||\boldsymbol{X}_n^T||_2^2 \cos(\theta) - ||\boldsymbol{X}_n^T||_2^2 = ||\boldsymbol{X}||_2^2 (\cos(\theta) - 1),$$

where the last equality follows from the definition of the Frobenius norm. Similarly, we have

$$\epsilon_2 = ||\boldsymbol{X}||_2^2 (\cos(\theta + \pi/2) - \cos(\pi/2)) = -||\boldsymbol{X}||_2^2 \sin(\theta).$$

**Unknown rotation angle.** Now assume that we know $||\boldsymbol{X}||_2$ and $||\boldsymbol{\Delta}||_2$, but do not know the rotation angle $\theta$. We have

$$\begin{aligned}
||\boldsymbol{\Delta}||_2^2 &= \left\langle \boldsymbol{X}\boldsymbol{R}(\theta)^T - \boldsymbol{X}, \boldsymbol{X}\boldsymbol{R}(\theta)^T - \boldsymbol{X} \right\rangle_{\mathrm{F}} \\
&= ||\boldsymbol{X}\boldsymbol{R}(\theta)^T||_2^2 + ||\boldsymbol{X}||_2^2 - 2\left\langle \boldsymbol{X}, \boldsymbol{X}\boldsymbol{R}(\theta)^T \right\rangle_{\mathrm{F}} \\
&= ||\boldsymbol{X}||_2^2 + ||\boldsymbol{X}||_2^2 - 2||\boldsymbol{X}||_2^2 \cos(\theta) \\
&= 2||\boldsymbol{X}||_2^2 (1 - \cos(\theta)).
\end{aligned}$$

Evidently, such an adversarial rotation is only possible if $||\boldsymbol{\Delta}|| \leq 2||\boldsymbol{X}||$, which corresponds to a rotation by angle $\pi$. Solving for $\theta$ yields

$$\theta = \pm \arccos\left(1 - \frac{||\boldsymbol{\Delta}||_2^2}{2||\boldsymbol{X}||_2^2}\right)$$

Inserting into our formulae for $\epsilon_1$ and $\epsilon_2$ and using that $\sin(\arccos(a)) = \sqrt{1 - a^2}$ shows that

$$\epsilon_1 = -\frac{1}{2}||\boldsymbol{\Delta}||_2^2,$$

$$\epsilon_2 = \pm \frac{1}{2}\sqrt{||\boldsymbol{\Delta}||_2^2 \left(4||\boldsymbol{X}||_2^2 - ||\boldsymbol{\Delta}||_2^2\right)}.$$