# OpenReview forum: "Invariance-Aware Randomized Smoothing Certificates"
_NeurIPS.cc/2022/Conference — NeurIPS 2022 Accept_

### Official Review · Reviewer_nK8B · 2022-06-27

**Rating:** 5
**Confidence:** 4
**Soundness:** 3 good
**Presentation:** 2 fair
**Contribution:** 2 fair

**Summary:**

This paper proposes an approach, leveraging the invariance property of the base classifier to enhance the randomized smoothing certificate.

**Questions:**

1. Is Eq. (91) in Lemma 14 (appendix) wrong? It might be a typo. I think the left-hand side is the same as the right-hand side.
2. Is group averaging (line 222) a common technique for improving classification performance, or just for the invariance property in the field of 2D/3D point cloud classification?
3. This paper should include more discussions on the connection to [Li et al., 2021], which is also an important work on transformation-specific smoothing

[Li et al., 2021] Tss: Transformation-specific smoothing for robustness certification, CCS 2021.

**Ethics Review Area:**

["I don’t know"]

**Limitations:**

The contribution of this paper is limited since invariant models are rarely studied for some reasons (might be the poor performance or the limited generalization ability?).

**Strengths And Weaknesses:**

Strength:
1. The paper is quite complete.
2. The presentation is good.

Weaknesses:
1. The contribution of this paper is a bit incremental. Specifically, Theorem1,2,5 are obvious to me.
2. The significance of invariance-aware randomized smoothing is limited. The neural network classifiers that satisfy invariance are extremely rare, so the author needs to construct invariant versions of PointNet and DGCNN via some tricks.

---

> ### Author Response · Authors · 2022-08-01
> **Rebuttal Reviewer nK8B**
>
> We would like to thank the Reviewer for their helpful comments and questions, which have allowed us to further improve upon our submission.
>
> Before going through them, let us first address your two main comments concerning the paper's quality:
>
> ## Comments
>
> ### The contribution of this paper is a bit incremental. Specifically, Theorem1,2,5 are obvious to me.
>
> Firstly, we  disagree with the assessment that this paper was incremental. No prior work has studied the interplay of invariance and certifiable robustness.
>
> Theorems 1 and 2 are very straight-forward, but it is important to consider them within  the overall context of our paper.
> Since no one has studied the certification of invariant models before, we first propose the easily understandable "post-processing-based certificate" (Theorems 1 and 2).
> This method serves as a baseline against which we can compare the provably tight certificates we derive in Section 6 (which, aside from studying invariance and certification for the first time, are our main contribution).
>
> We have updated the introduction to clarify this from the very beginning.
>
> We strongly disagree with Theorem 5 being obvious.
> Only the right-hand side of Theorem 5 is obvious, because it can be obtained by simply applying Theorem 2 to translation invariance.
> What makes Theorem 5 interesting is that it is derived using our method for tight certification from Section 6.1 (that is why we have an equality with $\min_{h \in \mathbb{H}_T} \Pr\left[h(Z) = y^*\right]$, where $\mathbb{H}_T$ is the set of all translation-invariant classifiers).
> The interesting and non-trivial aspect is that this very simple certificate is actually tight -- it is impossible to find a better one.
>
> ### The significance of invariance-aware randomized smoothing is limited. The neural network classifiers that satisfy invariance are extremely rare, so the author needs to construct invariant versions of PointNet and DGCNN via some tricks.
> Invariant classifiers are not rare at all (unless one is exclusively interested in a few specific computer vision tasks for planar images -- and ignores the translation-invariance of CNNs).
>
> Invariant neural networks have been extensively and actively studied since the publication of "Group Equivariant Convolutional Networks" [[1]](https://proceedings.mlr.press/v48/cohenc16.html) in 2016.
> More importantly, invariant and equivariant models have become the de-facto standard in multiple large and active fields.
> For instance:
> * Message-passing GNNs are the standard for graph classification and invariant to graph isometries.
> * Virtually all point cloud classifiers since PointNet are built around permutation invariance.
> * Point-cloud models that are equivariant or invariant under spatial transformations (which we study in this submission), have been the SOTA for tasks like molecular property prediction, molecular dynamics or drug discovery for years (see, for instance, this leaderboard for [QM9](https://paperswithcode.com/sota/formation-energy-on-qm9) and the papers for models like [MPNN](https://proceedings.mlr.press/v70/gilmer17a.html), [SchNet](https://arxiv.org/abs/1706.08566), [GemNet](https://arxiv.org/abs/2106.08903), [SpookyNet](https://arxiv.org/abs/2105.00304) or [PaiNN](https://arxiv.org/abs/2102.03150)).
>
> See also Section 1 of our submission.
>
> Furthermore, we would find it questionable to dismiss the work of Li et al. (ICCV 21) [[2]](https://openaccess.thecvf.com/content/ICCV2021/html/Li_A_Closer_Look_at_Rotation-Invariant_Deep_Point_Cloud_Analysis_ICCV_2021_paper.html) and Xiao et al. (ICME 20) [[3]](https://ieeexplore.ieee.org/abstract/document/9102947)  that the models in our experiments are based on as "some tricks".
> We opted for these specific models because all prior work on (non-invariant) randomized smoothing for point clouds also used PointNet and DGCNN as base classifiers.
>
> Given the ubiquity of invariances in machine learning (and the on-going interest in robust machine learning), we believe the joint study of both aspects to indeed be significant.

---

> > ### Author Response · Authors · 2022-08-01
> > **Rebuttal Reviewer nK8B - Questions**
> >
> > ## Questions
> > ### Is Eq. (91) in Lemma 14 (appendix) wrong? It might be a typo.
> > Thank you for pointing out that typo. The right-hand side should have $\mathbb{H}$ (the set of all classifiers) instead of $\mathbb{H}_T$ (the set of all invariant classifiers).
> > We have corrected it in the updated submission.
> >
> > ### Is group averaging (line 222) a common technique for improving classification performance, or just for the invariance property in the field of 2D/3D point cloud classification?
> > Group-averaging is a technique for achieving invariance for arbitrary data types and invariances. As discussed above, invariant models are often more accurate than models without such inductive biases.
> >
> > ### This paper should include more discussions on the connection to [Li et al., 2021],
> > While our original submission already cited (Li et al., 2021), we have expanded our discussion of transformation-specific certification.
> > We now distinguish between white-box and black-box transformation-specific certificates (including that of Li et al. 2021) and highlight the unique aspects of different randomized-smoothing-based methods.
> >
> > ---
> >
> > We hope that we have addressed all of your comments and questions to your satisfaction.
> > We further hope that we have been at least somewhat successful in convincing you that equivariance and invariance do indeed play a significant role in the larger machine learning community.
> >
> > Please let us know if you have any further questions during the discussion period.
> >
> > [1] [Group Equivariant Convolutional Networks](https://proceedings.mlr.press/v48/cohenc16.html); Taco Cohen, Max Welling; ICML 2016
> > [2] [A Closer Look at Rotation-Invariant Deep Point Cloud Analysis](https://openaccess.thecvf.com/content/ICCV2021/html/Li_A_Closer_Look_at_Rotation-Invariant_Deep_Point_Cloud_Analysis_ICCV_2021_paper.html); Feiran Li, Kent Fujiwara, Fumio Okura, Yasuyuki Matsushita; ICCV 2021
> > [3] [Endowing Deep 3D Models with Rotation Invariance Based on Principal Component Analysis](https://ieeexplore.ieee.org/abstract/document/9102947); Zelin Xiao; Hongxin Lin; Renjie Li; Lishuai Geng; Hongyang Chao; Shengyong Ding; ICME 2020

---

### Official Review · Reviewer_XHgH · 2022-07-12

**Rating:** 5
**Confidence:** 3
**Soundness:** 3 good
**Presentation:** 2 fair
**Contribution:** 2 fair

**Summary:**

The paper develops a number of robustness certificates for models applied on spatial data (e.g. point-clouds). It builds upon prior work on using randomized smoothing to derive provable robustness certificates for models in a black box manner, and extends the scope of certificates by leveraging information about model invariances to create what the authors call "gray-box certificates". For example, if a model is known to be translation invariant, then any input that can be moved to a region covered by a certified to receive a specific output, will also receive the same output under the model.
Using this approach, the paper introduces certificates leveraging invariances to Euclidean isometries (translations, rotations, reflections) and and permutations. It proves that the certificates for translation and certain rotation transformations are tight and conducts empirical experiments to support the findings as well.

**Questions:**

- In theorem 6, shouldn't g have the property of being rotation instead of translation invariant?
- I am not an expert in the spatial-data domain, so I am not sure how realistic for instance attacks using point translations or rotations are, it might be worth to discuss the threat-model or other ways in which such transformations might enter the test data in more detail.
- Could you clarify why the set of invariance transformations need to form a group?

**Limitations:**

The limitations of the paper seem to be sufficiently addressed.

**Strengths And Weaknesses:**

### Strengths
- Robustness certificates are usually used to prove that certain invariances hold for a model. Using a-priori knowledge about invariances to increase the scope of provable (derived) robustness is an interesting avenue.
- The paper conducts a solid theoretical analysis and further supports its conclusions using empirical evidence.

### Weaknesses
- The applicability of the work might be somewhat limited, because it is difficult to construct models that are provably robust to certain input transformations, and not just weakly invariant due to training. The certificates developed in the paper only hold when strict invariance can be assumed.
- There are some clarity issues, especially regarding the different types of certificates. There are a number of different certificates (black-box, gray-box, post-processing, pre-processing, tight), and it's difficult to keep track of their meanings relationships. The different types of certificates should be clearly defined (ideally already in the intro, at least roughly) and delineated. For instance, it's not clear to me what the role of the pre-processing certificates is? Are they more than post-processing certificates proven to be tight?
- A central quantity in the theoretical analysis is the probability $p_{X', y^\ast}$ (line 92) of predicting the target class under randomized smoothing, which is used to bound classification probabilities. However, in the original randomized smoothing work, this quantity is compared to the probability of the next most likely class, which is absent from the analysis here. As far as I can see the workaround is to assume that $p_{X', y^\ast} > 0.5$, though this seems like an overly strict assumption in a multi-class scenario. Wouldn't comparing $p_{X', y^\ast}$ against the next most likely probability allow for deriving tighter bounds? Or is the analysis limited to binary classification? Footnote 2 seems to suggest something like that, but the definition of g only mentions a general label set $\mathbb{Y}$. Please clarify this.

---

> ### Author Response · Authors · 2022-08-01
> **Rebuttal Reviewer XHgH**
>
> We would like to thank the reviewer for the detailed comments on our submission and are glad to hear that they also find the combination of robustness certfication and invariant machine learning interesting.
>
> In the following, we will first address your three comments concerning potential weaknesses of our paper, before responding to your additional questions.
>
> ## Comments
>
> ### The applicability of the work might be somewhat limited, because it is difficult to construct models that are provably robust to certain input transformations [...]
> While the construction of invariant and equivariant models is certainly challenging, it has been extensively and actively studied since the publication of "Group Equivariant Convolutional Networks" in 2016 [[1]](https://proceedings.mlr.press/v48/cohenc16.html).
> At this point, the ways of enforcing these properties are theoretically well-established (e.g. equivariant linear layers are necessarily group convolutions [[2]](https://proceedings.mlr.press/v80/kondor18a.html)) and there exist methods to build models that are invariant to arbitrary symmetry groups (see, for instance [[3]](https://openreview.net/forum?id=WE4qe9xlnQw)).
>
> More importantly, invariant and equivariant models have become the de-facto standard in multiple large and active fields (outside computer vision for planar images).
> For instance:
> * Message-passing GNNs are the standard for graph classification and invariant to graph isometries.
> * Virtually all point cloud classifiers since PointNet are built around permutation invariance.
> * Point-cloud models that are equivariant or invariant under spatial transformations (which we study in this submission), have been the SOTA for tasks like molecular property prediction, molecular dynamics or drug discovery for years (see, for instance, this leaderboard for [QM9](https://paperswithcode.com/sota/formation-energy-on-qm9) and the papers for models like [MPNN](https://proceedings.mlr.press/v70/gilmer17a.html), [SchNet](https://arxiv.org/abs/1706.08566), [GemNet](https://arxiv.org/abs/2106.08903), [SpookyNet](https://arxiv.org/abs/2105.00304) or [PaiNN](https://arxiv.org/abs/2102.03150)).
>
> See also Section 1 of our submission.
>
> While our method is of course -- by design -- constrained to invariant models, this class of models is interesting, important and popular enough that we would not consider it a weakness.
>
>
> ### There are some clarity issues [...] The different types of certificates should be clearly defined (ideally already in the intro, at least roughly) and delineated. For instance, it’s not clear to me what the role of the pre-processing certificates is?
>
> Post-processing and pre-processing are two views or characterizations of the same certificate.
> The post-processing certificate takes an original certified region $B$ and applies all transformations from the symmetry group to construct a larger certified region $\tilde{B}$.
> The pre-processing formulation simply describes how you would determine whether $X' \in \tilde{B}$ for a specific $X'$, i.e. it is an implicit characterization of region $\tilde{B}$.
>
> The only reason we discuss the pre-processing perspective separately is that allows for a more convenient comparison with the tight certificates we derive in subsequent sections (e.g. in Section 6.2, where we show that the tight and post-processing-based certificates for translation invariance are identical.)
>
> Based on your feedback, we have added a short delineation of the different certificate types to Section 1 and further clarified that post-processing and pre-processing are indeed two views on the same certificate in Section 5.
>
> ### Wouldn't comparing $p_{X',y^*}$ against the next most likely probability allow for deriving tighter bounds? Or is the analysis limited to binary classification?
> The analysis is not limited to binary classification.
> Like in the original randomized smoothing paper [4], there are two valid ways of certifying robustness for an arbitrary number of classes:
> 1. Proving that $p_{X',y^*} > 0.5$, i.e. $y^*$ is more likely than all other classes combined
> 2. Proving that $p_{X',y^*} > \max_{y \neq y^*} p_{X',y}$, i.e. $y^*$ is more likely than the next most likely class.
>
> For the sake of exposition we focus on (1.) throughout the main text. We discuss how to compute version (2.) of all certificates in Appendix F.
> Our original submission referred to (1.) as a "binary certificate", but it is not to be confused with a certificate that is limited to binary classifiers.
>
> To address your comment, we have moved the footnote into the main text and and included the above clarifications. We have further removed the potentially confusing "binary certificate" name from the main text.

---

> > ### Author Response · Authors · 2022-08-01
> > **Rebuttal Reviewer XHgH - Questions**
> >
> > ## Questions
> >
> > ### In theorem 6, shouldn't g have the property of being rotation instead of translation invariant?
> > Thank you for pointing out that typo. We have corrected it in the updated submission.
> >
> > ###  I am not sure how realistic for instance attacks using point translations or rotations are, it might be worth to discuss the threat-model or other ways in which such transformations might enter the test data in more detail.
> > Based on your feedback, we have expanded the paragraph discussing adversarial attacks on point cloud data in Section 2.
> > We now:
> > * highlight that the field has evolved beyond simply adapting methods from the image domain
> > * reference work that demonstrates realistic, physically realizable atacks (e.g. adversarial examples that can attack models through LiDAR sensors)
> > * reference work on adversarial attacks via parametric transformations (e.g. translations or rotations) and discuss how it relates to our work
> >
> > Note that we are mostly concerned with adversarial attacks that modify points and are *not* translations or rotations.
> > Since our model is invariant, we already know that such attacks can not influence our model's prediction.
> >
> > ### Could you clarify why the set of invariance transformations need to form a group?
> > We primarily require the group property so that the inverse of each transformation is also an element of the group. This is needed for our derivation of Theorem 3 (see Appendix C.3). The inverses are also necessary to ensure that the post-processing and pre-processing perspective describe the same certificate. Otherwise, there may be points $X'$ that can be reached by applying transformations from group $T$ to a certified region $B$ (post-processing perspective) but not mapped back into $B$ (pre-processing perspective).
> >
> > Aside from that, most commonly studied invariances (translation, rotation, reflections, permututation, graph isometries ...) correspond to groups and the group formalism is used in most literature on invariant machine learning.
> >
> > ---
> >
> > We hope that we have addressed all comments and questions to your satisfaction. Please let us know if you have any further questions during the discussion period.
> >
> > [1] [Group Equivariant Convolutional Networks](https://proceedings.mlr.press/v48/cohenc16.html); Taco Cohen, Max Welling; ICML 2016
> > [2] [On the Generalization of Equivariance and Convolution in Neural Networks to the Action of Compact Groups](https://proceedings.mlr.press/v80/kondor18a.html); Risi Kondor, Shubhendu Trivedi, ICML 2018
> > [3] [A Program to Build E(N)-Equivariant Steerable CNNs](https://openreview.net/forum?id=WE4qe9xlnQw); Gabriele Cesa, Leon Lang, Maurice Weiler; ICLR 2022
> > [4] [Certified Adversarial Robustness via Randomized Smoothing](https://proceedings.mlr.press/v97/cohen19c.html); Jeremy M Cohen, Elan Rosenfeld, J. Zico Kolter ; ICML 2019

---

### Official Review · Reviewer_nCYS · 2022-07-15

**Rating:** 6
**Confidence:** 3
**Soundness:** 3 good
**Presentation:** 3 good
**Contribution:** 3 good

**Summary:**

This paper considers the issue of certifiably-robust machine learning
predictors in the 'randomized smoothing' framework of Cohen et al. (take a
classifier $g$; add isotropic gaussian noise to the input to generate a
distribution over predictions $g(x + \varepsilon)$, and let $f$ denote the
predictor that selects the most probable output; prove robustness guarantees
about $f$ using Neyman-Pearson), but with the additional consideration that the
base classifier $g$ is invariant to certain transformations. The authors focus
on the case where the input to $g$ is a batch of physical data $\mathbf{X} \in
\mathbb{R}^{N \times D}$ in $\mathbb{R}^D$ (e.g., point clouds, molecular data, ...), and
the group actions are rigid motions acting on the rows of $\mathbf{X}$ and
permutations of the rows. To make the randomized smoothing approach
computational, probabilities are estimated in a Monte-Carlo fashion and
robustness guarantees are probabilistic; the authors extend the same
computational procedures to the 'invariant $g$' setting by arguing that for the
specific invariances they consider, the randomized smoothing procedure also
yields an invariant classifier $f$, and that the certification procedure on a
new test input can be done efficiently (minimization over the group action for
this family of invariances can be done simply in closed form, e.g. mean
subtraction, orthogonal procrustes). These extensions give exact (the authors
use 'tight') certificates for translation invariance, meaning that the
robustness guarantee matches the lower bound given by Neyman-Pearson; for more
general transformations there is some looseness, and the authors provide
theoretical studies of the gap in the case of 2D rotations and experiments
which show that the gap is often small (but there is extra Monte Carlo
complexity/error associated with the tight certificate approach here that makes
some experimental results contrary to what one would expect).



**Questions:**

- I would appreciate some clarifications on the contributions and the
  relationship to prior work. The contributions claimed at line 40 are quite
  narrow -- is this also the first work that takes up the systematic study of
  randomized smoothing combined with invariant classifiers? The section in the
  related work at line 52 seems to suggest this is the case, but doesn't claim
  it directly.

- The expressions in Theorem 7 seem a bit too complex to easily parse, even
  when looking at the full details in the appendix -- one is curious whether
  some pathological behavior emerges for 'strange' values of $\mathbf{X}$,
  $\mathbf{X}'$, and $\boldsymbol{\Delta}$; whether there is a 'typical'
  setting (random? incoherent?) where it is possible to get more insight into
  the result's predictions; etc. If the authors can point to any details in the
  appendices that help interpret these issues + could be integrated into the
  main body, this could help.



**Limitations:**

Yes. The authors provide a concrete Limitations and Broader Impact section
(Section 7).


**Strengths And Weaknesses:**


## Strengths

- Invariant models for computing with point cloud data are becoming more and
  more prevalent, and understanding issues of robustness with these models and
  their connection to certified robustness seems timely. This paper seems to
  initiate this line of work.
- The paper is well-written, with ample details on background material,
  derivations, and technical points. The appendices are extensive and catalogue
  conceptual/computational aspects of the experiments.
- It seems to be a useful methodological contribution to show that it is
  'enough' in some practical cases to use the simple post-processing approach
  to certification, even if it is not Neyman-Pearson-exact (although see some
  caveats below).

## Weaknesses

- The contributions here seem somewhat narrow -- it would be better if theory
  and experiments relevant to the case of 3D rotations were presented, given
  that this seems to be most important for the applications used as motivation
  (point cloud nets, molecular prediction, ...). Most of the experiments are
  used to motivate the idea that the 'post-processing' invariance approach is
  comparable to the tight certificates approach for 2D rotations; one would
  like to see that this continues to hold in 3D.
- It seems strange that the tight certificates approach does not yield superior
  accuracy in Figure 6.
- The theorems and derivations throughout the paper are presented in a
  formalism-heavy way that often belies significant underlying conceptual
  simplicity. I think the clarity of the paper could be improved as a whole by
  also mentioning some of these simple conceptual intuitions: for example,
  eventually the post-processing approach that is argued to be sufficient
  (Theorem 2) is quite intuitive (it might be nice to have an Algorithm
  environment somewhere that summarizes exactly what approach the authors are
  ultimately proposing that one takes away from the paper, to have a useful
  methodological contribution -- with all the competing approaches it is easy
  to get a bit lost reading through); the proof of Theorem 3 essentially only
  depends on rotation invariance of the gaussian distribution; the 'canonical
  representation' formalism from definition 3 (is this a standard term?) and
  its instantiations throughout section 6 seem to be understandable in terms of
  whether or not you can quotient $\mathbb{R}^{N \times D}$ in a sufficiently nice
  sense by the action of the symmetry group; the observation discussed in the
  paragraph at line 217 can probably be explained using some classical insights
  from invariant theory.
- It seems that generalizing this to images will be infeasible in general,
  given that invariances one is interested in here act on the image plane
  rather than on the image values, which means that the analogue of Theorem 3
  will not transfer. (c.f. Section 7) More generally, one imagines it would be
  challenging to extend this approach to any similar 'homogeneous space'
  setting of the invariant group action.

## Minor Issues

- Line 45: it seems perhaps premature to call a method from 2019 "classic".
- Line 84: spurious period
- Line 91: should this line 'First, one can use the fact that ...' state
  $f(\mathbf{X}) = y^\ast$ rather than $\mathbf{X'}$? The discussion below
  seems to demand that this is true.
- Line 92: I would recommend stating the restriction to binary classification
  outside of a footnote, as it is done systematically throughout the paper.
  In appendix F, it would make sense to reference the general version of the
  Neyman-Pearson calculation proved in [27] for general values of the
  probabilities -- I imagine the current omission won't bother a reader who is
  familiar with randomized smoothing, but I found it necessary to do the
  calculation myself and then reference what was shown in [27] to convince
  myself that something wasn't being omitted here.
- Equation (5): why is there a slash through the bottom line on the $\geq$
  symbol? I do not understand this result.

---

> ### Author Response · Authors · 2022-08-01
> **Rebuttal Reviewer nCYS**
>
> We would like to thank the reviewer for their in-depth review and their constructive feedback.
>
> Before addressing the other comments and questions, we would like to respond to your first question concerning the contribution and relationship to prior work.
>
>
> # Contributions and prior work
>
> Our submission is the first to study the interplay between certifiable robustness and invariance (irrespective of the certification method).
> Thus, the contribution even goes beyond being "the first work that takes up the systematic study of randomized smoothing combined with invariant classifiers".
>
> Perhaps our statement about certificates for "specific operations with invariances" in Section 2 contributed to the lack of clarity concerning our contribution.
> We were trying to explain that neural networks commonly use a few specific operations (e.g. batch norm) that happen to have certain invariances and are compatible with existing white-box certification techniques.
> But their invariances have never been discussed in certifiable robustness literature, let alone used as a tool for certification.
>
> To clarify this, we have updated the list of contributions in Section 1 and the second part of the "orthogonal research directions" paragraph in Section 2.
>
> # Comments
>
> ### [...] it would be better if theory and experiments relevant to the case of 3D rotations were presented [...]
> First, let us reiterate that the post-processing based certificate is compatible with arbitrary Euclidean isometries, including 3D rotations.
> In Appendix A.2.4. we compare it to the tight certificate for elemental / axis-aligned rotations in 3D. The former is (expectedly) not affected by adversarial rotations and more effective than the latter.
> We have updated our experimental section to directly reference this section of the appendix.
>
> Regardless, we agree with you that deriving tight certificates for 3D rotations would be an interesting extension.
> However, we hope that the novelty of our work as a whole and our strong result for translation-invariance  should outweigh the fact that we have not derived this specific result yet.
>
> ###  I think the clarity of the paper could be improved as a whole by also mentioning some of these simple conceptual intuitions: [...]
> Based on your feedback and that of Reviewer XHgH, we have made the following changes to more easily convey certain concepts:
> * More clearly delineate the different types of certificates in Section 1 and 5.
> * Added intuitive explanation of Theorem 3 (how using zero-mean, isotropic noise preserves rotation invariance)
> * Renamed "canonical representation" to "canonical map" (to avoid confusion with group representations)
> * Improved explanation of tight certification approach (focus on how it amounts to finding distribution over orbits/equivalence classes and applying Neyman-Pearson, rather than on technical details)
>
> Concerning your comment about connections between the worst-case classifier and invariant theory: We were sadly unable to find this specific result in existing literature.
> However, the fact that the group-averaging performed by this specific classifiers leads to invariance should be sufficiently intuitive for readers.
>
> ### It seems strange that the tight certificates approach does not yield superior accuracy in Figure 6.
> As you correctly mentioned in your summary, "there is extra Monte Carlo complexity/error associated with the tight certificate approach". We also point this out in l.315-316.
>
> After our initial submission, we have repeated the experiment using $10\times$ as many samples and observed the tight Monte Carlo certificate to converge towards the post-processing certificate (https://figshare.com/articles/figure/Tight_certificate_with_more_MC_samples/20412063).
> Given the tight confidence bounds obtained by using this large number of samples, we expect the "true" (non-Monte Carlo) certificate to also
> be very close to the post-processing-based one.
>
> For the camera-ready version, we plan to repeat all experiments with a larger number of samples.
>
> ### It seems that generalizing this to images will be infeasible in general [...]
> You are right, the current approach is not applicable to symmetry groups acting on images or other signals defined over some homogenous space (aside from the translation certificate, which could be applied to intensity-invariant models).
>
> But, while computer vision is one important application domain, there are various other active fields (particularly ML for science and medicine) that work with spatial data and in which invariant models are much more prevalent (see also Section 1).
>
> We believe that, since this is the first forray into bringing together model invariance and certification, we can be excused from focusing on one (important) domain, rather than trying to tackle invariant machine learning as a whole.
> We do of course expect future work to generalize to other domains (either using randomized smoothing or some other, more suitable paradigm).

---

> > ### Author Response · Authors · 2022-08-01
> > **Rebuttal Reviewer nCYS - Minor Issues and Questions**
> >
> > # Minor issues
> >
> > * l.45: We removed the sentence about "classic" randomized smoothing
> > * l.84: We removed the spurious period
> > * l.91: No. We are interested in determining whether $f(X') = y^*$. To clarify this, we have updated the conditional clause to "if, for perturbed input $X'$, class $y^*$ is more likely than all other classes combined".
> > * l.92: We have moved the footnote about binary vs multi-class certificates into the main text. Note that the "binary certificate" is valid for arbitrary models, not just binary classifiers.
> > * Appendix F: We were not sure what you meant with "Neyman-Pearson calculation proved in [27] for general values of the probabilities". We suppose you meant how Neyman-Pearson can be used to prove both upper and lower bounds and have thus restated both Lemmata in Appendix F, and explicitly written out the black-box multiclass certificate from [27].
> > * Eq.5: The slash through the bottom of $\gneq$ was meant to highlight that it is a strict inequality. We have replaced it with a normal "$>$".
> >
> > # Questions
> > ### I would appreciate some clarifications on the contributions and the relationship to prior work.
> > See "Contributions and prior work" above.
> >
> > ### "The expressions in Theorem 7 seem a bit too complex to easily parse [...] -- one is curious whether some pathological behavior emerges for 'strange' values ; whether there is a 'typical' setting (random? incoherent?) "
> > We are sorry, but we are not entirely sure what you are interested in.
> >
> > Assuming you are interested in whether we have any insights into the functional dependence between the certificates output and its parameters:
> > 1. The certificate guarantees complete robustness for arbitrary adversarial rotations, i.e. $\Delta = R X - X$. We derive which specific parameter values corresponding to adversarial rotations in Appendix H.2.
> > 2. When $|\Delta|$ is large relativ to $|X|$, then $X'$ cannot be the result of adversarial rotations and we no longer have this nice special case (see also Appendix H.2 and Fig. 9d/e).
> > 3. The certificate appears to be symmetric w.r.t. parameter $\epsilon_4$, i.e. only depend on its magnitude (see Fig. 2).
> >
> > Note that our method is not the only randomized smoothing certificate that does not have a closed-form analytical expression that can be easily analyzed. For instance, certificates for discrete data require solving a linear program (see [1] and [2]).
> >
> > ---
> >
> > We hope that we have addressed all comments to your satisfaction and would be happy to respond to any further questions during the discussion period.
> >
> > ---
> >
> > [1] [Tight Certificates of Adversarial Robustness for Randomly Smoothed Classifiers](https://proceedings.neurips.cc/paper/2019/hash/fa2e8c4385712f9a1d24c363a2cbe5b8-Abstract.html) ; Guang-He Lee, Yang Yuan, Shiyu Chang, Tommi Jaakkola ; NeurIPS 2019
> > [2] [Efficient Robustness Certificates for Discrete Data: Sparsity-Aware Randomized Smoothing for Graphs, Images and More](https://proceedings.mlr.press/v119/bojchevski20a.html) ; Aleksandar Bojchevski, Johannes Gasteiger, Stephan Günnemann ; ICML 2020

---

> > > ### Comment · Reviewer_nCYS · 2022-08-09
> > > **response**
> > >
> > > Dear authors,
> > >
> > > Thank you for your hard work in responding exhaustively to my review, and for updating the submission.
> > >
> > > I will raise my rating by a point after the rebuttal and reading the reviews of the other referees. I think this work can be a useful investigative contribution for future studies of this topic, in spite of its relative density, and I am not sure I am convinced that the work is limited because it only applies to exactly-invariant models (as the authors state, you can always use some group averaging or other tricks to convert equivariant models into invariant models -- whether such models are performing state-of-the-art, I am not sure, but I think this is of secondary importance with regards to this submission).
> > >
> > > ### Follow-up questions
> > >
> > > Regarding the discussion around Theorem 7, please don't feel like you need to respond here -- I can take your point that it is not out of the ordinary to assert a theorem that contains an expression that is not easy to parse -- but I was trying to get at whether you can derive any insight (here, I mean 'closed form expressions' that do not involve four different epsilons, general "linear combinations" of the data/perturbation norms, and Bessel functions) into how the quantities in Theorem 7 behave for certain simple probability models. For example, when the points X are i.i.d. gaussian/uniform on the disc/uniform on the unit circle, and \Delta is perhaps i.i.d. gaussian with a controlled scale. I can see that the experiments in Section 8.1 are trying to get at this, and that some insights (e.g. the role of the scale of the data) emerge from this experiment, but it also seems to me that some clear mathematical insights into the 'typical' behavior (i.e., under one of the aforementioned random models) of the tight vs. post certificates could be derived from such a calculation that would be quite useful in understanding the gap. Investigating this in the case where the perturbation is a rotation of the data, as you do in Appendix H, could be a good first step here; note as well that, if it helps, the behavior under 'small rotations' (i.e. linearize the exponential map: $X \mapsto X(I + \Omega)$ where $\Omega$ is skew-symmetric) could be investigated first, as well.

---

### Author Response · Authors · 2022-08-02
**Revision changelog**

# Changelog
Based on the reviewers' comments we have made further improvements to our submission.
Changed or added sentences are highlighted in blue in the updated manuscript.
Line numbers below refer to the updated submission.

### Main changes
* Appendix is now included in the main submission pdf
* Section 1 (Introduction)
    * Delineate certificate types in Section 1 (ll.34-37)
    * Add "first study on interplay of invariance and certifiable robustness" to contributions list (l.44)
* Section 2 (Related work)
    * Expand discussion of prior work and threat models for spatial data (ll.65-73)
    * Expand discussion of transformation-specific randomized smoothing (ll. 81-91)
    * Improve clarity of discussion of "operations with invariances" (ll.93-95)
* Section 3 (Randomized smoothing)
    * Move discussion of binary vs multiclass certificates into main text (ll.109-111)
    * Remove ambiguous "binary certificate" term from main text (ll.109-111)
* Section 5 (Post-processing-based certificates)
    * Stress that pre-processing and post-processing are two alternative characterizations of / views on the same certificate (ll.146-147)
    * Provide intuitive explanation of isotropic smoothing distributions preserving invariance to isometries (ll.165-169)
* Section 6 (Tight gray-box certificates)
    * Improve intuitive explanation of tight certification approach (finding distributions over equivalence classes) (ll.190-195)
* Section 8 (Experimental evaluation)
    * Reference experiment on post-processing-based certificate for 3D rotation invariance from Appendix A.2.4. (ll.327-328)

### Minor changes
* removed sentence calling 2019 randomized smoothing paper "classic"
* removed spurios period (l.101)
* stress that certification requires bounding class probabilities for perturbed input, not the clean input (l.108)
* include Neyman-Pearson upper bound and black-box multiclass certificate in discussion of multi-class certificates (Appendix F)
* Replace "$\gneq$" with $>$ in Theorem 6
* Replaced erroneous "translation invariant" with "rotation invariant" in Theorem 6
* Replaced erroneous "$\mathbb{H}_\mathbb{T}$" with "$\mathbb{H}$" in Eq. 91 of Lemma 14.
* Replaced "canonical representation" with "canonical map" everywhere
* removed a few redundant or unnecessary words / clauses to stay within page limit

---

### Meta-Review · Area_Chair_WwFB · 2022-09-07

**Recommendation:** Accept
**Confidence:** Less certain

**Metareview:**

The paper presents the first study in the literature about the interplay of model invariance and robustness certification. While the approach is novel, the techniques are usual and the scope of the invariances considered are a bit narrow, which makes the paper somewhat borderline. Nevertheless, all reviewers recommend acceptance.

**Award:**

No

---

### Decision · Program_Chairs · 2022-09-14

Accept